# MULTI-LINEAR SUBSPACE DISTANCE: A NEW CRITERION FOR TENSOR FEATURE SELECTION

## ABSTRACT

Feature selection in tensor data poses greater challenges than in vector representations, since it must capture correlations spanning multiple modes rather than treating each mode in isolation. Existing tensor-based methods partially address this but often treat the feature space as a whole, selecting features globally without respecting mode-specific dependencies. This not only overlooks cross-mode interactions but also increases computational burden, as all features must be considered at once. Moreover, they lack a principled criterion for preserving the global structure of the original tensor. In this work, we introduce *Multi-Linear Subspace Learning Feature Selection* (MSLFS), a framework that overcomes these limitations by distributing feature selection across modes. Specifically, MSLFS selects a small number of representative slices along each mode, whose intersections yield the most informative features. The core innovation is a *multi-linear subspace distance*, which provides a principled measure of how well these selected features preserve the global multi-way structure of the data, while significantly reducing redundancy and computational cost. This objective is complemented by two novel regularizations: a *joint sparsity* constraint that enforces coordinated sparsity across modes to identify compact, non-redundant features, and a *higher-order graph* constraint that preserves local manifold geometry within the induced subtensor. Taken together, these components guarantee that the overall tensor structure as well as the local neighborhood relationships are preserved. Comprehensive experiments on image recognition and biomedical benchmarks demonstrate that MSLFS consistently surpasses state-of-the-art feature selection techniques in clustering tasks.

## 1 INTRODUCTION

Subspace learning has long served as a foundation for dimensionality reduction, with PCA (Zass & Shashua, 2006), LDA (Jelodar et al., 2019), and their variants (Song et al., 2025; Li et al., 2025) producing low-dimensional embeddings that preserve informative directions. However, these methods operate on vectorized data, discarding multi-way correlations and disrupting the natural geometry of tensorial data such as images and biomedical signals (Liu et al., 2017; Lu et al., 2020). As a result, classical subspace learning often misses key structural dependencies, leading to suboptimal representations for multi-way data (Chouchane et al., 2024).

Recent advances in tensor learning extend linear subspace analysis to multi-way data, enabling models to exploit richer structural information than traditional vector-based methods. Yet, most existing approaches still fall short in how they handle feature selection. In particular, they typically flatten the tensor into a single feature space and select features globally, overlooking the mode-specific dependencies that define the multi-way structure of the data (Chen et al., 2023). This global treatment masks the complementary roles of different modes and forces algorithms to operate over the entire feature set, which becomes computationally expensive in high dimensions. More critically, these methods lack a principled criterion for ensuring that the chosen features preserve the global subspace geometry of the tensor, often capturing only partial correlations (Sheehan & Saad, 2007).

To overcome these challenges, we introduce *Multi-linear Subspace Learning Feature Selection* (MSLFS), a framework that distributes the selection process across modes rather than treating the feature space as a rigid whole. Instead of picking features globally, MSLFS identifies a small num-

ber of representative slices along each mode; their intersections then form a compact set of features that best reflect the underlying structure of the data. This strategy both respects the multi-way organization of tensor data and reduces computational overhead. At the heart of the framework lies a new notion of *multi-linear subspace distance*, which serves as a principled measure of how well selected features preserve the original multi-way geometry. By optimizing this criterion, MSLFS ensures that the chosen features jointly capture mode-specific information and cross-mode dependencies.

Beyond the core formulation, we introduce two regularizers. The *joint sparsity* term enforces shared sparsity across modes, ensuring that only a compact and representative subset of features is retained. The *higher-order graph* term preserves local manifold geometry in the selected subtensor by extending neighborhood smoothness across all modes. Together, these constraints balance sparsity, global structure, and local geometry. In summary, the contributions of this work are presented as follows.

- A distributed selection strategy is designed to operate across tensor modes, where a small set of representative slices is chosen per mode. Informative features are yielded by their intersections, which respect the multi-way structure while reducing computational cost.
- A novel *multi-linear subspace distance* is introduced, providing a principled criterion by which the preservation of the global subspace structure across all tensor modes by the selected features is evaluated.
- A *joint sparsity* constraint is proposed to act simultaneously across multiple tensor modes, whereby a compact and non-redundant subset of features is encouraged while preserving the overall data structure.
- A *higher-order graph* regularization is proposed, through which neighborhood smoothness is extended to tensor data so that local manifold structures are preserved in the reduced representation.

## 2 RELATED WORK

**Vector-Based Unsupervised Feature Selection.** Unsupervised feature selection has been widely studied, though most methods target vectorized data rather than multi-dimensional structures. A summary of the most recent methods is presented as follows: ESUFS (Huang et al., 2025) mitigates the sensitivity and structural inconsistencies of graph-based models by jointly learning a discrete similarity graph and an indicator matrix, ensuring correct connectivity while emphasizing naturally discriminative features. UFS-CGL (Zhou et al., 2024) further improves graph-guided selection by preserving class-specific structure through contrastive affinity learning and an $\ell_{1,2}$-regularized projection that suppresses redundant shared features. To overcome the rigidity of linear projections in spectral methods, FOG-R (Chen et al., 2024) replaces hard dimensionality reduction with a flexible optimal graph that jointly optimizes graph learning and $\ell_{2,1}$-regularized feature selection. MRMGRFS (Zuo et al., 2025) addresses the common imbalance between feature relevance and redundancy by combining SCFS, which measures relevance via spectral clustering, with SJGRM, which refines these scores through Jensen–Shannon–based redundancy minimization. NNSE (You et al., 2023b) captures nonlinear feature–label relationships by replacing linear mappings with neural network–based self-expression enhanced by adaptive graph regularization. SDAE (Hassanieh & Chehade, 2024b) employs a deep autoencoder with a selective layer that identifies a compact set of features sufficient for reconstructing the original space, enabling nonlinear, globally representative, and fully unsupervised feature selection.

**Tensor-Based Unsupervised Feature Selection.** Recently, tensor-based methods have been introduced to overcome the drawbacks of vector-based feature selection, though their use in unsupervised settings remains limited. Among these, two notable approaches have been proposed. GRLTR (Su et al., 2018) integrates low-rank tensor representation, local geometry preservation, and $\ell_{2,1}$-norm feature selection, while CPUFS (Chen et al., 2023) combines a tensor-oriented linear classifier, graph-regularized non-negative CP decomposition, and pseudo-label regression. However, these methods still treat the feature space as flat, selecting features globally without considering mode-specific dependencies, which increases computational cost. Our approach instead selects a few representative slices from each mode, whose intersections yield the most informative features. This preserves the multi-way structure, reduces complexity, and ensures the selected features better capture the global data structure.

**Notations.**   For clarity, symbols used in this paper are summarized in Table 1, with detailed descriptions and preliminaries in Appendix 7.1.

Table 1: Summary of notations.

| Notation | Meaning |
|---|---|
| $x, \mathbf{x}, \mathbf{X}, \mathcal{X}$ | Scalar; vector; matrix; tensor. |
| $\mathbf{I}_m, \mathbf{e}_j^{(m)}$ | Identity matrix; $j$-th column. |
| $\mathbf{A}_{i,:}, \mathbf{A}_{:,j}$ | $i$-th row; $j$-th column of $\mathbf{A}$. |
| $\|\mathbf{A}\|_F, \|\mathbf{A}\|_{2,1}, \mathrm{Tr}(\mathbf{A})$ | Frobenius norm; $\ell_{2,1}$-norm; trace. |
| $\langle \mathbf{u}, \mathbf{v} \rangle, \langle \mathbf{A}, \mathbf{B} \rangle_F$ | Dot product; Frobenius inner product. |
| $\mathbf{A} \odot \mathbf{B}, \mathbf{A} \oslash \mathbf{B}, \mathbf{A} \otimes \mathbf{B}$ | Hadamard product; element-wise division; Kronecker product, where $(\mathbf{A} \otimes \mathbf{B})_{\overline{im},\overline{jn}} = a_{ij}b_{mn}$. |
| $\mathcal{X} \in \mathbb{R}^{I_1 \times \cdots \times I_N}, \mathcal{X}_j^{(N)}, \mathcal{X}_{(N)}$ | $N$-mode tensor, with $I_N$ samples and $I_1 \times \cdots \times I_{N-1}$ features; $j$-th frontal slice; mode-$N$ unfolding. |
| $\mathcal{X} \times_n \mathbf{A}, \mathcal{X} \bar{\times}_n \mathbf{v}$ | $n$-mode tensor–matrix; tensor–vector products. |
| $\mathrm{Ind}^{I_1 \times I_2}; \mathbb{R}_+$ | Indicator matrix; Set of non-negative real numbers. |

## 3  MULTI-LINEAR SUBSPACE LEARNING

In tensor analysis, multi-linear subspace learning maintains multi-mode structure instead of flattening data (Lu et al., 2011). A major challenge is defining a geometry-aware distance between subspaces spanned by tensor slices across modes. The goal of this section is to define a multi-linear subspace distance which quantifies similarities between these slice-based subspaces, preserving cross-mode dependencies and discriminative information. To this end, we first establish the formal definition of the subspace spanned by tensor slices, which serves as the basis for a similarity measure that precisely captures the underlying multi-linear relationships.

**Definition 1.** Let $\mathcal{X} \in \mathbb{R}^{I_1 \times I_2 \times \cdots \times I_N}$ be an $N$-mode tensor with the mode-$n$ slices $\mathcal{X}_1^{(n)}, \ldots, \mathcal{X}_{I_n}^{(n)}$, where $n \in \{1, 2, \ldots, N\}$. The space spanned by $\mathcal{X}^{(n)} = \{\mathcal{X}_i^{(n)}\}_{i=1}^{I_n}$ is denoted by $\mathcal{S}(\mathcal{X}^{(n)})$ and defined as $\mathcal{S}(\mathcal{X}^{(n)}) = \{\sum_{i=1}^{I_n} \alpha_i^{(n)} \mathcal{X}_i^{(n)} \mid \alpha_i^{(n)} \in \mathbb{R}\}$. Here, $\alpha^{(n)} = [\alpha_1^{(n)}, \alpha_2^{(n)}, \ldots, \alpha_{I_n}^{(n)}]^\top \in \mathbb{R}^{I_n}$ denotes the vector of scalar coefficients corresponding to the mode-$n$ slices.

This construction associates each set of tensor slices with a multi-linear subspace, turning the problem of comparing tensor data into a problem of comparing subspaces. To proceed, we need a principled way of measuring how close an external tensor is to such a subspace.

**Definition 2.** Given $\mathcal{X} \in \mathbb{R}^{I_1 \times I_2 \times \cdots \times I_N}$ and a tensor $\mathcal{Z}$ of the same dimension as a mode-$n$ slice of $\mathcal{X}$, where $n \in \{1, 2, \ldots, N\}$, the distance from $\mathcal{Z}$ to $\mathcal{S}(\mathcal{X}^{(n)})$ is defined as $\mathrm{dist}(\mathcal{Z}, \mathcal{S}(\mathcal{X}^{(n)})) = \min_{\mathcal{W} \in \mathcal{S}(\mathcal{X}^{(n)})} \|\mathcal{Z} - \mathcal{W}\|_F$.

This distance corresponds to the minimum discrepancy between $\mathcal{Z}$ and any element of the subspace. In other words, it quantifies the error incurred when approximating $\mathcal{Z}$ by linear combinations of the mode-$n$ slices of $\mathcal{X}$. It follows that $\min_{\mathcal{W} \in \mathcal{S}(\mathcal{X}^{(n)})} \|\mathcal{Z} - \mathcal{W}\|_F = \|\mathcal{Z} - \mathrm{Proj}_{\mathcal{S}(\mathcal{X}^{(n)})} \mathcal{Z}\|_F$, where $\mathrm{Proj}_{\mathcal{S}(\mathcal{X}^{(n)})} \mathcal{Z}$ denotes the orthogonal projection of $\mathcal{Z}$ onto the subspace. Since this projection is itself a linear combination of slices, there exists $\alpha^{(n)} = [\alpha_1^{(n)}, \ldots, \alpha_{I_n}^{(n)}]^\top \in \mathbb{R}^{I_n}$ such that $\mathrm{Proj}_{\mathcal{S}(\mathcal{X}^{(n)})} \mathcal{Z} = \sum_{i=1}^{I_n} \alpha_i^{(n)} \mathcal{X}_i^{(n)} = \mathcal{X} \bar{\times}_n \alpha^{(n)}$. Consequently, $\mathrm{dist}(\mathcal{Z}, \mathcal{S}(\mathcal{X}^{(n)})) = \|\mathcal{Z} - \mathcal{X} \bar{\times}_n \alpha^{(n)}\|_F$.

Beyond this general case, additional structure yields simplifications. If the slices $\{\mathcal{X}_i^{(n)}\}_{i=1}^{I_n}$ are orthonormal (i.e., $\langle \mathcal{X}_i^{(n)}, \mathcal{X}_j^{(n)} \rangle_F = 0$ for $i \neq j$ and $\|\mathcal{X}_i^{(n)}\|_F = 1$ for all $i$), the projection coefficients become explicit inner products: $\alpha^{(n)} = [\langle \mathcal{Z}, \mathcal{X}_1^{(n)} \rangle_F, \ldots, \langle \mathcal{Z}, \mathcal{X}_{I_n}^{(n)} \rangle_F]^\top$. In this case, $\mathrm{dist}(\mathcal{Z}, \mathcal{S}(\mathcal{X}^{(n)})) = \|\mathcal{Z} - \sum_{i=1}^{I_n} \langle \mathcal{Z}, \mathcal{X}_i^{(n)} \rangle_F \mathcal{X}_i^{(n)}\|_F$, which admits a simple geometric interpretation as subtracting the projection of $\mathcal{Z}$ onto the orthonormal basis formed by the mode-$n$ slices.

So far we have defined the distance between a single tensor and the subspace spanned by tensor mode-$n$ slices. Beyond this, the concept can be naturally extended to quantify the distance between two subspaces, each spanned by the mode-$n$ slices of two distinct tensors.

**Definition 3** (Multi-linear Subspace Distance). Let $\mathcal{X} \in \mathbb{R}^{I_1 \times I_2 \times \cdots \times I_N}$ be an $N$-mode tensor, and let $\mathcal{Y}$ be another $N$-mode tensor of the same dimensionality, except that its mode-$n$ size equals

$J_n$, where $n \in \{1, 2, \ldots, N\}$. The squared distance between the mode-$n$ subspaces $\mathcal{S}(\mathcal{X}^{(n)})$ and $\mathcal{S}(\mathcal{Y}^{(n)})$ is defined as $\text{dist}(\mathcal{S}(\mathcal{X}^{(n)}), \mathcal{S}(\mathcal{Y}^{(n)}))^2 = \sum_{i=1}^{I_n} \text{dist}(\mathcal{X}_i^{(n)}, \mathcal{S}(\mathcal{Y}^{(n)}))^2$.

It can be shown that $\text{dist}(\mathcal{S}(\mathcal{X}^{(n)}), \mathcal{S}(\mathcal{Y}^{(n)}))^2 = \sum_{i=1}^{I_n} \|\mathcal{X}_i^{(n)} - \mathcal{Y} \bar{\times}_n \alpha_i^{(n)}\|_F^2 = \|\mathcal{X} - \mathcal{Y} \times_n \mathbf{H}^{(n)}\|_F^2$, where $\mathbf{H}^{(n)} \in \mathbb{R}^{I_n \times J_n}$ is such that its $i$-th row is $\alpha_i^{(n)}$. Thus, the distance admits a compact tensor representation via a reconstruction error term. This formulation essentially measures how far each mode-$n$ slice of $\mathcal{X}$ lies from the $\mathcal{S}(\mathcal{Y}^{(n)})$, and aggregates these deviations across all mode-$n$ slices.

**Remark 1.** The concept of multi-linear subspace distance provides a key link between tensor geometry and feature selection. Concretely, let $\mathcal{X} \in \mathbb{R}^{I_1 \times I_2 \times \cdots \times I_N}$ denote the tensor data with $I_N$ samples and $I_1 \times I_2 \times \cdots \times I_{N-1}$ features. Each mode-$N$ fiber represents a single feature and can be seen as the intersection of its corresponding mode-1 through mode-$(N-1)$ slices. Thus, the ability of a fiber to characterize the feature space depends on how well the slices containing that fiber span the subspaces $\mathcal{S}(\mathcal{X}^{(1)}), \ldots, \mathcal{S}(\mathcal{X}^{(N-1)})$. The multi-linear subspace distance provides a natural measure to evaluate this, enabling us to identify informative slices across modes whose intersections yield fibers that faithfully preserve the global structure. By minimizing the distance between the full subspace and the one formed by selected slices, our framework ensures fidelity and coherence across modes. This principle provides the foundation for our feature selection strategy, which will be further developed in the following sections.

### 3.1 SUBTENSORS AND SLICE SELECTION

Building on the idea of multi-linear subspace distance, a natural way to reduce redundancy while preserving structure is to restrict attention to a subset of slices. Such subsets define subtensors, which retain the essential information needed to approximate the span of the full tensor. By working with subtensors, we can formalize slice selection as a principled step in feature selection, preparing the ground for our definition below.

**Definition 4.** For a tensor $\mathcal{X} \in \mathbb{R}^{I_1 \times I_2 \times \cdots \times I_N}$, a subtensor $\mathcal{X}^{(n;k)}$ is obtained by choosing $k$ mode-$n$ slices indexed by $\{i_1^{(n)}, \ldots, i_k^{(n)}\}$, where each $i_j^{(n)} \in \{1, \ldots, I_n\}$ and $n \in \{1, 2, \cdots, N\}$.

Any single mode-$n$ slice $\mathcal{X}_j^{(n)}$ can be written as $\mathcal{X}_j^{(n)} = \mathcal{X} \bar{\times}_n \mathbf{e}_j^{(n)}, \forall j \in \{1, \cdots I_n\}$, where $\mathbf{e}_j^{(n)}$ is the $j$-th column of the identity $\mathbf{I}_{I_n}$. More generally, a subtensor $\mathcal{X}^{(n;k)}$ formed from $\{\mathcal{X}_{i_1}^{(n)}, \ldots, \mathcal{X}_{i_k}^{(n)}\}$ can be expressed as $\mathcal{X}^{(n;k)} = \mathcal{X} \times_n \mathbf{W}^{(n;k)}$, where $\mathbf{W}^{(n;k)} \in \mathbb{R}^{k \times I_n}$ is a selection matrix whose rows are standard basis vectors.

Building on this, the distance between the span of all slices and that of a selected subset follows directly. By Definition 3, we obtain

$$\text{dist}(\mathcal{S}(\mathcal{X}^{(n)}), \mathcal{S}(\mathcal{X}^{(n;k)})) = \|\mathcal{X} - \mathcal{X}^{(n;k)} \times_n \mathbf{H}^{(n;k)}\|_F = \|\mathcal{X} - \mathcal{X} \times_n \mathbf{W}^{(n;k)} \times_n \mathbf{H}^{(n;k)}\|_F$$
$$= \|\mathcal{X} - \mathcal{X} \times_n (\mathbf{H}^{(n;k)} \mathbf{W}^{(n;k)})\|_F. \tag{1}$$

This characterization shows that the distances between full and reduced subspaces can be understood as the error of reconstructing the original tensor using only selected slices and suitable weighting.

### 3.2 CORE REPRESENTATION VIA INTERSECTION FIBERS

The subspace framework developed in (1) can be naturally extended to a compact tensor representation in terms of mode-$N$ fibers. By selecting slices along modes $1, \ldots, N-1$ that span the corresponding mode subspaces $\mathcal{S}(\mathcal{X}^{(1)}), \ldots, \mathcal{S}(\mathcal{X}^{(N-1)})$, we obtain a reduced set of mode-$N$ fibers located at their intersections. These intersection fibers act as structural representatives, capturing the same subspace as the full collection of mode-$N$ fibers. Consequently, the entire tensor can be approximated using a core representation derived from this smaller, more informative subset, whose validity is rigorously established by the following theorem.

**Theorem 3.1.** Let $\mathcal{X} \in \mathbb{R}^{I_1 \times I_2 \times \cdots \times I_N}$ be an $N$-mode tensor. Suppose that, for each mode $n \in \{1, 2, \ldots, N-1\}$, the subspace $\mathcal{S}(\mathcal{X}^{(n)})$ has a basis of dimension $R_n \leq I_n$ with in-

dex set $T_n = \{i_1^{(n)}, \ldots, i_{R_n}^{(n)}\}$. Let $\mathbf{W}^{(n;R_n)} \in \mathrm{Ind}^{R_n \times I_n}$ denote the corresponding indicator matrix. For each $(N-1)$-tuple $(i_1, \ldots, i_{N-1}) \in \{1, \ldots, I_1\} \times \cdots \times \{1, \ldots, I_{N-1}\}$, let $\mathbf{f}_{i_1, \ldots, i_{N-1}} = \mathcal{X}_{i_1, \ldots, i_{N-1},:} \in \mathbb{R}^{I_N}$ denote the mode-$N$ fiber. **(Part I: Core Dictionary).** The $\prod_{n=1}^{N-1} R_n$ intersection fibers $\{\mathbf{f}_{i_{r_1}^{(1)}, \ldots, i_{r_{N-1}}^{(N-1)}}\}_{r_1, \ldots, r_{N-1}=1}^{R_1, \cdots R_{N-1}}$ form a *core dictionary* that spans all mode-$N$ fibers of $\mathcal{X}$. Stacking them columnwise yields the core matrix

$$\mathbf{F}_{\mathrm{core}} = \left(\mathcal{X} \times_{n=1}^{N-1} \mathbf{W}^{(n;R_n)}\right)_{(N)} = \mathbf{X}_{(N)} \bigotimes_{n=1}^{N-1} \mathbf{W}^{(N-n;R_{N-n})^\top}, \tag{2}$$

where $\bigotimes_{n=1}^{N-1} \mathbf{W}^{(N-n;R_{N-n})^\top}$ acts as the indicator matrix selecting precisely those core fibers. **(Part II: Separable Reconstruction).** There exist coefficient matrices $\mathbf{H}^{(n;R_n)} \in \mathbb{R}^{I_n \times R_n}$, $n \in \{1, 2, \ldots, N-1\}$ such that every mode-$N$ fiber admits the separable expansion

$$\mathbf{f}_{i_1, \cdots i_{N-1}} = \sum_{r_1=1}^{R_1} \cdots \sum_{r_{N-1}=1}^{R_{N-1}} \left(\prod_{n=1}^{N-1} h_{i_n, r_n}^{(n;R_n)}\right) \mathbf{f}_{i_{r_1}^{(1)}, \ldots, i_{r_{N-1}}^{(N-1)}}, \tag{3}$$

and equivalently, the unfolding satisfies

$$\mathbf{X}_{(N)} = \left(\mathcal{X} \times_{n=1}^{N-1} \mathbf{H}^{(n;R_n)} \mathbf{W}^{(n;R_n)}\right)_{(N)} = \mathbf{F}_{\mathrm{core}} \bigotimes_{n=1}^{N-1} \mathbf{H}^{(N-n;R_{N-n})^\top}. \tag{4}$$

**Proof.** A detailed proof of this theorem is presented in Appendix 7.2.

**Intuition.** Fixing bases for $\mathcal{S}(\mathcal{X}^{(1)}), \ldots, \mathcal{S}(\mathcal{X}^{(N-1)})$ encodes the tensor's structure in their $\prod_{n=1}^{N-1} R_n$ intersection fibers, which act as a compact *core dictionary*, capturing the interactions between the mode $1, \ldots, N-1$ subspaces. The coefficient matrices $\mathbf{H}^{(n;R_n)}, n \in \{1, \ldots, N-1\}$ provide separable weights to reconstruct all fibers. Exact recovery is guaranteed when the chosen slices form true bases; otherwise, approximate bases yield reconstructions with errors tied to the residuals, for which we derive explicit upper bounds in Appendix 7.3.

**Remark 2.** Theorem 3.1 underpins multi-way feature selection. When modes $1, \cdots, N-1$ correspond to features and mode-$N$ indexes samples, each mode-$N$ fiber represents a feature's response across samples. Feature selection thus reduces to choosing representative bases along modes $1, \ldots, N-1$, whose intersection fibers form the most informative representatives of the full feature space.

## 4 TENSOR-BASED FEATURE SELECTION

In this section, we formalize the task of feature selection in tensor data. The model developed in this section is presented under the assumption that the input is a non-negative 3-mode tensor. This assumption is well aligned with many practical multi-way datasets such as images, videos, and medical scans, where entries naturally take non-negative values (Bi et al., 2025). Nonetheless, the framework can be readily extended to general tensor data, and we provide a discussion of this extension in Appendix 7.7. Let $\mathcal{X} \in \mathbb{R}_+^{I_1 \times I_2 \times I_3}$ be a non-negative 3-mode data tensor with $I_3$ samples, each described by $I_1 \times I_2$ multi-way features. The problem is to select a subset of mode-3 fibers that best preserve the structure of the full tensor.

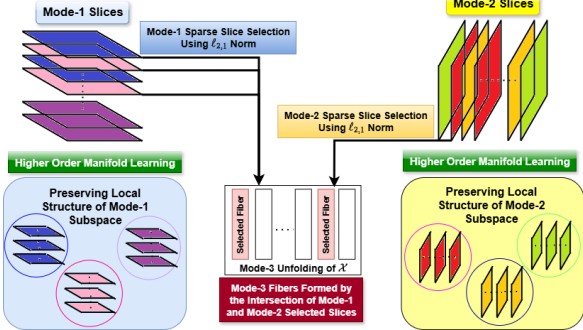

Figure 1: Schematic illustration of multi-linear subspace learning feature selection (MSLFS). Mode-1 and mode-2 slices of the input tensor are processed via $\ell_{2,1}$-norm based sparse selection, where the joint row sparsity regularization ensures that only a limited number of slice combinations are retained as informative representatives. The first term of the objective function ensures reconstruction fidelity by using the intersection of the selected slices to form representative mode-3 fibers. The second term enforces local manifold preservation within each mode, thereby maintaining the geometric structure of the data subspaces.

**Feature Selection via Core Theorem.**
According to Theorem 3.1, this can be
achieved by choosing $m_1 \leq I_1$ slices
along mode-1 and $m_2 \leq I_2$ slices along
mode-2, which approximate $\mathcal{S}(\mathbf{X}^{(1)})$ and $\mathcal{S}(\mathbf{X}^{(2)})$, respectively. The intersection of these selected
slices yields a compact yet expressive set of representative mode-3 fibers that best span the feature
subspace. Concretely, the feature selection problem can be formulated as follows:

$$\min_{\mathbf{H}^{(1;m_1)}, \mathbf{H}^{(2;m_2)}, \mathbf{W}^{(1;m_1)}, \mathbf{W}^{(2;m_2)} \geq 0} \|\mathcal{X} - \mathcal{X} \times_1 \mathbf{H}^{(1;m_1)} \mathbf{W}^{(1;m_1)} \times_2 \mathbf{H}^{(2;m_2)} \mathbf{W}^{(2;m_2)}\|_F^2$$

$$\text{s.t.} \quad \mathbf{W}^{(1;m_1)} \in \text{Ind}^{m_1 \times I_1}, \mathbf{W}^{(2;m_2)} \in \text{Ind}^{m_2 \times I_2}. \quad (5)$$

Here, $\mathbf{W}^{(1;m_1)}$ and $\mathbf{W}^{(2;m_2)}$ are indicator matrices marking the selected slices, and $\mathbf{H}^{(1;m_1)} \in \mathbb{R}^{I_1 \times m_1}$ and $\mathbf{H}^{(2;m_2)} \in \mathbb{R}^{I_2 \times m_2}$ are the corresponding coefficient matrices.

**Relaxation via Orthogonality.** Since the minimization problem (5) is NP-hard, directly using indicator matrices is impractical. We relax this by enforcing orthogonality on $\mathbf{W}^{(1;m_1)}$ and $\mathbf{W}^{(2;m_2)}$, equivalently on their Kronecker product. Combined with non-negativity, this ensures each column remains one-hot, preserving selection while keeping the optimization tractable.

**Row-Sparsity Regularization.** Given the sparsity of $\mathbf{W}^{(1;m_1)}$ and $\mathbf{W}^{(2;m_2)}$, their Kronecker product, which acts as the indicator matrix for the $m_1 \times m_2$ intersection mode-3 fibers, inherits this property. To emphasize only the most informative slice combinations, we impose joint row-sparsity on $(\mathbf{W}^{(2;m_2)} \otimes \mathbf{W}^{(1;m_1)})^\top$, ensuring that only a few mode-3 fibers dominate the reconstruction and redundancy is reduced. To formalize this idea, we employ the $\ell_{2,1}$ norm. For $(\mathbf{W}^{(2;m_2)} \otimes \mathbf{W}^{(1;m_1)})^\top$ this becomes:

$$\|(\mathbf{W}^{(2;m_2)} \otimes \mathbf{W}^{(1;m_1)})^\top\|_{2,1} = \text{Tr}\big((\mathbf{W}^{(2;m_2)} \otimes \mathbf{W}^{(1;m_1)}) \mathbf{U} (\mathbf{W}^{(2;m_2)} \otimes \mathbf{W}^{(1;m_1)})^\top\big), \quad (6)$$

where $\mathbf{U} \in \mathbb{R}^{I_1 I_2 \times I_1 I_2}$ is diagonal with entries equal to the reciprocals of the $\ell_2$ norms of the columns of $\mathbf{W}^{(2;m_2)} \otimes \mathbf{W}^{(1;m_1)}$.

**Mode-Wise Factorization of the Penalty.** Because each column of $\mathbf{W}^{(2;m_2)} \otimes \mathbf{W}^{(1;m_1)}$ is a Kronecker product of a column of $\mathbf{W}^{(2;m_2)}$ and one of $\mathbf{W}^{(1;m_1)}$, the matrix $\mathbf{U}$ decomposes as $\mathbf{U}^{(2)} \otimes \mathbf{U}^{(1)}$, where $\mathbf{U}^{(1)} \in \mathbb{R}^{I_1 \times I_1}$ and $\mathbf{U}^{(2)} \in \mathbb{R}^{I_2 \times I_2}$ are diagonal matrices whose entries depend only on the columns of $\mathbf{W}^{(1;m_1)}$ and $\mathbf{W}^{(2;m_2)}$, respectively. Substituting this gives:

$$\|(\mathbf{W}^{(2;m_2)} \otimes \mathbf{W}^{(1;m_1)})^\top\|_{2,1} = \text{Tr}\big((\mathbf{W}^{(2;m_2)} \otimes \mathbf{W}^{(1;m_1)})(\mathbf{U}^{(2)} \otimes \mathbf{U}^{(1)})(\mathbf{W}^{(2;m_2)} \otimes \mathbf{W}^{(1;m_1)})^\top\big). \quad (7)$$

Using standard Kronecker product identities, this expression simplifies to

$$\text{Tr}((\mathbf{W}^{(2;m_2)} \mathbf{U}^{(2)} \mathbf{W}^{(2;m_2)^\top}) \otimes (\mathbf{W}^{(1;m_1)} \mathbf{U}^{(1)} \mathbf{W}^{(1;m_1)^\top})),$$

and since the trace of a Kronecker product factorizes into the product of traces, we finally obtain

$$\|(\mathbf{W}^{(2;m_2)} \otimes \mathbf{W}^{(1;m_1)})^\top\|_{2,1} = \|\mathbf{W}^{(2;m_2)^\top}\|_{2,1} \|\mathbf{W}^{(1;m_1)^\top}\|_{2,1}$$

$$= \text{Tr}(\mathbf{W}^{(2;m_2)} \mathbf{U}^{(2)} \mathbf{W}^{(2;m_2)^\top}) \text{Tr}(\mathbf{W}^{(1;m_1)} \mathbf{U}^{(1)} \mathbf{W}^{(1;m_1)^\top}). \quad (8)$$

**Interpretation.** The $\ell_{2,1}$ penalty factorizes across modes, with each trace term measuring the representational quality of slices in its subspace while penalizing redundancy. This separation reduces computation and enables mode-wise control, ensuring balanced selection that retains only the most informative fibers.

### 4.1 GRAPH REGULARIZATION FOR HIGHER-ORDER MANIFOLD LEARNING

In multi-way feature selection, it is crucial to preserve both the global span and the intrinsic geometry of the data. Graph regularization enforces local neighborhood consistency, ensuring proximity in the original space is maintained in the learned representation. Extending this to tensors requires jointly modeling local structures across all modes.

**Fiber Representation.** Let $\mathbf{H}^{(1;m_1)} \in \mathbb{R}^{I_1 \times m_1}$ and $\mathbf{H}^{(2;m_2)} \in \mathbb{R}^{I_2 \times m_2}$ denote coefficient matrices for the selected slices along modes 1 and 2. By Theorem 3.1, each mode-3 fiber $\mathbf{f}_{i_1,i_2}$ can be approximated in terms of the core fibers as: $\mathbf{f}_{i_1,i_2} = \mathbf{F}_{\text{core}} \left( (\mathbf{H}^{(2;m_2)} \otimes \mathbf{H}^{(1;m_1)})^\top \right)_{:,\overline{i_1 i_2}}$, where the coefficient vector $\left( (\mathbf{H}^{(2;m_2)} \otimes \mathbf{H}^{(1;m_1)})^\top \right)_{:,\overline{i_1 i_2}}$ encodes how the fiber is reconstructed from the shared subspace. Intuitively, if two fibers $\mathbf{f}_{i_1,i_2}$ and $\mathbf{f}_{j_1,j_2}$ are similar in the original space, their coefficient vectors should also be close, reflecting their functional similarity in reconstruction.

**Graph Regularization.** To enforce this locality, we minimize the squared distance between coefficient vectors, weighted by their similarity:

$$\frac{1}{2} \sum_{i_1,i_2} \sum_{j_1,j_2} \left\| \left( (\mathbf{H}^{(2;m_2)} \otimes \mathbf{H}^{(1;m_1)})^\top \right)_{:,\overline{i_1 i_2}} - \left( (\mathbf{H}^{(2;m_2)} \otimes \mathbf{H}^{(1;m_1)})^\top \right)_{:,\overline{j_1 j_2}} \right\|_2^2 b_{\overline{i_1 i_2}, \overline{j_1 j_2}}, \quad (9)$$

where $b_{\overline{i_1 i_2}, \overline{j_1 j_2}}$ encodes the similarity between $\mathbf{f}_{i_1,i_2}$ and $\mathbf{f}_{j_1,j_2}$. This term can be rewritten compactly in matrix form as: $\text{Tr}\left[ (\mathbf{H}^{(2;m_2)} \otimes \mathbf{H}^{(1;m_1)})^\top \mathbf{L} (\mathbf{H}^{(2;m_2)} \otimes \mathbf{H}^{(1;m_1)}) \right]$, where $\mathbf{L} \in \mathbb{R}^{I_1 I_2 \times I_1 I_2}$ is the Laplacian of the feature similarity graph.

**Mode-Wise Decomposition.** To ease computation, we exploit the fact that similarities between fibers factorize across modes. This induces a Kronecker structure in the joint Laplacian, expressed as $\mathbf{L} = \mathbf{L}^{(2)} \otimes \mathbf{L}^{(1)}$. For each mode $n \in \{1, 2\}$, the Laplacian $\mathbf{L}^{(n)} = \mathbf{A}^{(n)} - \mathbf{B}^{(n)}$ is constructed from the degree matrix $\mathbf{A}^{(n)}$ and similarity matrix $\mathbf{B}^{(n)}$, with $\mathbf{L}^{(n)}, \mathbf{A}^{(n)}, \mathbf{B}^{(n)} \in \mathbb{R}^{I_n \times I_n}$. Substituting this decomposition yields:

$$\text{Tr}\left[ (\mathbf{H}^{(2;m_2)} \otimes \mathbf{H}^{(1;m_1)})^\top (\mathbf{L}^{(2)} \otimes \mathbf{L}^{(1)})(\mathbf{H}^{(2;m_2)} \otimes \mathbf{H}^{(1;m_1)}) \right] =$$
$$\text{Tr}(\mathbf{H}^{(2;m_2)^\top} \mathbf{L}^{(2)} \mathbf{H}^{(2;m_2)}) \, \text{Tr}(\mathbf{H}^{(1;m_1)^\top} \mathbf{L}^{(1)} \mathbf{H}^{(1;m_1)}). \quad (10)$$

**Interpretation.** The factorization shows that preserving local geometry among fibers indexed by $(i_1, i_2)$ decomposes into two preservation tasks, one per mode. Each trace term enforces neighborhood smoothness along its mode, while the Kronecker structure captures their joint effect. This regularization encourages nearby slices in the tensor to share similar coefficients in the reduced space, aligning feature selection with the data manifold. The mode-wise decomposition also lowers computational cost and clarifies each mode's contribution to locality preservation.

**Similarity Construction.** The similarity matrices $\mathbf{B}^{(1)} = [b_{i_1,i_2}^{(1)}] \in \mathbb{R}^{I_1 \times I_1}$ and $\mathbf{B}^{(2)} = [b_{i_1,i_2}^{(2)}] \in \mathbb{R}^{I_2 \times I_2}$ are built via a heat kernel. For example, the similarity between two mode-$n$ slices $\mathbf{X}_{i_1}^{(n)}$ and $\mathbf{X}_{i_2}^{(n)}$, where $n \in \{1, 2\}$, is defined as: $b_{i_1,i_2}^{(n)} = \exp\left( -\|\mathbf{X}_{i_1}^{(n)} - \mathbf{X}_{i_2}^{(n)}\|_F^2 / \sigma^2 \right)$ if $\mathbf{X}_{i_1}^{(n)} \in \mathcal{N}_k(\mathbf{X}_{i_2}^{(n)})$ or vice versa; otherwise $b_{i_1,i_2}^{(n)} = 0$, where $\sigma > 0$ is the kernel width and $\mathcal{N}_k(\cdot)$ denotes the set of $k$ nearest neighbors.

**Overall Objective Function.** Bringing together the reconstruction fidelity, sparsity control, and manifold preservation, the MSLFS framework can be formulated as

$$\min_{\mathbf{H}^{(n;m_n)}, \mathbf{W}^{(n;m_n)} \geq 0, \forall n \in \{1,2\}} \frac{1}{2} \| \mathcal{X} - \mathcal{X} \times_1 \mathbf{H}^{(1;m_1)} \mathbf{W}^{(1;m_1)} \times_2 \mathbf{H}^{(2;m_2)} \mathbf{W}^{(2;m_2)} \|_F^2$$
$$+ \frac{\alpha}{2} \text{Tr}(\mathbf{H}^{(2;m_2)^\top} \mathbf{L}^{(2)} \mathbf{H}^{(2;m_2)}) \, \text{Tr}(\mathbf{H}^{(1;m_1)^\top} \mathbf{L}^{(1)} \mathbf{H}^{(1;m_1)})$$
$$+ \frac{\beta}{2} \text{Tr}(\mathbf{W}^{(2;m_2)} \mathbf{U}^{(2)} \mathbf{W}^{(2;m_2)^\top}) \, \text{Tr}(\mathbf{W}^{(1;m_1)} \mathbf{U}^{(1)} \mathbf{W}^{(1;m_1)^\top})$$
$$\text{s.t.} \quad \mathbf{W}^{(2;m_2)} \mathbf{W}^{(2;m_2)^\top} \otimes \mathbf{W}^{(1;m_1)} \mathbf{W}^{(1;m_1)^\top} = \mathbf{I}_{m_1 m_2}. \quad (11)$$

Details of the optimization procedure, convergence analysis, and computational complexity are provided in Appendices 7.4, 7.5, and 7.6, respectively. In brief, Algorithm 1 outlines the optimization steps for solving the minimization problem (11).

---

**Algorithm 1** MSLFS Algorithm

---

**Input**: Data tensor $\mathcal{X} \in \mathbb{R}^{I_1 \times I_2 \times I_3}$; numbers of selected slices $m_1, m_2$; parameters $\alpha, \beta, \gamma$; max_iter.
**Output**: Compute $\ell_2$-norm of columns in $\mathbf{W}^{(1;m_1)}$, $\mathbf{W}^{(2;m_2)}$, sort descending. Select top $m_1$ columns of $\mathbf{W}^{(1;m_1)}$, top $m_2$ of $\mathbf{W}^{(2;m_2)}$ for mode-1, mode-2 slices. Output $m_1 \times m_2$ features at their intersection.
1: Initialize $\mathbf{W}^{(1;m_1)} \in \mathbb{R}^{m_1 \times I_1}$, $\mathbf{W}^{(2;m_2)} \in \mathbb{R}^{m_2 \times I_2}$, $\mathbf{H}^{(1;m_1)} \in \mathbb{R}^{I_1 \times m_1}$, $\mathbf{H}^{(2;m_2)} \in \mathbb{R}^{I_2 \times m_2}$ randomly; build similarity matrices $\mathbf{B}^{(1)}$, $\mathbf{B}^{(2)}$.
2: **for** $t = 0$ to max_iter **do**
3:    Update $\mathbf{W}^{(1;m_1)}$ via (17), $\mathbf{H}^{(1;m_1)}$ via (19), $\mathbf{W}^{(2;m_2)}$ via (23), $\mathbf{H}^{(2;m_2)}$ via (25).
4: **end for**

---

## 5 EXPERIMENTS

In this section, we demonstrate the effectiveness of MSLFS through extensive experiments, comparing it with top-performing feature selection models on real-world benchmark datasets.

**Datasets and Compared Methods.** To evaluate the effectiveness of MSLFS, we conduct experiments on several benchmark datasets, including **COIL20** (Nene et al., 1996), **Kinetic Fluorescence** (Nikolajsen et al., 2023), **ORL** (Cai et al., 2010), **UMIST** (Graham & Allinson, 1998), **Pixraw10P** (Li et al., 2017), **Orlraws10P** (Li et al., 2017), **FashionMNIST** (Xiao et al., 2017), **BreastMNIST** (Yang et al., 2021), **PneumoniaMNIST** (Yang et al., 2023), **OrganCMNIST** (Yang et al., 2023), **OrganSMNIST** (Yang et al., 2021), and **COVID-19 Systems Serology** (Tan et al., 2023). For comparison, we select 13 top-tier models: **LS** (He et al., 2005b), **UDFS** (Yang et al., 2011), **SAE** (Guo et al., 2017), **ILFS** (Roffo et al., 2017), **GRLTR** (Su et al., 2018), **CAE** (Balın et al., 2019), **FSPCA** (Tian et al., 2020), **CPUFS** (Chen et al., 2023), **SPCAFS** (Li et al., 2023), **NNSE** (You et al., 2023a), **GRSSLFS** (Tiwari et al., 2024), **SDAE** (Hassanieh & Chehade, 2024a), and **SPDFS** (Dong et al., 2025).

**Experimental Settings.** To ensure fair evaluation, all methods are tuned under comparable settings. For graph-based approaches, the $k$-neighborhood is selected from $\{2, 5, 10, 15\}$. We fix $\gamma = 10^8$ to enforce orthogonality and set the kernel width $\sigma = 10^3$. Regularization parameters are searched over $\{10^{-4}, 10^{-3}, \dots, 10^4\}$, and the number of selected features is varied across $\{50, 100, 150, 200, 250, 300\}$. Clustering is performed with the true number of clusters, and the maximum iterations of iterative methods are tuned within $\{5, 10, 30\}$, where 5 or 10 iterations offer a good trade-off between efficiency and convergence. $k$-means is applied to the selected features and repeated 10 times with random initializations; average results are reported. Performance is assessed by ACC and NMI (Solorio-Fernández et al., 2020), where higher values indicate better results.

**Clustering Results.** Table 2 presents ACC and NMI results across eight benchmarks against 10 leading baselines. MSLFS consistently achieves top performance, with large improvements on COIL20, ORL, and Orlraws10P, and robust results on challenging datasets such as FashionMNIST and BreastMNIST. These gains come from its slice-based subspace modeling, which leverages cross-mode structure, and graph-regularized selection, which maintains local geometry, producing compact and discriminative features that drive clustering accuracy.

Table 2: Clustering results of the MSLFS vs. 13 cutting-edge models on benchmark datasets (I).

| Model | COIL20 | | ORL | | UMIST | | Pixraw10P | | Orlraws10P | | FashionMNIST | | BreastMNIST | | OrganSMNIST | |
|---|---|---|---|---|---|---|---|---|---|---|---|---|---|---|---|---|
| | ACC | NMI | ACC | NMI | ACC | NMI | ACC | NMI | ACC | NMI | ACC | NMI | ACC | NMI | ACC | NMI |
| LS (NeurIPS 2005) | 54.34 | 72.11 | 48.11 | 71.44 | 41.05 | 59.31 | 67.37 | 83.52 | 69.24 | 78.76 | 50.49 | 51.17 | 62.22 | 4.78 | 33.17 | 37.69 |
| UDFS (IJCAI 2011) | 55.47 | 71.19 | 47.95 | 71.50 | 36.52 | 53.03 | 70.54 | 79.94 | 57.64 | 67.57 | 52.46 | 51.22 | 62.67 | 5.55 | 33.52 | 37.34 |
| SAE (IJCNN 2017) | 62.11 | 74.22 | 52.76 | 72.11 | 44.12 | 57.08 | 88.56 | 89.09 | 77.64 | 84.47 | 53.23 | 50.62 | 61.67 | 5.78 | 31.24 | 34.34 |
| ILFS (CVPR 2017) | 61.45 | 73.56 | 56.68 | 75.92 | 45.52 | 58.74 | 73.29 | 83.74 | 74.52 | 82.26 | 63.57 | 60.31 | 63.57 | 7.43 | 28.86 | 34.58 |
| GRLTR (JVCIR 2018) | 68.78 | 77.84 | 54.32 | 75.00 | 49.68 | 63.21 | 92.44 | 93.67 | 82.90 | 87.51 | 54.92 | 51.01 | 59.19 | 5.00 | 33.38 | 32.16 |
| CAE (ICML 2019) | 59.93 | 72.17 | 56.25 | 74.93 | 54.34 | 69.22 | 86.27 | 91.75 | 74.45 | 81.23 | **67.57** | 64.26 | 74.88 | 9.36 | 39.81 | 41.96 |
| FSPCA (NeurIPS 2020) | 67.14 | 79.43 | 57.07 | 73.97 | 52.38 | 65.54 | 85.66 | 92.16 | 80.41 | 87.74 | 63.26 | 61.68 | 71.42 | 8.55 | 38.15 | 40.81 |
| CPUFS (TPAMI 2022) | 64.72 | 76.21 | 57.38 | 75.39 | 49.46 | 63.37 | 77.27 | 89.40 | 76.81 | 85.36 | 60.53 | 58.52 | 67.87 | 8.26 | 37.24 | 39.57 |
| SPCAFS (TPAMI 2023) | 63.15 | 74.74 | 52.21 | 71.76 | 44.23 | 58.21 | 82.16 | 88.91 | 73.36 | 80.44 | 54.36 | 51.53 | 60.46 | 5.42 | 34.01 | 33.26 |
| NNSE (PATCOG 2023) | 69.22 | 79.34 | 57.21 | 77.23 | 55.63 | 66.11 | 86.63 | 92.31 | 80.37 | 85.93 | 57.11 | 64.63 | 64.55 | 11.34 | 38.01 | 39.26 |
| GRSSLFS (TMLR 2024) | 67.47 | 78.76 | 53.95 | 74.58 | **58.06** | 68.06 | 89.30 | 92.17 | 79.10 | 86.04 | 56.65 | 62.43 | 53.85 | 10.00 | 32.74 | 30.94 |
| SDAE (AAAI 2024) | 70.21 | 80.86 | 60.45 | 74.31 | 56.76 | 68.32 | 90.36 | 92.22 | 85.45 | 89.67 | 65.32 | 65.96 | 72.94 | 10.98 | 39.72 | 41.88 |
| SPDFS (TPAMI 2025) | 67.66 | 78.96 | 53.64 | 73.01 | 48.37 | 61.15 | 78.36 | 89.13 | 75.45 | 82.21 | 56.76 | 52.96 | 61.12 | 7.66 | 33.41 | 34.25 |
| MSLFS (Ours) | **73.15** | **84.67** | **64.43** | **79.61** | 56.79 | **70.17** | **93.16** | **94.28** | **88.33** | **91.42** | 66.42 | **66.74** | **76.93** | **12.85** | **44.25** | **44.87** |
| Improvement | +2.94 | +3.81 | +3.98 | +4.38 | – | +0.95 | +0.72 | +0.61 | +2.88 | +1.75 | – | +0.78 | +2.05 | +1.51 | +4.44 | +2.91 |

Table 3: Clustering results of the MSLFS vs. 13 cutting-edge models on benchmark datasets (II).

| Model | Kinetic Fluorescence | | PneumoniaMNIST | | OrganCMNIST | | COVID-19 Systems Serology | |
|---|---|---|---|---|---|---|---|---|
| | ACC | NMI | ACC | NMI | ACC | NMI | ACC | NMI |
| LS (NeurIPS 2005) | 69.86 | 77.66 | 54.33 | 63.17 | 48.02 | 51.79 | 65.55 | 81.12 |
| UDFS (IJCAI 2011) | 70.99 | 76.74 | 54.17 | 63.23 | 43.49 | 45.51 | 68.72 | 77.54 |
| SAE (IJCNN 2017) | 77.63 | 79.77 | 58.98 | 63.84 | 51.09 | 49.56 | 86.74 | 86.69 |
| ILFS (CVPR 2017) | 76.97 | 79.11 | 62.90 | 67.65 | 52.49 | 51.22 | 71.47 | 81.34 |
| GRLTR (JVCIR 2018) | 84.30 | 83.39 | 60.54 | 66.73 | 58.65 | 57.69 | 90.62 | 91.27 |
| CAE (ICML 2019) | 75.45 | 77.72 | 62.47 | 66.66 | 61.31 | 61.70 | 84.45 | 89.35 |
| FSPCA (NeurIPS 2020) | 82.66 | 84.98 | 63.29 | 65.70 | 59.35 | 58.02 | 83.84 | 89.76 |
| CPUFS (TPAMI 2022) | 80.24 | 81.76 | 63.60 | 67.12 | 56.43 | 55.85 | 75.45 | 87.00 |
| SPCAFS (TPAMI 2023) | 78.67 | 80.29 | 58.43 | 63.49 | 51.20 | 50.69 | 80.34 | 86.51 |
| NNSE (PATCOG 2023) | 83.74 | 84.89 | 63.43 | 68.96 | 62.60 | 58.59 | 84.81 | 89.91 |
| GRSSLFS (TMLR 2024) | 82.99 | 84.31 | 60.17 | 66.31 | 62.03 | 60.54 | 87.48 | 89.77 |
| SDAE (AAAI 2024) | 85.73 | 86.41 | 68.67 | 68.04 | **75.69** | **74.80** | 88.54 | 89.82 |
| SPDFS (TPAMI 2025) | 83.18 | 84.51 | 59.86 | 64.74 | 55.34 | 53.63 | 76.54 | 86.73 |
| **MSLFS (Ours)** | **88.67** | **90.22** | **71.65** | **72.34** | 73.76 | 72.65 | **95.34** | **95.88** |
| **Improvement** | +2.94 | +3.81 | +2.98 | +3.38 | - | - | +4.72 | +4.61 |

**Ablation Study.** The MSLFS objective includes two regularizations: locality preservation ($\alpha$) to capture local geometry and sparsity ($\beta$) to enhance discriminability. An ablation study on six datasets (Table 4) shows that full model consistently outperforms reduced variants. Removing either term lowers performance, with the sharpest drop when both are omitted, confirming their complementary importance for robust clustering.

Table 4: Ablation study results on eight datasets.

| Case | COIL20 | | Pixraw10P | | ORL | | BreastMNIST | | UMIST | | OrganCMNIST | | Kinetic Fluorescence | | COVID-19 Systems Serology | |
|---|---|---|---|---|---|---|---|---|---|---|---|---|---|---|---|---|
| | ACC | NMI | ACC | NMI | ACC | NMI | ACC | NMI | ACC | NMI | ACC | NMI | ACC | NMI | ACC | NMI |
| $\alpha, \beta \neq 0$ | 72.88 | 83.44 | 93.66 | 94.11 | 64.13 | 79.45 | 76.13 | 12.69 | 56.33 | 69.79 | 73.69 | 72.49 | 88.34 | 89.98 | 95.56 | 96.02 |
| $\alpha = 0$ | 66.87 | 77.22 | 89.80 | 91.37 | 61.12 | 77.54 | 63.14 | 8.44 | 49.42 | 63.38 | 72.21 | 72.38 | 85.34 | 85.76 | 91.76 | 94.56 |
| $\beta = 0$ | 68.13 | 78.97 | 90.45 | 92.32 | 58.98 | 75.66 | 68.73 | 10.89 | 49.01 | 61.88 | 71.34 | 71.77 | 86.22 | 87.03 | 93.33 | 95.43 |
| $\alpha, \beta = 0$ | 64.86 | 74.78 | 85.20 | 88.03 | 56.90 | 74.05 | 61.13 | 7.89 | 46.56 | 57.22 | 69.78 | 70.87 | 83.23 | 82.79 | 90.09 | 91.13 |

**Convergence Curves.** This section analyzes the convergence of MSLFS on four benchmark datasets. Figure 2 shows objective values versus iterations (up to 50). In all cases, the loss drops quickly at first and then stabilizes, demonstrating fast and robust convergence across diverse datasets.

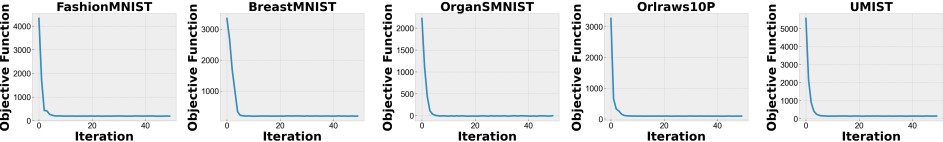

Figure 2: Convergence curves of the MSLFS on the image datasets.

**Computational Complexity.** Table 5 shows that while many methods incur cubic costs in tensor dimensions, MSLFS reduces complexity to linear dependence on $I_1 I_2 I_3$ with only minor contributions from slice counts. Its mode-wise design distributes selection across modes and avoids costly global operations, yielding clear efficiency gains over prior approaches.

**Data Visualization using t-SNE.** Figure 3 presents t-SNE visualizations on UMIST. The raw data shows scattered and overlapping clusters, while MSLFS with varying feature counts produces progressively clearer, more compact, and better-separated groups. This demonstrates the ability of the MSLFS to extract discriminative features that enhance clustering quality.

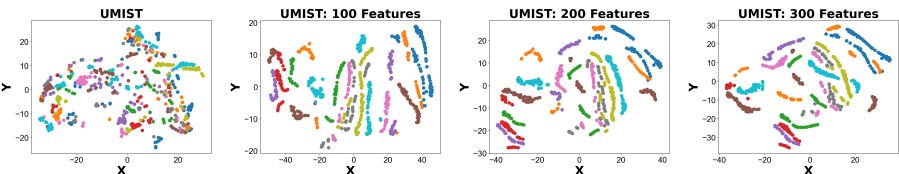

Figure 3: t-SNE plots of UMIST before and after feature reduction with MSLFS.

Table 5: Computational complexity of different models for each iteration. Here $t$ and $c$ denote the dimension of the reduced space and cluster number, respectively.

| Model | Computational Complexity |
|---|---|
| LS | $\mathcal{O}\big(I_1 I_2 I_3^2 + I_1 I_2 \log_2 I_1 I_2\big)$ |
| UDFS | $\mathcal{O}\big(I_1^3 I_3^3 + I_3^2 c\big)$ |
| FSPCA | $\mathcal{O}\big(\max\{I_1 I_2 m_1 m_2 t, m_1^3 m_2^3\} + I_1 I_2 m_1 m_2 t\big)$ |
| CPUFS | $\mathcal{O}\big((I_1 I_3 + I_2 I_3)c^2 + (I_1 I_2 I_3 + I_3^2)c\big)$ |
| SPCAFS | $\mathcal{O}\big(I_1^2 I_2^2 (I_3 + I_1 I_2)\big)$ |
| GRSSLFS | $\mathcal{O}\big(I_1^2 I_2^2 I_3^2\big)$ |
| GRLTR | $\mathcal{O}\big(I_1 I_2 I_3 \log_2 I_3 + I_1 I_2^2 I_3 + I_3^3\big)$ |
| SPDFS | $\mathcal{O}\big(\max\{I_1 I_2 I_3 t, I_1^2 I_2^2 t\} + I_1 I_2 I_3 t c^2 + \max\{I_1 I_2 I_3 t, I_1 I_2 \log_2 I_1 I_2, m_1^3 m_2^3\}\big)$ |
| MSLFS | $\mathcal{O}\big(I_1 I_2 I_3 (\max\{m_1, I_2\} + \max\{m_2, I_1\})\big)$ |

**Selected Features Visualization.** Figure 4 depicts feature selection on ORL and Pixraw10P with 100, 200, and 300 features. Fewer features capture broad structure, while more reveal finer details. Across datasets, the model consistently highlights informative regions, expressing its efficacy for image-based feature selection.

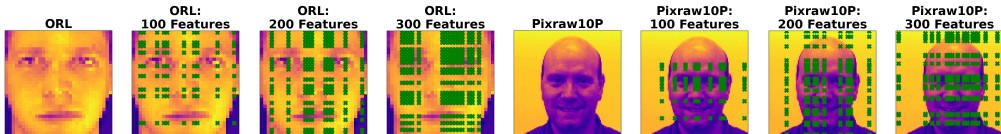

Figure 4: Image visualizations on ORL and Pixraw10P with 100, 200, and 300 selected features.

## 6 CONCLUSION

The proposed MSLFS introduces a novel approach to tensor-based feature selection by distributing the selection process across modes rather than treating the feature space as a rigid whole. Its key innovation, the multi-linear subspace distance, provides a principled criterion for preserving global structure while enabling efficient and interpretable feature selection. Complemented by joint sparsity and higher-order graph regularization, MSLFS captures both cross-mode dependencies and local manifold geometry, setting it apart from existing tensor-based methods. This framework opens new directions for multi-way learning, with future work aimed at extending MSLFS to broader tasks such as its integration with deep tensor architectures for large-scale representation learning. Comprehensive theoretical discussions and supplementary experiments can be found in Appendices 7 and 8*.

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

# 7 ADDITIONAL THEORETICAL RESULTS

## 7.1 NOTATIONS AND PRELIMINARIES

**Notations.** Throughout this paper, vectors are represented by bold lowercase letters (e.g., $\mathbf{v}$), matrices by bold uppercase letters (e.g., $\mathbf{A}$), and tensors by bold calligraphy letters (e.g., $\mathcal{X}$). The identity matrix of size $m$ is denoted by $\mathbf{I}_m$, and $\mathbf{e}_j^{(m)}$ denotes the $j$th column of $\mathbf{I}_m$. For a matrix $\mathbf{A} = (a_{ij}) \in \mathbb{R}^{m \times n}$, the $i$-th row and the $j$-th column are denoted by $\mathbf{A}_{i,:}$ and $\mathbf{A}_{:,j}$, respectively. The Frobenius norm of $\mathbf{A}$ is defined as $\|\mathbf{A}\|_F = \sqrt{\sum_{i=1}^m \sum_{j=1}^n a_{ij}^2}$, while the $\ell_{2,1}$-norm is given by $\|\mathbf{A}\|_{2,1} = \sum_{i=1}^m \|\mathbf{A}_{i,:}\|_2$. For a square matrix $\mathbf{A}$, $\mathrm{Tr}(\mathbf{A})$ denotes its trace. The dot product between two vectors $\mathbf{u}, \mathbf{v} \in \mathbb{R}^n$ is defined as $\langle \mathbf{u}, \mathbf{v} \rangle = \sum_{i=1}^n u_i v_i$, and the Frobenius inner product between two matrices $\mathbf{A}, \mathbf{B} \in \mathbb{R}^{m \times n}$ is defined as $\langle \mathbf{A}, \mathbf{B} \rangle_F = \sum_{i=1}^m \sum_{j=1}^n a_{ij} b_{ij}$. The Hadamard product of $\mathbf{A}, \mathbf{B} \in \mathbb{R}^{m \times n}$ is expressed as $\mathbf{A} \odot \mathbf{B} = (a_{ij} b_{ij})_{i=1, j=1}^{m,n}$. For $\mathbf{A} \in \mathbb{K}^{I \times J}$ and $\mathbf{B} \in \mathbb{K}^{M \times N}$, the Kronecker product is $\mathbf{A} \otimes \mathbf{B} \in \mathbb{K}^{(IM) \times (JN)}$; defining $\overline{im} = (i-1)M + m$ and $\overline{jn} = (j-1)N + n$, its entries are given by $(\mathbf{A} \otimes \mathbf{B})_{\overline{im}, \overline{jn}} = a_{ij} b_{mn}$ for $1 \le i \le I$, $1 \le j \le J$, $1 \le m \le M$, and $1 \le n \le N$. A third-order tensor is denoted by $\mathcal{X} = (x_{i_1, i_2, i_3})_{i_1=1,\ldots,I_1; i_2=1,\ldots,I_2; i_3=1,\ldots,I_3} \in \mathbb{R}^{I_1 \times I_2 \times I_3}$, where each entry $x_{i_1, i_2, i_3} \in \mathbb{R}$; the $j$-th frontal slice ($j = 1, \ldots, I_3$), denoted by $\mathbf{X}_j^{(3)}$, is obtained by fixing the third index and belongs to $\mathbb{R}^{I_1 \times I_2}$. Furthermore, the mode-3 unfolding of $\mathcal{X}$, denoted by $\mathbf{X}_{(3)}$, rearranges the entries of $\mathcal{X}$ into a matrix of size $I_3 \times I_1 I_2$ by mapping the mode-3 fibers to the columns of $\mathbf{X}_{(3)}$.

**Preliminaries.** Let $\mathcal{X} \in \mathbb{R}^{I_1 \times I_2 \times \cdots \times I_N}$ be an $N$-mode tensor, $\mathbf{A} \in \mathbb{R}^{J \times I_n}$ a matrix, and $\mathbf{v} \in \mathbb{R}^{I_n}$ a vector ($n = 1, \ldots, N$). The $n$-mode tensor-matrix product $\mathcal{X} \times_n \mathbf{A} \in \mathbb{R}^{I_1 \times \cdots \times J \times \cdots \times I_N}$ and the $n$-mode tensor-vector product $\mathcal{X} \bar{\times}_n \mathbf{v} \in \mathbb{R}^{I_1 \times \cdots \times I_{n-1} \times I_{n+1} \times \cdots \times I_N}$ are defined elementwise as $(\mathcal{X} \times_n \mathbf{A})_{i_1 \cdots j \cdots i_N} = \sum_{i_n=1}^{I_n} x_{i_1 \cdots i_N} a_{j i_n}$, $(\mathcal{X} \bar{\times}_n \mathbf{v})_{i_1 \cdots i_{n-1} i_{n+1} \cdots i_N} = \sum_{i_n=1}^{I_n} x_{i_1 \cdots i_N} v_{i_n}$. For $\mathbf{y} \in \mathbb{R}^J$, we have $\mathcal{X} \times_n \mathbf{A} \bar{\times}_n \mathbf{y} = \mathcal{X} \bar{\times}_n (\mathbf{y}^\top \mathbf{A})$, and the $j$-th mode-$n$ slice of $\mathcal{X} \times_n \mathbf{A}$ is $\mathcal{X} \bar{\times}_n \mathbf{A}_{j,:}$, $j = 1, \ldots, J$. In particular, if $\mathbf{y} = \mathbf{e}_j^{(n)\top}$ is the $j$-th column of the identity $\mathbf{I}_{I_n}$, then $\mathcal{X} \times_n \mathbf{y}$ extracts the $j$-th mode-$n$ slice: $\mathcal{X} \times_n \mathbf{e}_j^{(n)\top} = \mathcal{X}_j^{(n)}$.

The mode-$n$ unfolding of $\mathcal{X}$, denoted by $\mathbf{X}_{(n)} \in \mathbb{R}^{I_n \times (I_1 \cdots I_{n-1} I_{n+1} \cdots I_N)}$, rearranges $\mathcal{X}$ into a matrix by aligning all mode-$n$ fibers as its columns. Tensor-matrix products admit the following unfolding formulations: $(\mathcal{X} \times_n \mathbf{A})_{(n)} = \mathbf{A}\, \mathbf{X}_{(n)}$, $(\mathcal{X} \times_m \mathbf{A})_{(n)} = \mathbf{X}_{(n)} \big( \mathbf{I}_{I_{m+1} \cdots I_N} \otimes \mathbf{A} \otimes \mathbf{I}_{I_1 \cdots I_{m-1}} \big)^\top$, $m \ne n$. More generally, for a sequence of tensor-matrix products, we have $\big( \mathcal{X} \times_1 \mathbf{A}_1 \times_2 \mathbf{A}_2 \cdots \times_N \mathbf{A}_N \big)_{(n)} = \mathbf{A}_n\, \mathbf{X}_{(n)} \big( \mathbf{A}_N \otimes \cdots \otimes \mathbf{A}_{n+1} \otimes \mathbf{A}_{n-1} \otimes \cdots \otimes \mathbf{A}_1 \big)^\top$.

## 7.2 PROOF OF THEOREM 3.1

*Proof.* **Step 1 (Core matrix equals intersection fibers).** Define the subtensor obtained by selecting the chosen basis slices in modes $1, \cdots, N-1$

$$\mathcal{Y} := \mathcal{X} \times_{n=1}^{N-1} \mathbf{W}^{(n;R_n)} \in \mathbb{R}^{R_1 \times \cdots \times R_{N-1} \times I_N}.$$

Since $\mathbf{W}^{(n;R_n)}$ is the indicator selector for the index set $T_n$, the $(r_1, r_2, \cdots, r_{N-1}, :)$-entry of $\mathcal{Y}$ is precisely the intersection fiber $\mathbf{f}_{i_{r_1}^{(1)}, \cdots, i_{r_{N-1}}^{(N-1)}} = \mathcal{X}_{i_{r_1}^{(1)}, \cdots, i_{r_{N-1}}^{(N-1)},:}$. Hence, $\mathcal{Y}$ stacks exactly the $\prod_{n=1}^{N-1} R_n$ intersection fibers. Unfolding along mode-$N$ and using the standard product–unfolding identity yields

$$\mathbf{F}_{\text{core}} := \mathbf{Y}_{(N)} = \mathbf{X}_{(N)} \bigotimes_{n=1}^{N-1} \mathbf{W}^{(N-n;R_{N-n})^\top},$$

so $\mathbf{F}_{\text{core}}$ is exactly the matrix whose columns are the $\prod_{n=1}^{N-1} R_n$ intersection fibers. **Step 2 (Co-**

**efficient matrices along modes** $1, \cdots, N-1$**).** Because $\{\mathcal{X}^{(n)}_{i^{(n)}_{r_n}}\}^{R_n}_{r_n=1}$ is a basis of $\mathcal{S}(\mathcal{X}^{(n)})$, $n \in \{1, \cdots, N-1\}$, for every $i_n \in \{1, 2, \cdots, I_n\}$, there exist coefficients $\{h^{(n;R_n)}_{i_n,r_n}\}^{R_n}_{r_n=1}$ such that

$$\mathcal{X}^{(n)}_{i_n} = \sum_{r_n=1}^{R_n} h^{(n;R_n)}_{i_n,r_n} \mathcal{X}^{(n)}_{i^{(n)}_{r_n}}.$$

Collect these into $\mathbf{H}^{(n;R_n)} \in \mathbb{R}^{I_n \times R_n}$. **Step 3 (Fiber-level decomposition).** Fix an $(N-1)$-tuple $(i_1, \cdots, i_{N-1})$. Expanding along modes $1, \cdots, N-1$ gives

$$\mathbf{f}_{i_1,\cdots,i_{N-1}} = \mathcal{X}_{i_1,\cdots,i_{N-1},:} = \sum_{r_1=1}^{R_1} h^{(1;R_1)}_{i_1,r_1} \mathcal{X}_{i^{(1)}_{r_1},\cdots,i_{N-1},:} = \sum_{r_1=1}^{R_1} \cdots \sum_{r_{N-1}=1}^{R_{N-1}} \left(\prod_{n=1}^{N-1} h^{(n;R_n)}_{i_n,r_n}\right) \mathcal{X}_{i^{(1)}_{r_1},\cdots,i^{(N-1)}_{r_{N-1}},:},$$

i.e.,

$$\mathbf{f}_{i_1,\cdots,i_{N-1}} = \sum_{r_1=1}^{R_1} \cdots \sum_{r_{N-1}=1}^{R_{N-1}} \left(\prod_{n=1}^{N-1} h^{(n;R_n)}_{i_n,r_n}\right) \mathbf{f}_{i^{(1)}_{r_1},\cdots,i^{(N-1)}_{r_{N-1}}}.$$

**Step 4 (Tensor-level identity).** Stacking the identities in Step 3 over all $(i_1, \cdots, i_{N-1})$ shows that $\mathcal{X}$ is obtained by first selecting the basis slices and then recombining them with the coefficients:

$$\mathcal{X} = \mathcal{X} \times_{n=1}^{N-1} \mathbf{H}^{(n;R_n)} \mathbf{W}^{(n;R_n)}.$$

Unfolding this equality along mode $N$ and using the same product–unfolding identity as in Step 1 gives

$$\mathbf{X}_{(N)} = \left(\mathcal{X} \times_{n=1}^{N-1} \mathbf{H}^{(n;R_n)} \mathbf{W}^{(n;R_n)}\right)_{(N)} = \mathbf{X}_{(N)} \left(\bigotimes_{n=1}^{N-1} \mathbf{W}^{(N-n;R_{N-n})^\top}\right) \left(\bigotimes_{n=1}^{N-1} \mathbf{H}^{(N-n;R_{N-n})^\top}\right).$$

Substituting the expression for $\mathbf{F}_{\text{core}}$ from Step 1 yields

$$\mathbf{X}_{(N)} = \mathbf{F}_{\text{core}} \bigotimes_{n=1}^{N-1} \mathbf{H}^{(N-n;R_{N-n})^\top}.$$

This completes the proof. $\qquad\square$

## 7.3 Residual Error Bounds for Approximate Slice Bases

Theorem 3.1 establishes that when mode-wise bases are selected exactly, the full tensor can be reconstructed from the resulting core fibers without loss. In practice, however, the selected slices only approximate the true bases of the mode subspaces, leading to reconstruction errors tied to the residuals of these approximations. In this section, we provide two upper bounds that quantify how such residual errors behave when the projection operators deviate from the true subspace projections. Let $\mathcal{X} \in \mathbb{R}^{I_1 \times \cdots \times I_N}$ denote the original tensor, and let

$$\widehat{\mathcal{X}} = \mathcal{X} \times_{n=1}^{N-1} \mathbf{P}_n, \qquad \mathbf{P}_n = \mathbf{H}^{(n;R_n)} \mathbf{W}^{(n;R_n)},$$

be the reconstructed tensor obtained using approximate mode-wise bases. The following theorem characterizes the residual error $\|\mathcal{X} - \widehat{\mathcal{X}}\|_F$ through two complementary inequalities: one capturing the distortion caused by the mode-wise projection operators $\mathbf{P}_n$, and another bounding the accumulated spectral amplification introduced by the multi-linear Kronecker structure.

**Theorem 7.1.** Let $\mathcal{X}$ and $\widehat{\mathcal{X}}$ be defined as above. Then, the reconstruction error satisfies the following bounds:

(1)

$$\|\mathcal{X} - \widehat{\mathcal{X}}\|_F \leq \sum_{n=1}^{N-1} \|\mathbf{X}_{(n)}\|_F \left(\max_i |1 - \sigma_i(\mathbf{P}_n)| + \sigma_1(\mathbf{P}_n) \max\left(|1 - \sigma_{\min}|, |1 - \sigma_{\max}|\right)\right),$$

where $\mathbf{X}_{(n)}$ is the mode-$n$ unfolding of $\mathcal{X}$, $\sigma_i(\mathbf{P}_n)$ are the singular values of $\mathbf{P}_n$, and

$$\sigma_{\max} = \prod_{\ell \neq n} \sigma_1(\mathbf{P}_\ell), \qquad \sigma_{\min} = \prod_{\ell \neq n} \sigma_{I_\ell}(\mathbf{P}_\ell).$$

(2)

$$\|\mathcal{X} - \widehat{\mathcal{X}}\|_F \leq \|\mathbf{X}_{(N)}\|_F \left( 1 + \prod_{n=1}^{N-1} \|\mathbf{H}^{(n;R_n)^\top}\|_2 \right),$$

where $\mathbf{X}_{(N)}$ is the mode-$N$ unfolding of $\mathcal{X}$.

*Proof.* We first establish the bound in (1). Let $\mathbf{X}_{(n)}$ denote the mode-$n$ unfolding of $\mathcal{X}$. The reconstructed unfolding is

$$\widehat{\mathbf{X}}_{(n)} = \mathbf{P}_n \mathbf{X}_{(n)} \mathbf{Q}_n^\top, \qquad \mathbf{Q}_n = \bigotimes_{\ell \neq n} \mathbf{P}_\ell,$$

and the residual is

$$\mathbf{E}_{(n)} = \mathbf{X}_{(n)} - \widehat{\mathbf{X}}_{(n)} = \mathbf{X}_{(n)} - \mathbf{P}_n \mathbf{X}_{(n)} \mathbf{Q}_n^\top.$$

Applying the triangle inequality and submultiplicativity of the Frobenius norm gives

$$\|\mathbf{E}_{(n)}\|_F \leq \|\mathbf{X}_{(n)}\|_F \left( \|\mathbf{I} - \mathbf{P}_n\|_2 + \|\mathbf{P}_n\|_2 \|\mathbf{I} - \mathbf{Q}_n^\top\|_2 \right).$$

Because $\mathbf{Q}_n$ is a Kronecker product of $\mathbf{P}_\ell$ matrices, its singular values are all products of the singular values of these factors. Denoting

$$\sigma_{\max} = \prod_{\ell \neq n} \sigma_1(\mathbf{P}_\ell), \qquad \sigma_{\min} = \prod_{\ell \neq n} \sigma_{I_\ell}(\mathbf{P}_\ell),$$

the stated bound follows. Summing the contributions for all $n = 1, \dots, N-1$ yields (1). For (2), note that

$$\|\mathcal{X} - \widehat{\mathcal{X}}\|_F = \|\mathbf{X}_{(N)} - \widehat{\mathbf{X}}_{(N)}\|_F = \left\| \mathbf{X}_{(N)} \left( \mathbf{I} - \bigotimes_{n=1}^{N-1} \mathbf{P}_n \right)^\top \right\|_F.$$

Using $\|\mathbf{AB}\|_F \leq \|\mathbf{A}\|_F \|\mathbf{B}\|_2$ and submultiplicativity of the spectral norm,

$$\|\mathcal{X} - \widehat{\mathcal{X}}\|_F \leq \|\mathbf{X}_{(N)}\|_F \left( 1 + \prod_{n=1}^{N-1} \|\mathbf{P}_n^\top\|_2 \right) \leq \|\mathbf{X}_{(N)}\|_F \left( 1 + \prod_{n=1}^{N-1} \|\mathbf{H}^{(n;R_n)^\top}\|_2 \right),$$

which completes the proof. $\square$

## 7.4 Optimization

We now detail the optimization procedure of the proposed MSLFS method given in Problem (12), describing the iterative steps for solving its objective function and updating the associated optimization variables.

$$\min_{\mathbf{H}^{(n;m_n)}, \mathbf{W}^{(n;m_n)} \geq 0, \forall n \in \{1,2\}} \frac{1}{2} \|\mathcal{X} - \mathcal{X} \times_1 \mathbf{H}^{(1;m_1)} \mathbf{W}^{(1;m_1)} \times_2 \mathbf{H}^{(2;m_2)} \mathbf{W}^{(2;m_2)} \|_F^2$$

$$+ \frac{\alpha}{2} \operatorname{Tr}(\mathbf{H}^{(2;m_2)^\top} \mathbf{L}^{(2)} \mathbf{H}^{(2;m_2)}) \operatorname{Tr}(\mathbf{H}^{(1;m_1)^\top} \mathbf{L}^{(1)} \mathbf{H}^{(1;m_1)})$$

$$+ \frac{\beta}{2} \operatorname{Tr}(\mathbf{W}^{(2;m_2)} \mathbf{U}^{(2)} \mathbf{W}^{(2;m_2)^\top}) \operatorname{Tr}(\mathbf{W}^{(1;m_1)} \mathbf{U}^{(1)} \mathbf{W}^{(1;m_1)^\top})$$

$$\text{s.t.} \quad \mathbf{W}^{(2;m_2)} \mathbf{W}^{(2;m_2)^\top} \otimes \mathbf{W}^{(1;m_1)} \mathbf{W}^{(1;m_1)^\top} = \mathbf{I}_{m_1 m_2}. \qquad (12)$$

To derive the multiplicative updating rules for $\mathbf{W}^{(1;m_1)}$ and $\mathbf{H}^{(1;m_1)}$, one must calculate the derivatives of the objective function with respect to these variables and set them equal to zero. To this end, the first term of the objective function can be unfolded as follows:

$$
\min_{\mathbf{H}^{(n;m_n)}, \mathbf{W}^{(n;m_n)} \geq 0, \forall n \in \{1,2\}} \frac{1}{2}\|\mathbf{X}_{(1)} - \mathbf{H}^{(1;m_1)}\mathbf{W}^{(1;m_1)}\mathbf{X}_{(1)}(\mathbf{I}_{I_3} \otimes \mathbf{H}^{(2;m_2)}\mathbf{W}^{(2;m_2)})^\top\|_F^2
$$

$$
+ \frac{\alpha}{2}\mathrm{Tr}[\mathbf{H}^{(2;m_2)^\top}\mathbf{L}^{(2)}\mathbf{H}^{(2;m_2)}]\mathrm{Tr}[\mathbf{H}^{(1;m_1)^\top}\mathbf{L}^{(1)}\mathbf{H}^{(1;m_1)}]
$$

$$
+ \frac{\beta}{2}\mathrm{Tr}[\mathbf{W}^{(2;m_2)}\mathbf{U}^{(2)}\mathbf{W}^{(2;m_2)^\top}]\mathrm{Tr}[\mathbf{W}^{(1;m_1)}\mathbf{U}^{(1)}\mathbf{W}^{(1;m_1)^\top}]
$$

$$
\text{s.t.} \quad \mathbf{W}^{(2;m_2)}\mathbf{W}^{(2;m_2)^\top} \otimes \mathbf{W}^{(1;m_1)}\mathbf{W}^{(1;m_1)^\top} = \mathbf{I}_{m_1 m_2}. \tag{13}
$$

Simplifying the objective function leads us to:

$$
\frac{1}{2}\mathrm{Tr}[\mathbf{X}_{(1)}^\top\mathbf{X}_{(1)} - 2(\mathbf{I}_{I_3} \otimes \mathbf{H}^{(2;m_2)}\mathbf{W}^{(2;m_2)})\mathbf{X}_{(1)}^\top\mathbf{W}^{(1;m_1)^\top}\mathbf{H}^{(1;m_1)^\top}\mathbf{X}_{(1)}
$$

$$
+ (\mathbf{I}_{I_3} \otimes \mathbf{H}^{(2;m_2)}\mathbf{W}^{(2;m_2)})\mathbf{X}_{(1)}^\top\mathbf{W}^{(1;m_1)^\top}\mathbf{H}^{(1;m_1)^\top}\mathbf{H}^{(1;m_1)}\mathbf{W}^{(1;m_1)}\mathbf{X}_{(1)}(\mathbf{I}_{I_3} \otimes \mathbf{H}^{(2;m_2)}\mathbf{W}^{(2;m_2)})^\top]
$$

$$
+ \frac{\alpha}{2}\mathrm{Tr}[\mathbf{H}^{(2;m_2)^\top}\mathbf{L}^{(2)}\mathbf{H}^{(2;m_2)}]\mathrm{Tr}[\mathbf{H}^{(1;m_1)^\top}\mathbf{L}^{(1)}\mathbf{H}^{(1;m_1)}]
$$

$$
+ \frac{\beta}{2}\mathrm{Tr}[\mathbf{W}^{(2;m_2)}\mathbf{U}^{(2)}\mathbf{W}^{(2;m_2)^\top}]\mathrm{Tr}[\mathbf{W}^{(1;m_1)}\mathbf{U}^{(1)}\mathbf{W}^{(1;m_1)^\top}]
$$

$$
+ \frac{\gamma}{4}\mathrm{Tr}[\mathbf{W}^{(2;m_2)}\mathbf{W}^{(2;m_2)^\top}\mathbf{W}^{(2;m_2)}\mathbf{W}^{(2;m_2)^\top} \otimes \mathbf{W}^{(1;m_1)}\mathbf{W}^{(1;m_1)^\top}\mathbf{W}^{(1;m_1)}\mathbf{W}^{(1;m_1)^\top}
$$

$$
- 2\mathbf{W}^{(2;m_2)}\mathbf{W}^{(2;m_2)^\top} \otimes \mathbf{W}^{(1;m_1)}\mathbf{W}^{(1;m_1)^\top} + \mathbf{I}_{m_1 m_2}]. \tag{14}
$$

Now the the derivatives of the objective function w.r.t. $\mathbf{W}^{(1;m_1)}$ and $\mathbf{H}^{(1;m_1)}$ can be calculated as follows:

$$
\frac{\partial \mathcal{F}}{\partial \mathbf{W}^{(1;m_1)}} = -\mathbf{H}^{(1;m_1)^\top}\mathbf{X}_{(1)}(\mathbf{I}_{I_3} \otimes \mathbf{H}^{(2;m_2)}\mathbf{W}^{(2;m_2)})\mathbf{X}_{(1)}^\top
$$

$$
+ \mathbf{H}^{(1;m_1)^\top}\mathbf{H}^{(1;m_1)}\mathbf{W}^{(1;m_1)}\mathbf{X}_{(1)}(\mathbf{I}_{I_3} \otimes \mathbf{H}^{(2;m_2)}\mathbf{W}^{(2;m_2)})^\top(\mathbf{I}_{I_3} \otimes \mathbf{H}^{(2;m_2)}\mathbf{W}^{(2;m_2)})\mathbf{X}_{(1)}^\top
$$

$$
+ \beta\mathrm{Tr}[\mathbf{W}^{(2;m_2)}\mathbf{U}^{(2)}\mathbf{W}^{(2;m_2)^\top}]\mathbf{W}^{(1;m_1)}\mathbf{U}^{(1)}
$$

$$
+ \gamma\mathrm{Tr}[\mathbf{W}^{(2;m_2)}\mathbf{W}^{(2;m_2)^\top}\mathbf{W}^{(2;m_2)}\mathbf{W}^{(2;m_2)^\top}]\mathbf{W}^{(1;m_1)}\mathbf{W}^{(1;m_1)^\top}\mathbf{W}^{(1;m_1)}
$$

$$
- \gamma\mathrm{Tr}[\mathbf{W}^{(2;m_2)}\mathbf{W}^{(2;m_2)^\top}]\mathbf{W}^{(1;m_1)}. \tag{15}
$$

$$
\frac{\partial \mathcal{F}}{\partial \mathbf{H}^{(1;m_1)}} = -\mathbf{X}_{(1)}(\mathbf{I}_{I_3} \otimes \mathbf{H}^{(2;m_2)}\mathbf{W}^{(2;m_2)})\mathbf{X}_{(1)}^\top\mathbf{W}^{(1;m_1)^\top}
$$

$$
+ \mathbf{H}^{(1;m_1)}\mathbf{W}^{(1;m_1)}\mathbf{X}_{(1)}(\mathbf{I}_{I_3} \otimes \mathbf{H}^{(2;m_2)}\mathbf{W}^{(2;m_2)})^\top(\mathbf{I}_{I_3} \otimes \mathbf{H}^{(2;m_2)}\mathbf{W}^{(2;m_2)})\mathbf{X}_{(1)}^\top\mathbf{W}^{(1;m_1)^\top}
$$

$$
+ \alpha\mathrm{Tr}[\mathbf{H}^{(2;m_2)^\top}\mathbf{L}^{(2)}\mathbf{H}^{(2;m_2)}]\mathbf{L}^{(1)}\mathbf{H}^{(1;m_1)}. \tag{16}
$$

According to KKT conditions (Lee & Seung, 1999), we have the following updating rules:

$$
\mathbf{W}^{(1;m_1)} = (\mathbf{W}^{(1;m_1)} \odot \mathbf{H}^{(1;m_1)^\top}\mathbf{X}_{(1)}(\mathbf{I}_{I_3} \otimes \mathbf{H}^{(2;m_2)}\mathbf{W}^{(2;m_2)})\mathbf{X}_{(1)}^\top \tag{17}
$$

$$
+ \gamma\mathrm{Tr}[\mathbf{W}^{(2;m_2)}\mathbf{W}^{(2;m_2)^\top}]\mathbf{W}^{(1;m_1)})
$$

$$
\oslash (\mathbf{H}^{(1;m_1)^\top}\mathbf{H}^{(1;m_1)}\mathbf{W}^{(1;m_1)}\mathbf{X}_{(1)}(\mathbf{I}_{I_3} \otimes \mathbf{H}^{(2;m_2)}\mathbf{W}^{(2;m_2)})^\top(\mathbf{I}_{I_3} \otimes \mathbf{H}^{(2;m_2)}\mathbf{W}^{(2;m_2)})\mathbf{X}_{(1)}^\top
$$

$$
+ \beta\mathrm{Tr}[\mathbf{W}^{(2;m_2)}\mathbf{U}^{(2)}\mathbf{W}^{(2;m_2)^\top}]\mathbf{W}^{(1;m_1)}\mathbf{U}^{(1)}
$$

$$
+ \gamma\mathrm{Tr}[\mathbf{W}^{(2;m_2)}\mathbf{W}^{(2;m_2)^\top}\mathbf{W}^{(2;m_2)}\mathbf{W}^{(2;m_2)^\top}]\mathbf{W}^{(1;m_1)}\mathbf{W}^{(1;m_1)^\top}\mathbf{W}^{(1;m_1)}) \tag{18}
$$

$$
\mathbf{H}^{(1;m_1)} = \mathbf{H}^{(1;m_1)} \odot (\mathbf{X}_{(1)}(\mathbf{I}_{I_3} \otimes \mathbf{H}^{(2;m_2)}\mathbf{W}^{(2;m_2)})\mathbf{X}_{(1)}^\top\mathbf{W}^{(1;m_1)^\top}
$$

$$
+ \alpha\mathrm{Tr}[\mathbf{H}^{(2;m_2)^\top}\mathbf{L}^{(2)}\mathbf{H}^{(2;m_2)}]\mathbf{A}^{(1)}\mathbf{H}_1)
$$

$$
\oslash (\mathbf{H}^{(1;m_1)}\mathbf{W}^{(1;m_1)}\mathbf{X}_{(1)}(\mathbf{I}_{I_3} \otimes \mathbf{H}^{(2;m_2)}\mathbf{W}^{(2;m_2))})^\top(\mathbf{I}_{I_3} \otimes \mathbf{H}^{(2;m_2)}\mathbf{W}^{(2;m_2)})\mathbf{X}_{(1)}^\top\mathbf{W}^{(2;m_2)^\top}
$$

$$
+ \alpha\mathrm{Tr}[\mathbf{H}^{(2;m_2)^\top}\mathbf{L}^{(2)}\mathbf{H}^{(2;m_2)}]\mathbf{B}^{(1)}\mathbf{H}^{(1;m_1)}). \tag{19}
$$

To derive the update rules for $\mathbf{W}^{(2;m_2)}$ and $\mathbf{H}^{(2;m_2)}$, the first term of (12) must be reformulated using the mode-2 unfolding of the tensor. Then, the derivatives of (12) with respect to these variables are computed.

$$\min_{\mathbf{H}^{(n;m_n)},\mathbf{W}^{(n;m_n)}\geq 0,\forall n\in\{1,2\}} \frac{1}{2}\|\mathbf{X}_{(2)} - \mathbf{H}^{(2;m_2)}\mathbf{W}^{(2;m_2)}\mathbf{X}_{(2)}(\mathbf{I}_{I_3}\otimes\mathbf{H}^{(1;m_1)}\mathbf{W}^{(1;m_1)})^{\top}\|_F^2$$

$$+ \frac{\alpha}{2}\mathrm{Tr}[\mathbf{H}^{(2;m_2)^{\top}}\mathbf{L}^{(2)}\mathbf{H}^{(2;m_2)}]\mathrm{Tr}[\mathbf{H}^{(1;m_1)^{\top}}\mathbf{L}^{(1)}\mathbf{H}^{(1;m_1)}]$$

$$+ \frac{\beta}{2}\mathrm{Tr}[\mathbf{W}^{(2;m_2)}\mathbf{U}^{(2)}\mathbf{W}^{(2;m_2)^{\top}}]\mathrm{Tr}[\mathbf{W}^{(1;m_1)}\mathbf{U}^{(1)}\mathbf{W}^{(1;m_1)^{\top}}]$$

$$\text{s.t.}\quad \mathbf{W}^{(2;m_2)}\mathbf{W}^{(2;m_2)^{\top}}\otimes\mathbf{W}^{(1;m_1)}\mathbf{W}^{(1;m_1)^{\top}} = \mathbf{I}_{m_1 m_2}. \qquad (20)$$

$$\frac{\partial\mathcal{F}}{\partial\mathbf{W}^{(2;m_2)}} = -\mathbf{H}^{(2;m_2)^{\top}}\mathbf{X}_{(2)}(\mathbf{I}_{I_3}\otimes\mathbf{H}^{(1;m_1)}\mathbf{W}^{(1;m_1)})\mathbf{X}_{(2)}^{\top}$$

$$+ \mathbf{H}^{(2;m_2)^{\top}}\mathbf{H}^{(2;m_2)}\mathbf{W}^{(2;m_2)}\mathbf{X}_{(2)}(\mathbf{I}_{I_3}\otimes\mathbf{H}^{(1;m_1)}\mathbf{W}^{(1;m_1)})^{\top}(\mathbf{I}_{I_3}\otimes\mathbf{H}^{(1;m_1)}\mathbf{W}^{(1;m_1)})\mathbf{X}_{(2)}^{\top}$$

$$+ \beta\mathrm{Tr}[\mathbf{W}^{(1;m_1)}\mathbf{U}^{(1)}\mathbf{W}^{(1;m_1)^{\top}}]\mathbf{W}^{(2;m_2)}\mathbf{U}^{(2)}$$

$$+ \gamma\mathrm{Tr}[\mathbf{W}^{(1;m_1)}\mathbf{W}^{(1;m_1)^{\top}}\mathbf{W}^{(1;m_1)}\mathbf{W}^{(1;m_1)^{\top}}]\mathbf{W}^{(2;m_2)}\mathbf{W}^{(2;m_2)^{\top}}\mathbf{W}^{(2;m_2)}$$

$$- \gamma\mathrm{Tr}[\mathbf{W}^{(1;m_1)}\mathbf{W}^{(1;m_1)^{\top}}]\mathbf{W}^{(2;m_2)}. \qquad (21)$$

$$\frac{\partial\mathcal{F}}{\partial\mathbf{H}^{(2;m_2)}} = -\mathbf{X}_{(2)}(\mathbf{I}_{I_3}\otimes\mathbf{H}^{(1;m_1)}\mathbf{W}^{(1;m_1)})\mathbf{X}_{(2)}^{\top}\mathbf{W}^{(2;m_2)^{\top}}$$

$$+ \mathbf{H}^{(2;m_2)}\mathbf{W}^{(2;m_2)}\mathbf{X}_{(2)}(\mathbf{I}_{I_3}\otimes\mathbf{H}^{(1;m_1)}\mathbf{W}^{(1;m_1)})^{\top}(\mathbf{I}_{I_3}\otimes\mathbf{H}^{(1;m_1)}\mathbf{W}^{(1;m_1)})\mathbf{X}_{(2)}^{\top}\mathbf{W}^{(2;m_2)^{\top}}$$

$$+ \alpha\mathrm{Tr}[\mathbf{H}^{(1;m_1)^{\top}}\mathbf{L}^{(1)}\mathbf{H}^{(1;m_1)}]\mathbf{L}^{(2)}\mathbf{H}^{(2;m_2)}. \qquad (22)$$

According to KKT conditions (Lee & Seung, 1999), we have the following updating rules:

$$\mathbf{W}^{(2;m_2)} = \mathbf{W}^{(2;m_2)}\odot(\mathbf{H}^{(2;m_2)^{\top}}\mathbf{X}_{(2)}(\mathbf{I}_{I_3}\otimes\mathbf{H}^{(1;m_1)}\mathbf{W}^{(1;m_1)})\mathbf{X}_{(2)}^{\top} \qquad (23)$$

$$+ \gamma\mathrm{Tr}[\mathbf{W}^{(1;m_1)}\mathbf{W}^{(1;m_1)^{\top}}]\mathbf{W}^{(2;m_2)})$$

$$\oslash(\mathbf{H}^{(2;m_2)^{\top}}\mathbf{H}^{(2;m_2)}\mathbf{W}^{(2;m_2)}\mathbf{X}_{(2)}(\mathbf{I}_{I_3}\otimes\mathbf{H}^{(1;m_1)}\mathbf{W}^{(1;m_1)})^{\top}(\mathbf{I}_{I_3}\otimes\mathbf{H}^{(1;m_1)}\mathbf{W}^{(1;m_1)})\mathbf{X}_{(2)}^{\top}$$

$$+ \beta\mathrm{Tr}[\mathbf{W}^{(1;m_1)}\mathbf{U}^{(1)}\mathbf{W}^{(1;m_1)^{\top}}]\mathbf{W}^{(2;m_2)}\mathbf{U}^{(2)}$$

$$+ \gamma\mathrm{Tr}[\mathbf{W}^{(1;m_1)}\mathbf{W}^{(1;m_1)^{\top}}\mathbf{W}^{(1;m_1)}\mathbf{W}^{(1;m_1)^{\top}}]\mathbf{W}^{(2;m_2)}\mathbf{W}^{(2;m_2)^{\top}}\mathbf{W}^{(2;m_2)}), \qquad (24)$$

$$\mathbf{H}^{(2;m_2)} = \mathbf{H}^{(2;m_2)}\odot(\mathbf{X}_{(2)}(\mathbf{I}_{I_3}\otimes\mathbf{H}^{(1;m_1)}\mathbf{W}^{(1;m_1)})\mathbf{X}_{(2)}^{\top}\mathbf{W}^{(2;m_2)^{\top}}$$

$$+ \alpha\mathrm{Tr}[\mathbf{H}^{(1;m_1)^{\top}}\mathbf{L}^{(1)}\mathbf{H}^{(1;m_1)}]\mathbf{A}^{(2)}\mathbf{H}^{(2;m_2)})$$

$$\oslash(\mathbf{H}^{(2;m_2)}\mathbf{W}^{(2;m_2)}\mathbf{X}_{(2)}(\mathbf{I}_{I_3}\otimes\mathbf{H}^{(1;m_1)}\mathbf{W}^{(;m_1)})^{\top}(\mathbf{I}_{I_3}\otimes\mathbf{H}^{(1;m_1)}\mathbf{W}^{(1;m_1)})\mathbf{X}_{(2)}^{\top}\mathbf{W}^{(2;m_2)^{\top}}$$

$$+ \alpha\mathrm{Tr}[\mathbf{H}^{(1;m_1)^{\top}}\mathbf{L}^{(1)}\mathbf{H}^{(1;m_1)}]\mathbf{B}^{(2)}\mathbf{H}^{(2;m_2)}). \qquad (25)$$

## 7.5 CONVERGENCE ANALYSIS

This section iinvestigates the convergence analysis of MSLFS to explore the decreasing behavior of the objective function (12). It is first assumed that each matrix $\mathbf{W}^{(n;m_n)}$, $\mathbf{H}^{(n;m_n)}$, for $n\in\{1,2\}$ is individually updated while the others remain unchanged. Based on this assumption, the decreasing behavior of the objective function is analyzed for each variable. For this purpose, several important definitions and findings from (Lee & Seung, 1999) are examined.

**Definition 7.2** ((Lee & Seung, 1999))**.** The function $G(u, u^{(t)})$ is deemed an auxiliary function for $f(u)$ if it fulfills the subsequent criteria:

$$g(u, u^{(t)}) \geq f(u), \quad g(u, u) = f(u), \qquad (26)$$

for every $u\in\mathbb{R}$.

**Lemma 1** ((Lee & Seung, 1999))**.** *Suppose* $g(u, u^{(t)})$ *is an auxiliary function associated with* $f(u)$*. Then, the sequence* $\{f(u^{(t)})\}_{t=1}^{\infty}$ *is non-increasing when* $u$ *is updated according to*

$$u^{(t+1)} = \arg\min_{u \in \mathbb{R}} g(u, u^{(t)}).$$

In Proposition 7.3, an auxiliary function is created to ensure that the original objective function diminishes monotonically in line with the update rule for $\mathbf{W}^{(1;m_1)}$ specified in (17).

**Proposition 7.3.** Given that the matrices $\mathbf{H}^{(1;m_1)}, \mathbf{W}^{(2;m_2)}$, and $\mathbf{H}^{(2;m_2)}$ are fixed, the update rule (17) for $\mathbf{W}^{(1;m_1)}$ ensures that the objective function of the minimization problem (12) does not increase.

*Proof.* Assume that the matrices $\mathbf{H}^{(1;m_1)}, \mathbf{W}^{(2;m_2)}$, and $\mathbf{H}^{(2;m_2)}$ are fixed. Consider the objective function in the optimization problem (12) with respect to $\mathbf{W}^{(1;m_1)}$:

$$
\begin{aligned}
f(\mathbf{W}^{(1;m_1)}) =& \frac{1}{2} \|\mathcal{X} - \mathcal{X} \times_1 \mathbf{H}^{(1;m_1)}\mathbf{W}^{(1;m_1)} \times_2 \mathbf{H}^{(2;m_2)}\mathbf{W}^{(2;m_2)}\|_F^2 \\
&+ \frac{\beta}{2} \mathrm{Tr}[\mathbf{W}^{(2;m_2)}\mathbf{U}^{(2)}\mathbf{W}^{(2;m_2)^{\top}}] \mathrm{Tr}[\mathbf{W}^{(1;m_1)}\mathbf{U}^{(1)}\mathbf{W}^{(1;m_1)^{\top}}] \\
&+ \frac{\gamma}{4} \|\mathbf{W}^{(2;m_2)}\mathbf{W}^{(2;m_2)^{\top}} \otimes \mathbf{W}^{(1;m_1)}\mathbf{W}^{(1;m_1)^{\top}} - \mathbf{I}_{m_1 m_2}\|_F^2.
\end{aligned}
$$

To show that $f(\mathbf{W}^{(1;m_1)(t+1)}) \leq f(\mathbf{W}^{(1;m_1)(t)})$, define $g(w_1, f(w_{j_1,i_1}^{(1;m_1)(t)}))$ as follows:

$$
\begin{aligned}
g(w_1, w_{j_1,i_1}^{(1;m_1)(t)}) =& \mathcal{B}(w_{j_1,i_1}^{(1;m_1)(t)}) + \dot{\mathcal{B}}(w_{j_1,i_1}^{(1;m_1)(t)})(w_1 - w_{j_1,i_1}^{(1;m_1)(t)}) \\
&+ \Big( \mathbf{H}^{(1;m_1)^{\top}}\mathbf{H}^{(1;m_1)}\mathbf{W}^{(1;m_1)(t)}\mathbf{X}_{(1)}(\mathbf{I}_{I_3} \otimes \mathbf{H}^{(2;m_2)}\mathbf{W}^{(2;m_2)})^{\top}(\mathbf{I}_{I_3} \otimes \mathbf{H}^{(2;m_2)}\mathbf{W}^{(2;m_2)})\mathbf{X}_{(1)}^{\top} \\
&+ \beta \mathrm{Tr}[\mathbf{W}^{(2;m_2)}\mathbf{U}^{(2)}\mathbf{W}^{(2;m_2)^{\top}}]\mathbf{W}^{(1;m_1)(t)}\mathbf{U}^{(1)} \\
&+ \gamma \mathrm{Tr}[\mathbf{W}^{(2;m_2)}\mathbf{W}^{(2;m_2)^{\top}}\mathbf{W}^{(2;m_2)}\mathbf{W}^{(2;m_2)^{\top}}]\mathbf{W}^{(1;m_1)(t)}\big(\mathbf{W}^{(1;m_1)(t)}\big)^{\top}\mathbf{W}^{(1;m_1)(t)} \Big)_{j_1,i_1} \\
&\times \frac{(w_1 - w_{j_1,i_1}^{(1;m_1)(t)})^2}{2\,w_{j_1,i_1}^{(1;m_1)(t)}},
\end{aligned}
$$

for $j_1 = 1, 2, \cdots, m_1$ and $i_1 = 1, 2, \cdots, I_1$. Moreover, assume that $\mathcal{B}(w_1)$ indicates the part of $f(w)$ relevant to $\mathbf{W}_{j_1,i_1}^{(1;m_1)}$, and

$$
\begin{aligned}
\dot{\mathcal{B}}(w_1) :=& \left( \frac{\partial f}{\partial \mathbf{W}^{(1;m_1)}} \right)_{j_1,i_1} = \Big( -\mathbf{H}^{(1;m_1)^{\top}}\mathbf{X}_{(1)}(\mathbf{I}_{I_3} \otimes \mathbf{H}^{(2;m_2)}\mathbf{W}^{(2;m_2)})\mathbf{X}_{(1)}^{\top} \\
&+ \mathbf{H}^{(1;m_1)^{\top}}\mathbf{H}^{(1;m_1)}\mathbf{W}^{(1;m_1)}\mathbf{X}_{(1)}(\mathbf{I}_{I_3} \otimes \mathbf{H}^{(2;m_2)}\mathbf{W}^{(2;m_2)})^{\top}(\mathbf{I}_{I_3} \otimes \mathbf{H}^{(2;m_2)}\mathbf{W}^{(2;m_2)})\mathbf{X}_{(1)}^{\top} \\
&+ \beta \mathrm{Tr}[\mathbf{W}^{(2;m_2)}\mathbf{U}^{(2)}\mathbf{W}^{(2;m_2)^{\top}}]\mathbf{W}^{(1;m_1)}\mathbf{U}^{(1)} \\
&+ \gamma \mathrm{Tr}[\mathbf{W}^{(2;m_2)}\mathbf{W}^{(2;m_2)^{\top}}\mathbf{W}^{(2;m_2)}\mathbf{W}^{(2;m_2)^{\top}}]\mathbf{W}^{(1;m_1)}\mathbf{W}^{(1;m_1)^{\top}}\mathbf{W}^{(1;m_1)} \\
&- \gamma \mathrm{Tr}[\mathbf{W}^{(2;m_2)}\mathbf{W}^{(2;m_2)^{\top}}]\mathbf{W}^{(1;m_1)} \Big)_{j_1,i_1}.
\end{aligned}
$$

It can be seen that $g(w_1, w_{j_1,i_1}^{(1;m_1)(t)})$ is an auxiliary function of $\mathcal{B}(w_1)$. For this purpose, consider the Taylor expansion of $\mathcal{B}(w_1)$ around $w_{j_1,i_1}^{(1;m_1)(t)}$:

$$\mathcal{B}(w_1) = \mathcal{B}(w_{j_1,i_1}^{(1;m_1)(t)}) + \dot{\mathcal{B}}(w_{j_1,i_1}^{(1;m_1)(t)})(w_1 - w_{j_1,i_1}^{(1;m_1)(t)}) + \frac{1}{2}\ddot{\mathcal{B}}(w_{j_1,i_1}^{(1;m_1)(t)})(w_1 - w_{j_1,i_1}^{(1;m_1)(t)})^2,$$

where

$$
\ddot{\mathcal{B}}(w_1) := \left(\frac{\partial^2 F}{\partial \mathbf{W}^{(1;m_1)^2}}\right)_{j_1,i_1} = \left(\mathbf{H}^{(1;m_1)^\top}\mathbf{H}^{(1;m_1)}\right)_{j_1,j_1}
$$

$$
\times \left(\mathbf{X}_{(1)}(\mathbf{I}_{I_3}\otimes\mathbf{H}^{(2;m_2)}\mathbf{W}^{(2;m_2)})^\top(\mathbf{I}_{I_3}\otimes\mathbf{H}^{(2;m_2)}\mathbf{W}^{(2;m_2)})\mathbf{X}_{(1)}^\top\right)_{i_1,i_1}
$$

$$
+ \beta\mathrm{Tr}[\mathbf{W}^{(2;m_2)}\mathbf{U}^{(2)}\mathbf{W}^{(2;m_2)^\top}]u_{j_1,i_1}^{(1)}
$$

$$
+ \gamma\,\mathrm{Tr}[\mathbf{W}^{(2;m_2)}\mathbf{W}^{(2;m_2)^\top}\mathbf{W}^{(2;m_2)}\mathbf{W}^{(2;m_2)^\top}]\left(\left(\mathbf{W}^{(1;m_1)^\top}\mathbf{W}^{(1;m_1)}\right)_{i_1,i_1} + w_{j_1,i_1}^{(1,m_1)^2}\right.
$$

$$
\left. + \left(\mathbf{W}^{(1;m_1)}\mathbf{W}^{(1;m_1)^\top}\right)_{j_1,j_1}\right) - \gamma\,\mathrm{Tr}[\mathbf{W}^{(2;m_2)}\mathbf{W}^{(2;m_2)^\top}].
$$

It is easy to validate that $g(w_1, w_1) = \mathcal{B}(w_1)$. Moreover, in light of the following inequalities,

$$
\left(\mathbf{H}^{(1;m_1)^\top}\mathbf{H}^{(1;m_1)}\mathbf{W}^{(1;m_1)}\mathbf{X}_{(1)}(\mathbf{I}_{I_3}\otimes\mathbf{H}^{(2;m_2)}\mathbf{W}^{(2;m_2)})^\top(\mathbf{I}_{I_3}\otimes\mathbf{H}^{(2;m_2)}\mathbf{W}^{(2;m_2)})\mathbf{X}_{(1)}^\top\right)_{j_1,i_1}
$$

$$
= \sum_{r=1}^{m_1}\sum_{s=1}^{I_1}\left(\mathbf{H}^{(1;m_1)^\top}\mathbf{H}^{(1;m_1)}\right)_{j_1,r}w_{r,s}^{(1;m_1)}\times\left(\mathbf{X}_{(1)}(\mathbf{I}_{I_3}\otimes\mathbf{H}^{(2;m_2)}\mathbf{W}^{(2;m_2)})^\top(\mathbf{I}_{I_3}\otimes\mathbf{H}^{(2;m_2)}\mathbf{W}^{(2;m_2)})\mathbf{X}_{(1)}^\top\right)_{s,i_1}
$$

$$
\geq \left(\mathbf{H}^{(1;m_1)^\top}\mathbf{H}^{(1;m_1)}\right)_{j_1,j_1}\left(\mathbf{X}_{(1)}(\mathbf{I}_{I_3}\otimes\mathbf{H}^{(2;m_2)}\mathbf{W}^{(2;m_2)})^\top(\mathbf{I}_{I_3}\otimes\mathbf{H}^{(2;m_2)}\mathbf{W}^{(2;m_2)})\mathbf{X}_{(1)}^\top\right)_{i_1,i_1},
$$

$$
\left(\mathbf{W}^{(1;m_1)}\mathbf{U}^{(1)}\right)_{j_1,i_1} = \sum_{s=1}^{I_1}w_{j_1,s}^{(1,m_1)}u_{s,i_1}^{(1)} \geq u_{j_1,i_1}^{(1)},
$$

and

$$
\left(\mathbf{W}^{(1;m_1)}\mathbf{W}^{(1;m_1)^\top}\mathbf{W}^{(1;m_1)}\right)_{j_1,i_1} = \sum_{s=1}^{I_1}\sum_{r=1}^{m_1}w_{j_1,s}^{(1,m_1)}w_{r,s}^{(1,m_1)}w_{r,i_1}^{(1,m_1)} \geq \left(\mathbf{W}^{(1;m_1)^\top}\mathbf{W}^{(1;m_1)}\right)_{i_1,i_1}
$$

$$
+ w_{j_1,i_1}^{(1,m_1)^2} + \left(\mathbf{W}^{(1;m_1)}\mathbf{W}^{(1;m_1)^\top}\right)_{j_1,j_1},
$$

it can be observed that $g(w_1, w_{j_1,i_1}^{(1;m_1)(t)}) \geq \mathcal{B}(w_1)$, for each $w_1 \in \mathbb{R}$. Consequently, since the requirements of Definition 26 are met $g(w_1, w_{j_1,i_1}^{(1;m_1)})$ serves as an auxiliary function for $\mathcal{B}(w_1)$. Then, by minimizing $g(w_1, w_{j_1,i_1}^{(1;m_1)(t)})$ with respect to $w_1$, the updating rule of $\mathbf{W}^{(1;m_1)}$ can be obtained in the form

$$
\mathbf{W}^{(1;m_1)} = \left(\mathbf{W}^{(1;m_1)}\odot\mathbf{H}^{(1;m_1)^\top}\mathbf{X}_{(1)}(\mathbf{I}_{I_3}\otimes\mathbf{H}^{(2;m_2)}\mathbf{W}^{(2;m_2)})\mathbf{X}_{(1)}^\top + \gamma\,\mathrm{Tr}[\mathbf{W}^{(2;m_2)}\mathbf{W}^{(2;m_2)^\top}]\mathbf{W}^{(1;m_1)}\right)
$$

$$
\oslash\left(\mathbf{H}^{(1;m_1)^\top}\mathbf{H}^{(1;m_1)}\mathbf{W}^{(1;m_1)}\mathbf{X}_{(1)}(\mathbf{I}_{I_3}\otimes\mathbf{H}^{(2;m_2)}\mathbf{W}^{(2;m_2)})^\top(\mathbf{I}_{I_3}\otimes\mathbf{H}^{(2;m_2)}\mathbf{W}^{(2;m_2)})\mathbf{X}_{(1)}^\top\right.
$$

$$
+ \beta\mathrm{Tr}[\mathbf{W}^{(2;m_2)}\mathbf{U}^{(2)}\mathbf{W}^{(2;m_2)^\top}]\mathbf{W}^{(1;m_1)}\mathbf{U}^{(1)}
$$

$$
\left. + \gamma\,\mathrm{Tr}[\mathbf{W}^{(2;m_2)}\mathbf{W}^{(2;m_2)^\top}\mathbf{W}^{(2;m_2)}\mathbf{W}^{(2;m_2)^\top}]\mathbf{W}^{(1;m_1)}\mathbf{W}^{(1;m_1)^\top}\mathbf{W}^{(1;m_1)}\right).
$$

The obtained result is in exact agreement with the update rule (17) specified for the matrix $\mathbf{W}^{(1;m_1)}$. Collectively, this result and Lemma 1 establish that the proposed update rule guarantees the monotonic decrease of the original objective function. $\qquad\square$

In line with the strategy described in Proposition 7.3, two separate cases can be analyzed for the update rules of $\mathbf{W}^{(2;m_2)}$, $\mathbf{H}^{(1;m_1)}$ and $\mathbf{H}^{(2;m_2)}$. For each case, an auxiliary function is introduced to ensure the monotonic decrease of the original objective function. The cases are outlined as follows:

**Case 1:** Assuming that $\mathbf{W}^{(1;m_1)}$, $\mathbf{H}^{(1;m_1)}$, and $\mathbf{H}^{(2;m_2)}$ are fixed, the update rule (23) for $\mathbf{W}^{(2;m_2)}$ guarantees that the objective function in the minimization problem (12) is non-increasing.

Under this scenario, the objective function with respect to $\mathbf{W}^{(2;m_2)}$ is expressed as

$$f(\mathbf{W}^{(2;m_2)}) = \frac{1}{2}\|\mathcal{X} - \mathcal{X} \times_1 \mathbf{H}^{(1;m_1)}\mathbf{W}^{(1;m_1)} \times_2 \mathbf{H}^{(2;m_2)}\mathbf{W}^{(2;m_2)}\|_F^2$$

$$+ \frac{\beta}{2}\operatorname{Tr}[\mathbf{W}^{(2;m_2)}\mathbf{U}^{(2)}\mathbf{W}^{(2;m_2)^\top}]\operatorname{Tr}[\mathbf{W}^{(1;m_1)}\mathbf{U}^{(1)}\mathbf{W}^{(1;m_1)^\top}]$$

$$+ \frac{\gamma}{4}\|\mathbf{W}^{(2;m_2)}\mathbf{W}^{(2;m_2)^\top} \otimes \mathbf{W}^{(1;m_1)}\mathbf{W}^{(1;m_1)^\top} - \mathbf{I}_{m_1 m_2}\|_F^2.$$

Next, by defining the function

$$g(w_2, w_{j_2,i_2}^{(2;m_2)^{(t)}}) = \mathcal{B}(w_{j_2,i_2}^{(2;m_2)^{(t)}}) + \dot{\mathcal{B}}(w_{j_2,i_2}^{(2;m_2)^{(t)}})(w_2 - w_{j_2,i_2}^{(2;m_2)^{(t)}})$$

$$+ \Big(\mathbf{H}^{(2;m_2)^\top}\mathbf{H}^{(2;m_2)}\mathbf{W}^{(2;m_2)^{(t)}}\mathbf{X}_{(2)}(\mathbf{I}_{I_3} \otimes \mathbf{H}^{(1;m_1)}\mathbf{W}^{(1;m_1)})^\top(\mathbf{I}_{I_3} \otimes \mathbf{H}^{(1;m_1)}\mathbf{W}^{(1;m_1)})\mathbf{X}_{(2)}^\top$$

$$+ \beta\operatorname{Tr}[\mathbf{W}^{(1;m_1)}\mathbf{U}^{(1)}\mathbf{W}^{(1;m_1)^\top}]\mathbf{W}^{(2;m_2)^{(t)}}\mathbf{U}^{(2)}$$

$$+ \gamma\operatorname{Tr}[\mathbf{W}^{(1;m_1)}\mathbf{W}^{(1;m_1)^\top}\mathbf{W}^{(1;m_1)}\mathbf{W}^{(1;m_1)^\top}]\mathbf{W}^{(2;m_2)^{(t)}}\big(\mathbf{W}^{(2;m_2)^{(t)}}\big)^\top\mathbf{W}^{(2;m_2)^{(t)}}\Big)_{j_2,i_2}$$

$$\times \frac{(w_2 - w_{j_2,i_2}^{(2;m_2)^{(t)}})^2}{2\,w_{j_2,i_2}^{(2;m_2)^{(t)}}},$$

it can be demonstrated that $g(w_2, w_{j_2,i_2}^{(2;m_2)^{(t)}})$ serves as an auxiliary function for $\mathcal{B}(w_2)$, for $j_2 = 1, 2, \ldots, m_2$, and $i_2 = 1, 2, \ldots, I_2$. Note that $\mathcal{B}(w_2)$ represents the components of $f(\mathbf{W}^{(2;m_2)})$ associated with $w_{j_2,i_2}^{(2;m_2)}$ and takes the form

$$\mathcal{B}(w_2) = \mathcal{B}(w_{j_2,i_2}^{(2;m_2)^{(t)}}) + \dot{\mathcal{B}}(w_{j_2,i_2}^{(2;m_2)^{(t)}})(w_2 - w_{j_2,i_2}^{(2;m_2)^{(t)}})$$

$$+ \frac{1}{2}\ddot{\mathcal{B}}(w_{j_2,i_2}^{(2;m_2)^{(t)}})(w_2 - w_{j_2,i_2}^{(2;m_2)^{(t)}})^2,$$

with

$$\dot{\mathcal{B}}(w_2) := \left(\frac{\partial f}{\partial \mathbf{W}^{(2;m_2)}}\right)_{j_2,i_2} = \Big( - \mathbf{H}^{(2;m_2)^\top}\mathbf{X}_{(2)}(\mathbf{I}_{I_3} \otimes \mathbf{H}^{(1;m_1)}\mathbf{W}^{(1;m_1)})\mathbf{X}_{(2)}^\top$$

$$+ \mathbf{H}^{(2;m_2)^\top}\mathbf{H}^{(2;m_2)}\mathbf{W}^{(2;m_2)}\mathbf{X}_{(2)}(\mathbf{I}_{I_3} \otimes \mathbf{H}^{(1;m_1)}\mathbf{W}^{(1;m_1)})^\top$$

$$\times (\mathbf{I}_{I_3} \otimes \mathbf{H}^{(1;m_1)}\mathbf{W}^{(1;m_1)})\mathbf{X}_{(2)}^\top$$

$$+ \beta\operatorname{Tr}[\mathbf{W}^{(1;m_1)}\mathbf{U}^{(1)}\mathbf{W}^{(1;m_1)^\top}]\mathbf{W}^{(2;m_2)}\mathbf{U}^{(2)}$$

$$+ \gamma\operatorname{Tr}[\mathbf{W}^{(1;m_1)}\mathbf{W}^{(1;m_1)^\top}\mathbf{W}^{(1;m_1)}\mathbf{W}^{(1;m_1)^\top}]\mathbf{W}^{(2;m_2)}\mathbf{W}^{(2;m_2)^\top}\mathbf{W}^{(2;m_2)}$$

$$- \gamma\operatorname{Tr}[\mathbf{W}^{(1;m_1)}\mathbf{W}^{(1;m_1)^\top}]\mathbf{W}^{(2;m_2)}\Big)_{j_2,i_2},$$

and

$$\ddot{\mathcal{B}}(w_2) := \left(\frac{\partial^2 f}{\partial \mathbf{W}^{(2;m_2)^2}}\right)_{j_2,i_2} = \Big(\mathbf{H}^{(2;m_2)^\top}\mathbf{H}^{(2;m_2)}\Big)_{j_2,j_2}$$

$$\times \Big(\mathbf{X}_{(2)}(\mathbf{I}_{I_3} \otimes \mathbf{H}^{(1;m_1)}\mathbf{W}^{(1;m_1)})^\top(\mathbf{I}_{I_3} \otimes \mathbf{H}^{(1;m_1)}\mathbf{W}^{(1;m_1)})\mathbf{X}_{(2)}^\top\Big)_{i_2,i_2}$$

$$+ \beta\operatorname{Tr}[\mathbf{W}^{(1;m)}\mathbf{U}^{(1)}\mathbf{W}^{(1;m_1)^\top}]u_{j_2,i_2}^{(2)}$$

$$+ \gamma\operatorname{Tr}[\mathbf{W}^{(1;m_1)}\mathbf{W}^{(1;m_1)^\top}\mathbf{W}^{(1;m_1)}\mathbf{W}^{(1;m_1)^\top}]$$

$$\times \Bigg(\big(\mathbf{W}^{(2;m_2)^\top}\mathbf{W}^{(2;m_2)}\big)_{i_2,i_2} + w_{j_2,i_2}^{(2,m_2)^2}$$

$$+ \big(\mathbf{W}^{(2;m_2)}\mathbf{W}^{(2;m_2)^\top}\big)_{j_2,j_2}\Bigg) - \gamma\operatorname{Tr}[\mathbf{W}^{(1;m_1)}\mathbf{W}^{(1;m_1)^\top}].$$

**Case 2:** Assuming that $\mathbf{W}^{(1;m_1)}$, $\mathbf{W}^{(2;m_2)}$, and $\mathbf{H}^{(2;m_2)}$ are fixed, the update rule (19) for $\mathbf{H}^{(1;m_1)}$ guarantees that the objective function in the minimization problem (12) is non-increasing. Under this scenario, the objective function with respect to $\mathbf{H}^{(1;m_1)}$ is expressed as

$$f(\mathbf{H}^{(1;m_1)}) = \frac{1}{2}\|\mathcal{X} - \mathcal{X} \times_1 \mathbf{H}^{(1;m_1)}\mathbf{W}^{(1;m_1)} \times_2 \mathbf{H}^{(2;m_2)}\mathbf{W}^{(2;m_2)}\|_F^2$$
$$+ \frac{\alpha}{2}\,\mathrm{Tr}[\mathbf{H}^{(2;m_2)^\top}\mathbf{L}^{(2)}\mathbf{H}^{(2;m_2)}]\,\mathrm{Tr}[\mathbf{H}^{(1;m_1)^\top}\mathbf{L}^{(1)}\mathbf{H}^{(1;m_1)}].$$

Next, by defining the function

$$g(h_1, h_{i_1,j_1}^{(1;m_1)(t)}) = \mathcal{B}(h_{i_1,j_1}^{(1;m_1)(t)}) + \dot{\mathcal{B}}(h_{i_1,j_1}^{(1;m_1)(t)})(h_1 - h_{i_1,j_1}^{(1;m_1)(t)})$$
$$+ \left(\mathbf{H}^{(1;m_1)(t)}\mathbf{W}^{(1;m_1)}\mathbf{X}_{(1)}(\mathbf{I}_{I_3} \otimes \mathbf{H}^{(2;m_2)}\mathbf{W}^{(2;m_2)})^\top\right.$$
$$\times (\mathbf{I}_{I_3} \otimes \mathbf{H}^{(2;m_2)}\mathbf{W}^{(2;m_2)})\mathbf{X}_{(1)}^\top W^{(2;m_2)^\top}$$
$$+ \alpha\,\mathrm{Tr}[\mathbf{H}^{(2;m_2)^\top}\mathbf{L}^{(2)}\mathbf{H}^{(2;m_2)}]\mathbf{B}^{(1)}\mathbf{H}^{(1;m_1)}\Big)_{i_1,j_1}\frac{(h_1 - h_{i_1,j_1}^{(1;m_1)(t)})^2}{2\,h_{i_1,j_1}^{(1;m_1)(t)}},$$

it can be demonstrated that $g(h_1, h_{i_1,j_1}^{(1;m_1)(t)})$ serves as an auxiliary function for $\mathcal{B}(h_1)$, for $i_1 = 1, \ldots, I_1$, and $j_2 = 1, \ldots, m_1$. Note that $\mathcal{B}(h_1)$ represents the components of $f(\mathbf{H}^{(1;m_1)})$ associated with $h_{i_1,j_1}^{(1;m_1)}$ and takes the form

$$\mathcal{B}(h_1) = \mathcal{B}(h_{i_1,j_1}^{(1;m_1)(t)}) + \dot{\mathcal{B}}(h_{i_1,j_1}^{(1;m_1)(t)})(h_1 - h_{i_1,j_1}^{(1;m_1)(t)})$$
$$+ \frac{1}{2}\ddot{\mathcal{B}}(h_{i_1,j_1}^{(1;m_1)(t)})(h_1 - h_{i_1,j_1}^{(1;m_1)(t)})^2,$$

with

$$\dot{\mathcal{B}}(h_1) := \left(\frac{\partial f}{\partial \mathbf{H}^{(1;m_1)}}\right)_{i_1,j_1} = \left(-\mathbf{X}_{(1)}(\mathbf{I}_{I_3} \otimes \mathbf{H}^{(2;m_2)}\mathbf{W}^{(2;m_2)})\mathbf{X}_{(1)}^\top\mathbf{W}^{(1;m_1)^\top}\right.$$
$$+ \mathbf{H}^{(1;m_1)}\mathbf{W}^{(1;m_1)}\mathbf{X}_{(1)}(\mathbf{I}_{I_3} \otimes \mathbf{H}^{(2;m_2)}\mathbf{W}^{(2;m_2)})^\top$$
$$\times (\mathbf{I}_{I_3} \otimes \mathbf{H}^{(2;m_2)}\mathbf{W}^{(2;m_2)})\mathbf{X}_{(1)}^\top\mathbf{W}^{(1;m_1)^\top}$$
$$+ \alpha\mathrm{Tr}[\mathbf{H}^{(2;m_2)^\top}\mathbf{L}^{(2)}\mathbf{H}^{(2;m_2)}]\mathbf{L}^{(1)}\mathbf{H}^{(1;m_1)}\Big)_{i_1,j_1},$$

and

$$\ddot{\mathcal{B}}(h_1) := \left(\frac{\partial^2 f}{\partial \mathbf{H}^{(1;m_1)^2}}\right)_{i_1,j_1} = \left(\mathbf{W}^{(1;m_1)}\mathbf{X}_{(1)}(\mathbf{I}_{I_3} \otimes \mathbf{H}^{(2;m_2)}\mathbf{W}^{(2;m_2)})^\top\right.$$
$$\times (\mathbf{I}_{I_3} \otimes \mathbf{H}^{(2;m_2)}\mathbf{W}^{(2;m_2)})\mathbf{X}_{(1)}^\top\mathbf{W}^{(1;m_1)^\top}\Big)_{j_1,j_1}$$
$$+ \alpha\mathrm{Tr}[\mathbf{H}^{(2;m_2)^\top}\mathbf{L}^{(2)}\mathbf{H}^{(2;m_2)}]\ell_{i_1,i_1}^{(1)}.$$

**Case 3:** Assuming that $\mathbf{W}^{(1;m_1)}$, $\mathbf{W}^{(2;m_2)}$, and $\mathbf{H}^{(1;m_1)}$ are fixed, the update rule (25) for $\mathbf{H}^{(2;m_2)}$ guarantees that the objective function in the minimization problem (12) is non-increasing. Under this scenario, the objective function with respect to $\mathbf{H}^{(2;m_2)}$ is expressed as

$$f(\mathbf{H}^{(2;m_2)}) = \frac{1}{2}\|\mathcal{X} - \mathcal{X} \times_1 \mathbf{H}^{(1;m_1)}\mathbf{W}^{(1;m_1)} \times_2 \mathbf{H}^{(2;m_2)}\mathbf{W}^{(2;m_2)}\|_F^2$$
$$+ \frac{\alpha}{2}\,\mathrm{Tr}[\mathbf{H}^{(2;m_2)^\top}\mathbf{L}^{(2)}\mathbf{H}^{(2;m_2)}]\,\mathrm{Tr}[\mathbf{H}^{(1;m_1)^\top}\mathbf{L}^{(1)}\mathbf{H}^{(1;m_1)}].$$

Next, by defining the function

$$g(h_2, h_{i_2,j_2}^{(2;m_2)(t)}) = \mathcal{B}(h_{i_2,j_2}^{(2;m_2)(t)}) + \dot{\mathcal{B}}(h_{i_2,j_2}^{(2;m_2)(t)})(h_1 - h_{i_2,j_2}^{(2;m_2)(t)})$$
$$+ \left(\mathbf{H}^{(2;m_2)(t)}\mathbf{W}^{(2;m_2)}\mathbf{X}_{(2)}(\mathbf{I}_{I_3} \otimes \mathbf{H}^{(1;m_1)}\mathbf{W}^{(1;m_1)})^\top\right.$$
$$\times (\mathbf{I}_{I_3} \otimes \mathbf{H}^{(1;m_1)}\mathbf{W}^{(1;m_1)})\mathbf{X}_{(2)}^\top\mathbf{W}^{(2;m_2)^\top}$$
$$+ \alpha\,\mathrm{Tr}[\mathbf{H}^{(1;m_1)^\top}\mathbf{L}^{(1)}\mathbf{H}^{(1;m_1)}]\mathbf{B}^{(2)}\mathbf{H}^{(2;m_2)}\Big)_{i_2,j_2},$$

it can be demonstrated that $g(h_2, h_{i_2,j_2}^{(2;m_2)(t)})$ serves as an auxiliary function for $\mathcal{B}(h_2)$, for $i_2 = 1, \ldots, I_2$, and $j_2 = 1, \ldots, m_2$. Note that $\mathcal{B}(h_2)$ represents the components of $f(\mathbf{H}^{(2;m_2)})$ associated with $h_{i_2,j_2}^{(2;m_2)}$ and takes the form

$$\mathcal{B}(h_2) = \mathcal{B}(h_{i_2,j_2}^{(2;m_2)(t)}) + \dot{\mathcal{B}}(h_{i_2,j_2}^{(2;m_2)(t)})(h_2 - h_{i_2,j_2}^{(2;m_2)(t)}) + \frac{1}{2}\ddot{\mathcal{B}}(h_{i_2,j_2}^{(2;m_2)(t)})(h_2 - h_{i_2,j_2}^{(2;m_2)(t)})^2,$$

with

$$\dot{\mathcal{B}}(h_2) := \left(\frac{\partial f}{\partial \mathbf{H}^{(2;m_2)}}\right)_{i_2,j_2} = \Big( -\mathbf{X}_{(2)}(\mathbf{I}_{I_3} \otimes \mathbf{H}^{(1;m_1)}\mathbf{W}^{(1;m_1)})\mathbf{X}_{(2)}^\top \mathbf{W}^{(2;m_2)^\top}$$
$$+ \mathbf{H}^{(2;m_2)}\mathbf{W}^{(2;m_2)}\mathbf{X}_{(2)}(\mathbf{I}_{I_3} \otimes \mathbf{H}^{(1;m_1)}\mathbf{W}^{(1;m_1)})^\top$$
$$\times (\mathbf{I}_{I_3} \otimes \mathbf{H}^{(1;m_1)}\mathbf{W}^{(1;m_1)})\mathbf{X}_{(2)}^\top \mathbf{W}^{(2;m_2)^\top}$$
$$+ \alpha \mathrm{Tr}[\mathbf{H}^{(1;m_1)^\top}\mathbf{L}^{(1)}\mathbf{H}^{(1;m_1)}]\mathbf{L}^{(2)}\mathbf{H}^{(2;m_2)}\Big)_{i_2,j_2},$$

and

$$\ddot{\mathcal{B}}(h_2) := \left(\frac{\partial^2 f}{\partial \mathbf{H}^{(2;m_2)^2}}\right)_{i_2,j_2} = \Big(\mathbf{W}^{(2;m_2)}\mathbf{X}_{(1)}(\mathbf{I}_{I_3} \otimes \mathbf{H}^{(1;m_1)}\mathbf{W}^{(1;m_1)})^\top$$
$$\times (\mathbf{I}_{I_3} \otimes \mathbf{H}^{(1;m_1)}\mathbf{W}^{(1;m_1)})\mathbf{X}_{(2)}^\top \mathbf{W}^{(2;m_2)^\top}\Big)_{j_2,j_2}$$
$$+ \alpha \mathrm{Tr}[\mathbf{H}^{(1;m_1)^\top}\mathbf{L}^{(1)}\mathbf{H}^{(1;m_1)}]\ell_{i_2,i_2}^{(2)}.$$

## 7.6 COMPUTATIONAL COMPLEXITY

The purpose of this section is to evaluate the computational complexity of the suggested MSLFS method to offer a clear insight into its efficiency. Assessing the time complexity of each phase in Algorithm 1 allows for the calculation of the total computational expense. This evaluation will also emphasize the performance and scalability of the algorithm when managing large-scale applications. Initially, it is crucial to emphasize that for specific matrices $\mathbf{A} \in \mathbb{R}^{m \times n}$, $\mathbf{B} \in \mathbb{R}^{n \times r}$, $\mathbf{C} \in \mathbb{R}^{m \times nk}$, and $\mathbf{E} \in \mathbb{R}^{n \times n}$, the calculations for $\mathbf{AB}$ and $\mathbf{C}(\mathbf{I}_k \otimes \mathbf{E})$ consist of $2mnr - mr$ and $2mn^2k - mnk$ arithmetic operations, respectively. It is important to note that the calculation $(\mathbf{I}_k \otimes \mathbf{E})$ requires no arithmetic operations since it is a diagonal matrix. Accordingly, the computational cost of updating the matrices $\mathbf{W}^{(1;m_1)}$, $\mathbf{H}^{(1;m_1)}$, $\mathbf{W}^{(2;m_2)}$ and $\mathbf{H}^{(2;m_2)}$ appears as follows:

1. The computational expense of updating the matrix $\mathbf{W}^{(1;m_1)}$ is Total flops$(\mathbf{W}^{(1;m_1)})$
$$\approx 6\,m_1 I_1 I_2 I_3 + 4\,I_1 I_2^2 I_3 + 2\,m_1 I_2^2 I_3 + 6\,m_2^2 I_2 + 6\,m_2 I_2^2 + 8\,m_1^2 I_1 + 2\,m_1 I_1 + 2\,m_2^3$$
$$= \mathcal{O}(m_1 I_1 I_2 I_3 + I_1 I_2^2 I_3) = \mathcal{O}(I_1 I_2 I_3 \max\{m_1, I_2\}).$$

2. The computational expense of updating the matrix $\mathbf{H}^{(1;m_1)}$ is Total flops$(\mathbf{H}^{(1;m_1)})$
$$\approx 8\,m_1 I_1 I_2 I_3 + 6\,I_1 I_2^2 I_3 + 8\,m_2 I_2^2 + 4\,m_1 I_1^2 + 2\,m_2^2 I_2 + 4\,m_1 I_1^2$$
$$= \mathcal{O}(m_1 I_1 I_2 I_3 + I_1 I_2^2 I_3) = \mathcal{O}(I_1 I_2 I_3 \max\{m_1, I_2\}).$$

3. The computational expense of updating the matrix $\mathbf{W}^{(2;m_2)}$ is Total flops$(\mathbf{W}^{(2;m_2)})$
$$\approx 6\,m_2 I_1 I_2 I_3 + 4\,I_2 I_1^2 I_3 + 2\,m_2 I_1^2 I_3 + 6\,m_1^2 I_1 + 6\,m_1 I_1^2 + 8\,m_2^2 I_2 + 2\,m_2 I_2 + 2\,m_1^3$$
$$= \mathcal{O}(m_2 I_1 I_2 I_3 + I_1^2 I_2 I_3) = \mathcal{O}(I_1 I_2 I_3 \max\{I_1, m_2\}).$$

4. The computational expense of updating the matrix $\mathbf{H}^{(2;m_2)}$ is Total flops$(\mathbf{H}^{(2;m_2)})$
$$\approx 8\,m_2 I_1 I_2 I_3 + 6\,I_2 I_1^2 I_3 + 8\,m_1 I_1^2 + 4\,m_2 I_2^2 + 2\,m_1^2 I_1 + 4\,m_2 I_2^2$$
$$= \mathcal{O}(m_2 I_1 I_2 I_3 + I_2 I_1^2 I_3) = \mathcal{O}(I_1 I_2 I_3 \max\{m_2, I_1\}).$$

To sum up, the computational expense of a single iteration of Algorithm 1 can be determined as follows:

$$\text{Overall Total flops} = \mathcal{O}\bigg(I_1 I_2 I_3\big(\max\{m_1, I_2\} + \max\{m_2, I_1\}\big)\bigg).$$

## 7.7 MSLFS Updating Rules for a Real-valued Tensor Data

To derive multiplicative updating rules when $\mathcal{X}$ may be signed but all learned variables remain non-negative, we follow the same derivative computations as before and then apply elementwise positive/negative splitting to the matrix expressions that involve $\mathbf{X}_{(n)}$, where $n \in \{1, 2\}$. For any real matrix $\mathbf{M}$ we denote $\mathbf{M}_+ := \max(\mathbf{M}, 0)$ and $\mathbf{M}_- := \max(-\mathbf{M}, 0)$ (elementwise), so $\mathbf{M} = \mathbf{M}_+ - \mathbf{M}_-$. The multiplicative update rule for a non-negative variable $\mathbf{Z}$ with gradient decomposed as $\nabla_{\mathbf{Z}}\mathcal{F} = \mathbf{G}^+ - \mathbf{G}^-$ (with $\mathbf{G}^{\pm} \geq 0$) is $\mathbf{Z} \leftarrow \mathbf{Z} \odot \mathbf{G}^- \oslash \mathbf{G}^+$. The gradients are fully developed in the previous section. Using the elementwise positive/negative splitting described above, the multiplicative updates (for non-negative factors while $\mathcal{X}$ may be signed) are:

$$
\begin{aligned}
\mathbf{W}^{(1;m_1)} = \mathbf{W}^{(1;m_1)} \odot\; & [(\mathbf{H}^{(1;m_1)^\top}\mathbf{X}_{(1)}\mathbf{I}_{I_3} \otimes (\mathbf{H}^{(2;m_2)}\mathbf{W}^{(2;m_2)})\mathbf{X}_{(1)}^\top)_- \\
& + \gamma\mathrm{Tr}[\mathbf{W}^{(2;m_2)}\mathbf{W}^{(2;m_2)^\top}]\mathbf{W}^{(1;m_1)}] \\
& \oslash [(\mathbf{H}^{(1;m_1)^\top}\mathbf{H}^{(1;m_1)}\mathbf{W}^{(1;m_1)}\mathbf{X}_{(1)}(\mathbf{I}_{I_3} \otimes \mathbf{H}^{(2;m_2)}\mathbf{W}^{(2;m_2)})^\top(\mathbf{I}_{I_3} \otimes \mathbf{H}^{(2;m_2)}\mathbf{W}^{(2;m_2)})\mathbf{X}_{(1)}^\top)_+ \\
& + \beta\mathrm{Tr}[\mathbf{W}^{(2;m_2)}\mathbf{U}^{(2)}\mathbf{W}^{(2;m_2)^\top}]\mathbf{W}^{(1;m_1)}\mathbf{U}^{(1)} \\
& + \gamma\mathrm{Tr}[\mathbf{W}^{(2;m_2)}\mathbf{W}^{(2;m_2)^\top}\mathbf{W}^{(2;m_2)}\mathbf{W}^{(2;m_2)^\top}]\mathbf{W}^{(1;m_1)}\mathbf{W}^{(1;m_1)^\top}\mathbf{W}^{(1;m_1)}],
\end{aligned}
$$

$$
\begin{aligned}
\mathbf{H}^{(1;m_1)} = \mathbf{H}^{(1;m_1)} \odot\; & [(\mathbf{X}_{(1)}(\mathbf{I}_{I_3} \otimes \mathbf{H}^{(2;m_2)}\mathbf{W}^{(2;m_2)})\mathbf{X}_{(1)}^\top\mathbf{W}^{(1;m_1)^\top})_- \\
& + \alpha\mathrm{Tr}[\mathbf{H}^{(2;m_2)^\top}\mathbf{L}^{(2)}\mathbf{H}^{(2;m_2)}]\mathbf{A}^{(1)}\mathbf{H}_1] \\
& \oslash [(\mathbf{H}^{(1;m_1)}\mathbf{W}^{(1;m_1)}\mathbf{X}_{(1)}(\mathbf{I}_{I_3} \otimes \mathbf{H}^{(2;m_2)}\mathbf{W}^{(2;m_2)})^\top(\mathbf{I}_{I_3} \otimes \mathbf{H}^{(2;m_2)}\mathbf{W}^{(2;m_2)})\mathbf{X}_{(1)}^\top\mathbf{W}^{(2;m_2)^\top})_+ \\
& + \alpha\mathrm{Tr}[\mathbf{H}^{(2;m_2)^\top}\mathbf{L}^{(2)}\mathbf{H}^{(2;m_2)}]\mathbf{B}^{(1)}\mathbf{H}^{(1;m_1)}],
\end{aligned}
$$

$$
\begin{aligned}
\mathbf{W}^{(2;m_2)} = \mathbf{W}^{(2;m_2)} \odot\; & [(\mathbf{H}^{(2;m_2)^\top}\mathbf{X}_{(2)}(\mathbf{I}_{I_3} \otimes \mathbf{H}^{(1;m_1)}\mathbf{W}^{(1;m_1)})\mathbf{X}_{(2)}^\top)_- \\
& + \gamma\mathrm{Tr}[\mathbf{W}^{(1;m_1)}\mathbf{W}^{(1;m_1)^\top}]\mathbf{W}^{(2;m_2)}] \\
& \oslash [(\mathbf{H}^{(2;m_2)^\top}\mathbf{H}^{(2;m_2)}\mathbf{W}^{(2;m_2)}\mathbf{X}_{(2)}(\mathbf{I}_{I_3} \otimes \mathbf{H}^{(1;m_1)}\mathbf{W}^{(1;m_1)})^\top(\mathbf{I}_{I_3} \otimes \mathbf{H}^{(1;m_1)}\mathbf{W}^{(1;m_1)})\mathbf{X}_{(2)}^\top)_+ \\
& + \beta\mathrm{Tr}[\mathbf{W}^{(1;m_1)}\mathbf{U}^{(1)}\mathbf{W}^{(1;m_1)^\top}]\mathbf{W}^{(2;m_2)}\mathbf{U}^{(2)} \\
& + \gamma\mathrm{Tr}[\mathbf{W}^{(1;m_1)}\mathbf{W}^{(1;m_1)^\top}\mathbf{W}^{(1;m_1)}\mathbf{W}^{(1;m_1)^\top}]\mathbf{W}^{(2;m_2)}\mathbf{W}^{(2;m_2)^\top}\mathbf{W}^{(2;m_2)}],
\end{aligned}
$$

$$
\begin{aligned}
\mathbf{H}^{(2;m_2)} = \mathbf{H}^{(2;m_2)} \odot\; & [(\mathbf{X}_{(2)}(\mathbf{I}_{I_3} \otimes \mathbf{H}^{(1;m_1)}\mathbf{W}^{(1;m_1)})\mathbf{X}_{(2)}^\top\mathbf{W}^{(2;m_2)^\top})_- \\
& + \alpha\mathrm{Tr}[\mathbf{H}^{(1;m_1)^\top}\mathbf{L}^{(1)}\mathbf{H}^{(1;m_1)}]\mathbf{A}^{(2)}\mathbf{H}^{(2;m_2)}] \\
& \oslash [(\mathbf{H}^{(2;m_2)}\mathbf{W}^{(2;m_2)}\mathbf{X}_{(2)}(\mathbf{I}_{I_3} \otimes \mathbf{H}^{(1;m_1)}\mathbf{W}^{(;m_1)})^\top(\mathbf{I}_{I_3} \otimes \mathbf{H}^{(1;m_1)}\mathbf{W}^{(1;m_1)})\mathbf{X}_{(2)}^\top\mathbf{W}^{(2;m_2)^\top})_+ \\
& + \alpha\mathrm{Tr}[\mathbf{H}^{(1;m_1)^\top}\mathbf{L}^{(1)}\mathbf{H}^{(1;m_1)}]\mathbf{B}^{(2)}\mathbf{H}^{(2;m_2)}].
\end{aligned}
$$

# 8 ADDITIONAL EXPERIMENTAL RESULTS

## 8.1 Datasets

Table 6 summarizes the key statistics of the eight benchmark datasets used in our experiments, including the number of samples, feature dimensions, number of classes, and the range of selected features. These datasets together provide a comprehensive and diverse evaluation environment for assessing the proposed method across different domains, sample sizes, and feature complexities. **COIL20** (Nene et al., 1996), **ORL** (Cai et al., 2010), and **UMIST** (Graham & Allinson, 1998) are classical image recognition benchmarks encompassing objects and human faces. COIL20 contains 20 objects imaged from multiple viewpoints, effectively testing robustness to pose variation. ORL consists of 40 subjects captured under relatively controlled conditions, whereas UMIST presents 20 subjects with more pronounced pose and illumination variations, creating a more challenging low-sample scenario. **Pixraw10P** and **Orlraws10P** (Li et al., 2017) are high-dimensional raw image subsets with limited samples, designed to evaluate the scalability of feature selection in situations where the number of features far exceeds the number of observations. Moving beyond traditional

object and face recognition, **FashionMNIST** (Xiao et al., 2017) serves as a modern drop-in replacement for the classic MNIST handwritten digit dataset, sharing the same grayscale $28 \times 28$ format but comprising clothing images with richer visual variability and finer inter-class distinctions, thus providing a more challenging benchmark while remaining compatible with MNIST's experimental protocols. In the biomedical domain, **BreastMNIST** and **OrganSMNIST** (Yang et al., 2021) focus on medical imaging tasks, with BreastMNIST providing a binary classification task based on breast ultrasound scans and OrganSMNIST involving multi-class organ recognition from MRI slices, thereby testing the applicability of the proposed approach to real-world medical scenarios. Collectively, these datasets span a wide range of sample sizes (from 100 to 1,440), feature dimensions (from $23 \times 28$ to $100 \times 100$), and class cardinalities (from 2 to 40), ensuring that the empirical evaluation thoroughly examines the method's robustness, scalability, and generalization ability across diverse, small-sample, high-dimensional, and domain-shifted settings.

Table 6: Detailed Statistics of the Eight Datasets.

| Dataset | # of Samples | Feature Size | # of Classes | Range of Selected Features |
|---|---|---|---|---|
| COIL20 | 1,440 | $32 \times 32$ | 20 | [50, 100, ..., 300] |
| ORL | 400 | $32 \times 32$ | 40 | [50, 100, ..., 300] |
| UMIST | 575 | $23 \times 28$ | 20 | [50, 100, ..., 300] |
| Pixraw10P | 100 | $100 \times 100$ | 10 | [50, 100, ..., 300] |
| Orlraws10P | 100 | $92 \times 112$ | 10 | [50, 100, ..., 300] |
| FashionMNIST | 1,000 | $28 \times 28$ | 10 | [50, 100, ..., 300] |
| BreastMNIST | 546 | $28 \times 28$ | 2 | [50, 100, ..., 300] |
| OrganSMNIST | 500 | $28 \times 28$ | 11 | [50, 100, ..., 300] |
| OrganCMNIST | 20,000 | $224 \times 224$ | 11 | [50, 100, ..., 300] |
| PneumoniaMNIST | 5,800 | $224 \times 224$ | 2 | [50, 100, ..., 300] |
| Kinetic Fluorescence | 10,000 | $64 \times 12 \times 10$ | 7 | [50, 100, ..., 300] |
| COVID-19 Systems Serology | 5,200 | $20 \times 35 \times 74$ | 5 | [50, 100, ..., 300] |

## 8.2 COMPARISON MODELS

This section summarizes the feature selection methods used for comparison, highlighting the core mechanism and strategy of each model to identify informative features while preserving relevant data structures.

- **LS** (He et al., 2005a): Assesses each feature individually based on how well it can maintain the local geometric structure of the data.

- **UDFS** (Yang et al., 2011): Selects the most informative features by performing both $\ell_{2,1}$ norm-based feature selection and local discriminative analysis at the same time.

- **ILFS** (Roffo et al., 2017): A probabilistic feature selection method that ranks features by considering all possible subsets while avoiding combinatorial complexity.

- **GRLTR** (Su et al., 2018): Combines low-rank tensor representation with local geometry preservation and $\ell_{2,1}$ norm-based feature selection.

- **CAE** (Balın et al., 2019): An end-to-end global feature selection approach that simultaneously trains a neural network to reconstruct the input data while selecting a representative subset of features.

- **FSPCA** (Tian et al., 2020): Simultaneously conducts feature selection and PCA by directly estimating the leading eigenvectors under row-sparsity constraints.

- **CPUFS** (Chen et al., 2023): Integrates a tensor-based linear classifier with graph-regularized non-negative CP decomposition and pseudo-label regression.

- **SPCAFS** (Li et al., 2023): Applies a $\ell_{2,p}$-norm sparsity regularization to the PCA projection matrix for feature selection.

- **GRSSLFS** (Tiwari et al., 2024): Selects high-variance basis features and integrates self-representation, subspace learning, and manifold regularization to enhance feature selection.

- **SPDFS** (Dong et al., 2025): Performs discriminative feature selection via $ell_{2,0}$-norm constrained sparse projection, combining fuzzy membership learning with globally and iteratively optimized projection strategies.

## 8.3 Runtime Efficiency Evaluation

To assess how efficiently each method runs, we compare the per-iteration CPU time on datasets with more than 5,000 samples (Table 7). Across the board, MSLFS is the fastest method, noticeably outperforming both the matrix-based and tensor-based baselines. On the COVID-19 Systems Serology dataset, for example, MSLFS completes an iteration in 31.3 seconds, faster than ILFS and FSPCA, and well ahead of more complex deep or graph-regularized models like SAE, NNSE, and GRSSLFS. The gap grows larger on the much bigger Kinetic Fluorescence dataset: MSLFS is the only approach that stays under 130 seconds per iteration, while several baselines exceed 180 seconds or even fail due to memory limitations. This speed comes from MSLFS's mode-wise slice selection and separable reconstruction strategy, which avoid heavy global computations and reduce unnecessary redundancy, leading to linear complexity with respect to both the number of samples and the mode-wise feature dimensions. Overall, the results show that MSLFS is consistently the most computationally efficient method among all those we tested.

Table 7: Per-iteration CPU time comparison across methods on datasets with more than 5,000 samples. Here, "OM" indicates out-of-memory errors.

| Methods | COVID-19 Systems Serology | PneumoniaMNIST | Kinetic Fluorescence | OrganCMNIST |
|---|---|---|---|---|
| UDFS | 101.2 | 156.6 | 385.3 | OM |
| SAE | 72.1 | 106.7 | 211.3 | 389.4 |
| ILFS | 46.5 | 64.5 | 144.2 | 237.8 |
| GRLTR | 161.9 | 257.6 | OM | OM |
| CAE | 66.3 | 87.7 | 143.9 | 256.7 |
| FSPCA | 45.4 | 78.9 | 156.2 | 281.5 |
| CPUFS | 56.7 | 89.5 | 154.6 | 303.1 |
| SPCAFS | 74.4 | 103.6 | 164.2 | 266.4 |
| NNSE | 97.3 | 142.2 | 186.7 | 314.8 |
| GRSSLFS | 125.6 | 184.4 | 343.1 | OM |
| SDAE | 86.7 | 102.5 | 193.4 | 299.3 |
| SPDFS | 89.7 | 112.4 | 232.1 | 356.2 |
| **MSLFS** | **31.3** | **57.2** | **124.5** | **232.4** |

## 8.4 Feature Visualization Across Models

Figure 5 shows how each method selects features on the Pixraw10P dataset by marking the chosen pixels on a sample face image. The three rows correspond to different feature numbers (100, 200, and 300), and the columns compare a tensor-based approach (CPUFS), two autoencoder models (NNSE and SDAE), a matrix-based baseline (SPDFS), and our proposed method, MSLFS. Looking across the figure, the models behave quite differently. CPUFS tends to pick scattered but meaningful points around the eyes, nose, and mouth, suggesting that it makes reasonable use of the tensor structure. The autoencoder methods also concentrate on facial regions but in a denser and less organized way, which hints that while nonlinear encoders can detect useful signals, they do not enforce spatial or mode-wise consistency. The matrix-based SPDFS shows the least structure, spreading its selections almost uniformly across the image, a typical outcome when flattening destroys the original tensor geometry. In contrast, MSLFS produces the most coherent and visually interpretable selections: its grid-like patterns remain stable as the number of selected features increases, reflecting its slice-driven multilinear design. This leads MSLFS to preserve the underlying tensor structure of the image far better than the autoencoder and matrix baselines that use vectorized data.

## 8.5 t-SNE Visualization on Additional Datasets

Figure 6 compares the t-SNE embeddings obtained from features selected by the competing methods under three feature budgets. Across all settings, the vectorized approaches, whether linear (SPDFS) or nonlinear (SDAE, NNSE), produce clusters that are only partially separated and often exhibit substantial overlap. SPDFS displays the weakest structure, with diffuse and intermixed clusters, reflecting its inability to model relationships that span multiple modes once the tensor is flattened. The autoencoder methods fare slightly better: their nonlinear encoders extract useful global patterns, but the resulting clusters remain elongated or entangled, indicating that mode interactions are not consistently preserved even when deep architectures are used. These limitations arise from the fact that

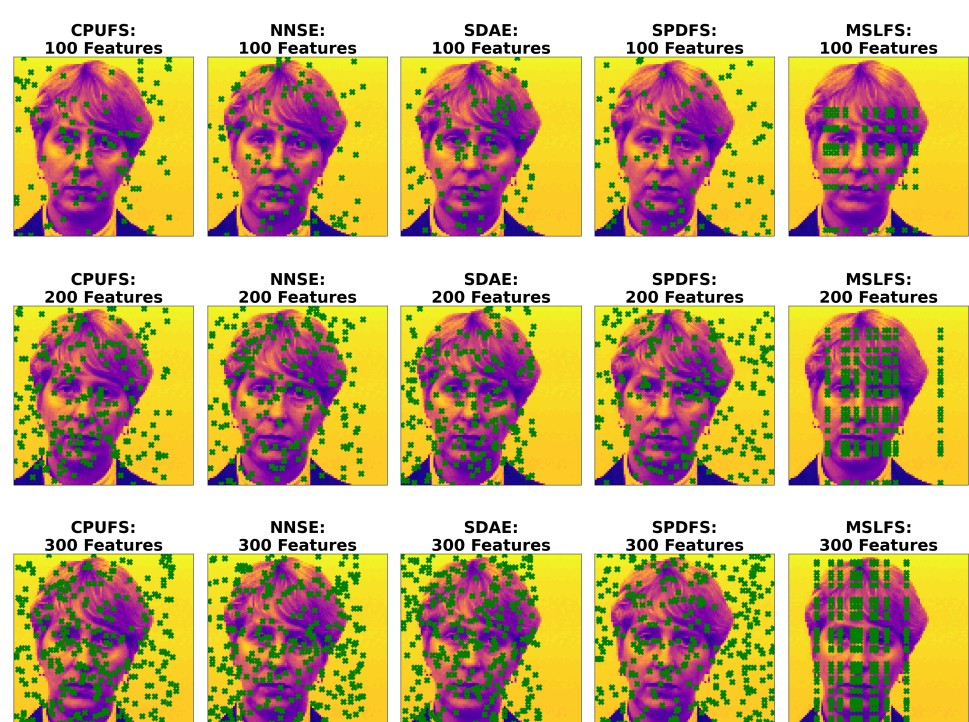

Figure 5: Comparison of selected feature visualizations across five models on Pixraw10P dataset.

all vectorized models process the data as a long feature vector, discarding the inherent multi-way structure that governs how information is distributed across modes. As a result, the features they select do not fully reflect the underlying tensor manifold, and therefore fail to produce cleanly separated embeddings. In contrast, MSLFS consistently yields compact, well-separated clusters across all feature budgets. By performing slice-based selection within a multilinear framework, MSLFS explicitly preserves inter-mode dependencies and respects the geometry of the tensor space. This allows it to isolate features that are genuinely discriminative on the underlying manifold, resulting in substantially more coherent and class-aligned t-SNE embeddings.

## 8.6 Customizing Feature Selection via Mode Combinations

To further evaluate the flexibility of MSLFS in distributing features across different tensor modes, we conducted an experiment on the Pixraw10P dataset by fixing the total number of selected features to 300 while varying the distribution of mode-1 and mode-2 slices. As illustrated in Figure 7, MSLFS can generate multiple valid configurations, such as $100 \times 3$, $50 \times 6$, or $10 \times 30$, each corresponding to 300 intersection fibers. Across these different allocations, the selected slices capture meaningful vertical and horizontal structures. However, the results indicate that balanced selections across the two modes (e.g., $15 \times 20$ or $12 \times 25$) better preserve the overall inherent structure spanned by modes 1 and 2, while extreme allocations to a single mode tend to lose complementary information. This highlights that although MSLFS is flexible in how features are distributed, balanced configurations most effectively maintain both local and global structures.

## 8.7 Clustering on Selected Features

The experimental results with varying numbers of selected features further highlight the effectiveness of MSLFS. As shown in Figure 8, MSLFS is compared against 10 state-of-the-art models across eight benchmark datasets, where the performance curves illustrate both the absolute clustering accuracy and the stability of each method under different feature dimensions. Overall, MSLFS consistently outperforms competing approaches, achieving the best or near-best results in terms of

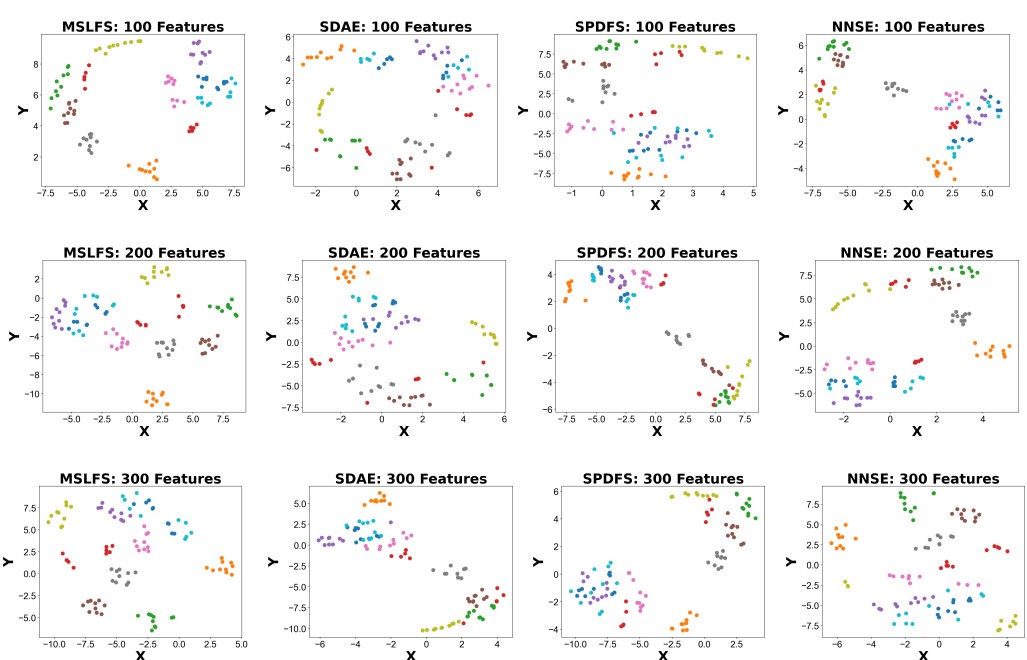

Figure 6: Comparison of t-SNE embeddings derived from features selected by comparative models.

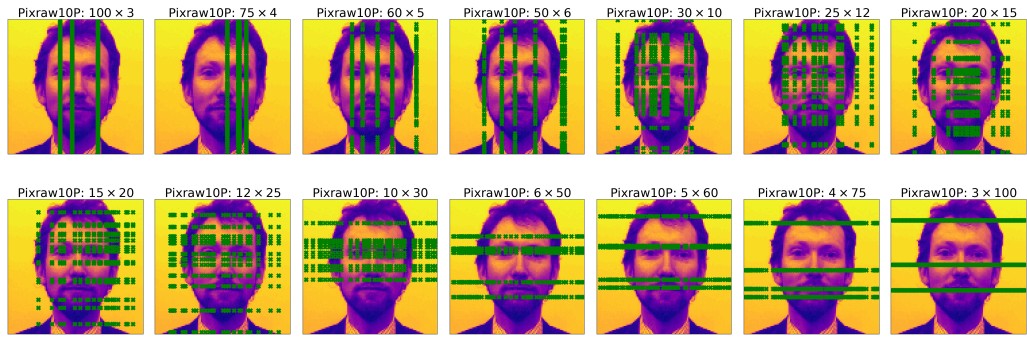

Figure 7: Visualization of mode-wise feature selection flexibility on Pixraw10P

ACC and NMI across nearly all datasets. The improvements are especially notable on COIL20, ORL, and UMIST where MSLFS maintains clear superiority across varying feature subsets. Even on more challenging datasets such as BreastMNIST and OrganSMNIST, where existing methods often suffer from instability, MSLFS achieves significant margins, underscoring its robustness to data variability and imbalance. Furthermore, unlike other models that exhibit sharp fluctuations as the number of selected features changes, MSLFS demonstrates smooth and reliable performance trends, consistently producing discriminative feature subsets. This stability is largely attributed to its slice-based selection mechanism and higher-order graph regularization, which together preserve informative structures while effectively suppressing redundancy.

## 8.8 SENSITIVITY ANALYSIS

To further investigate the influence of the regularization parameters $\alpha$ and $\beta$ on the clustering performance of MSLFS, a sensitivity analysis is conducted. Figure 9 presents the heatmaps of NMI and ACC values across six datasets, including UMIST, Pixraw10P, Orlraws10P, ORL, OrganSM-NIST, and FashionMNIST. From Figure 9, it can be observed that the proposed method exhibits relatively stable behavior across a wide range of parameter values, though some dataset-specific

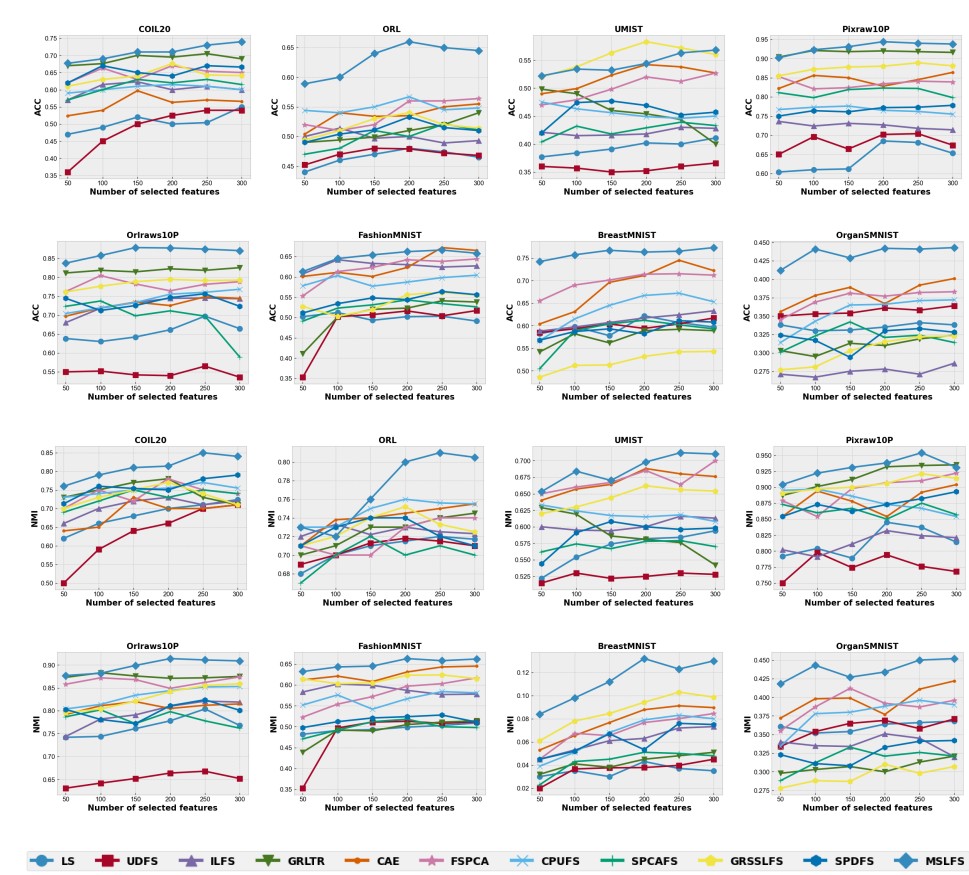

Figure 8: ACC and NMI curves of different feature selection methods on the eight datasets

trends emerge. For the UMIST dataset, both NMI and ACC remain stable with small fluctuations, and the best results are achieved when $\alpha$ lies within $\{10^1, 10^2, 10^3\}$ and $\beta$ takes values around $\{10^1, 10^2, 10^3\}$. For the Pixraw10P dataset, MSLFS shows more sensitivity to $\beta$, with superior performance observed when $\alpha \in \{10^{-1}, 10^0, 10^1\}$ and $\beta$ is set within $\{10^{-3}, 10^{-2}, 10^{-1}\}$. In the Orlraws10P dataset, MSLFS achieves consistently high NMI and ACC values, with optimal performance emerging when $\alpha \in \{10^{-3}, 10^{-2}, 10^3\}$ and $\beta \in \{10^{-2}, 10^{-1}, 10^0, 10^4\}$.

For the ORL dataset, the clustering performance is relatively insensitive to variations in $\beta$, while the most favorable results occur when $\alpha$ is chosen from $\{10^0, 10^1\}$. In the case of the OrganSMNIST dataset, both NMI and ACC show more noticeable fluctuations, but relatively better performance is achieved when $\alpha \in \{10^{-4}, 10^{-3}, 10^{-2}, 10^4\}$ and $\beta$ lies between $\{10^{-1}, 10^0, 10^1\}$. Finally, for the FashionMNIST dataset, the results indicate higher stability across parameter values, with the best performance obtained for $\alpha \in \{10^{-2}, 10^{-1}, 10^2\}$ and $\beta \in \{10^{-1}, 10^0, 10^4\}$.

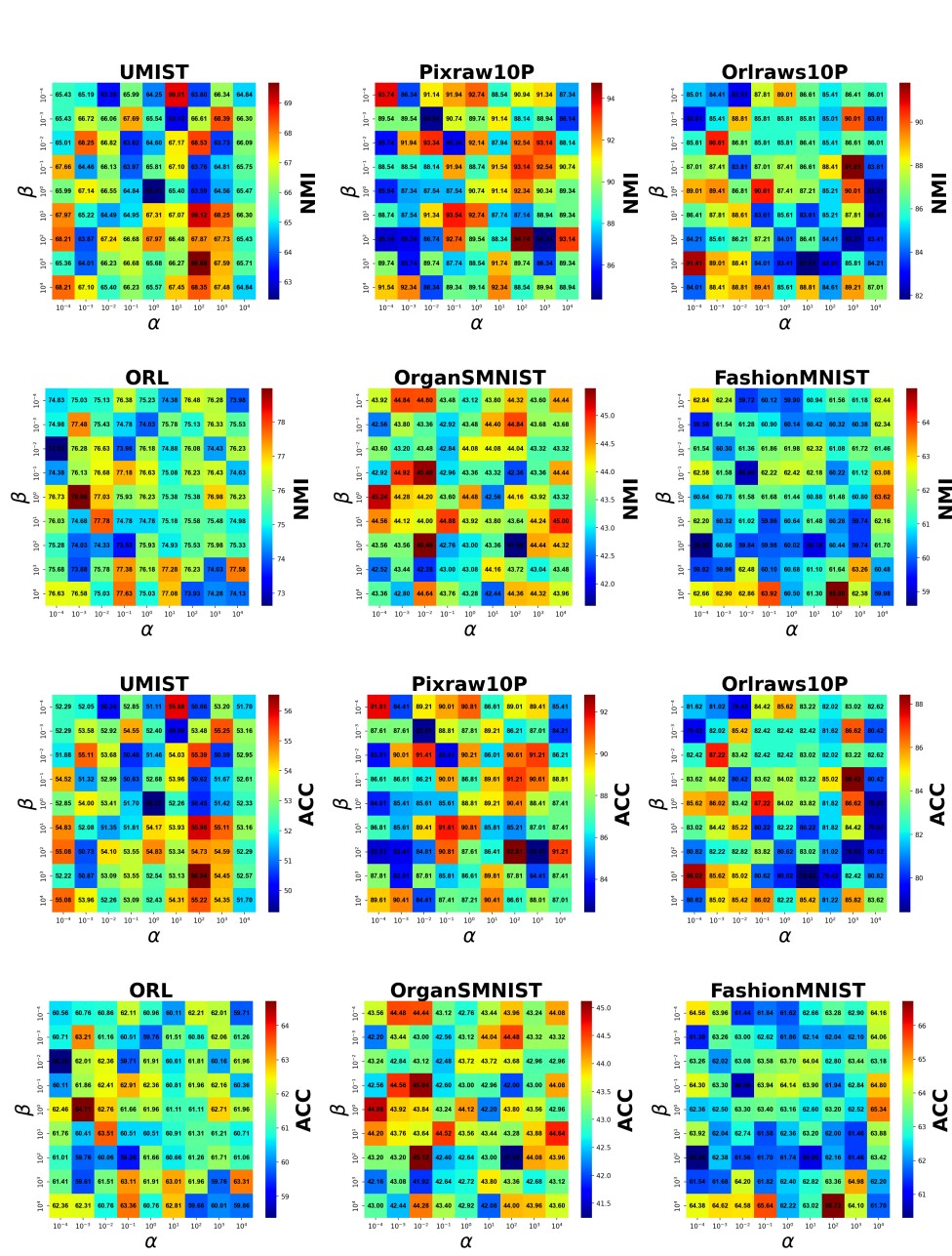

Figure 9: A comparison of the NMI and ACC scores obtained by MSLFS with different values of the parameters $\alpha$, and $\beta$ on six datasets.

