# OpenReview forum: "Multi-Linear Subspace Distance: A New Criterion for Tensor Feature Selection"
_ICLR.cc/2026/Conference — Submitted to ICLR 2026_

### Official Review · Reviewer_ADWM · 2025-10-23

**Soundness:** 2
**Presentation:** 3
**Contribution:** 2
**Rating:** 2
**Confidence:** 4

**Summary:**

This paper introduces Multi-linear Subspace Learning Feature Selection (MSLFS), a novel tensor-based feature selection framework that operates distributively across tensor modes. It proposes a multi-linear subspace distance to preserve global tensor structure and incorporates joint sparsity and higher-order graph regularization to reduce redundancy and maintain local geometry. Extensive experiments on image and biomedical datasets demonstrate that MSLFS consistently outperforms state-of-the-art feature selection methods in clustering tasks. The method also offers computational efficiency and flexibility in selecting features through slice intersections along different modes.

**Strengths:**

1. The writing and presentation are relative smooth, the readers can easily follow.
2. The mathematic prove is complete, ensuring its theoretical solidness.

**Weaknesses:**

1. This is an incremental work. In the introduction, there is no in-depth analysis of existing methods and necessary experimental verification to prove the defects of the current methods.
2. Sparse regularization and graph regularization have been widely studied and applied in a large number of works, and no novelties can be found.
3. The author's proposed algorithm can improve the running efficiency, but no comparison of running time can be seen in the experiment.
4. In the experiment, the datasets used are old and small. In addition, I noticed that the clustering effect after feature selection is not very good. As far as I know, some clustering results based on deep networks are already very excellent. Therefore, the significance of feature selection has to be questioned.

**Questions:**

Please refer to the weaknesses.

---

> ### Author Response · Authors · 2025-11-17
>
> > **Q1. This is an incremental work. In the introduction, there is no in-depth analysis of existing methods and necessary experimental verification to prove the defects of the current methods.**
>
> We thank the reviewer for this helpful suggestion. Following the comment, we have significantly expanded the related work section to include a more comprehensive discussion of recent advances in unsupervised feature selection (2024–2025). Specifically, we now include six notable linear and nonlinear models from the matrix-based unsupervised feature selection domain. In contrast, only two tensor-based frameworks have been proposed recently, despite their substantial potential for handling tensor data. This gap motivated the development of our method, **MSLFS**, which leverages the full multi-way structure of tensor data without resorting to vectorization, thereby preserving critical cross-mode dependencies. Below, we summarize the most recent advancements:
>
> **Matrix-Based Unsupervised Feature Selection**.
> Unsupervised feature selection in vectorized data has been actively researched. Key models include:
>
> * **ESUFS** [1]: A method that addresses key challenges in structured-graph approaches by introducing a discrete structured-graph model and a data discrepancy learning module to improve graph structure and emphasize naturally discriminative features without labels.
>
> * **UFS-CGL** [2]: A unified framework that overcomes earlier graph-guided methods' limitations by retaining class-specific information and using contrastive learning to capture richer discriminative relationships, alongside an $\ell_{1,2}$-regularized projection matrix for feature selection.
>
> * **FOG-R** [3]: A flexible framework that uses graph learning to replace rigid linear projections in spectral-based feature selection, addressing the limitations of traditional low-dimensional projections in capturing the true data manifold.
>
> * **MRMGRFS** [4]: A model that balances relevance and redundancy in feature selection. It combines spectral clustering and Jensen–Shannon divergence to refine relevance scores and minimize global redundancy.
>
> * **NNSE** [5]: This model replaces traditional linear projection with neural networks, capturing richer feature–label relationships. It integrates adaptive graph regularization to preserve local structure and enhance representational power.
>
> * **SDAE** [6]: A deep autoencoder-based framework that jointly performs feature selection and reconstruction. It identifies the most informative features to reduce dimensionality while preserving the global structure.
>
> **Tensor-Based Unsupervised Feature Selection**.
> Tensor-based methods have emerged to address the limitations of vectorized feature selection for multi-dimensional data. Notable recent models include:
>
> * **CPUFS**: This method preserves the multi-dimensional structure of tensor data through graph-regularized nonnegative CP decomposition, combined with tensor-oriented pseudo-label regression and feature selection.
>
> * **GRLTR**: A low-rank tensor representation framework that integrates graph regularization to preserve both global structure and local geometry, operating directly on tensor data to maintain its inherent multi-dimensional properties.
>
> [1]: Pei Huang, Zhaoming Kong, Limin Wang, Xuming Han, and Xiaowei Yang. Efficient and Stable
> Unsupervised Feature Selection Based on Novel Structured Graph and Data Discrepancy Learn-
> ing. IEEE Transactions on Neural Networks and Learning Systems, 36(4):6229–6243, 2025.
>
> [2]: Qian Zhou, Qianqian Wang, Quanxue Gao, Ming Yang, and Xinbo Gao. Unsupervised Discrimina-
> tive Feature Selection via Contrastive Graph Learning. IEEE Transactions on Image Processing,
> 33:972–986, 2024.
>
> [3]: Hong Chen, Feiping Nie, Rong Wang, and Xuelong Li. Unsupervised Feature Selection with Flexi-
> ble Optimal Graph. IEEE Transactions on Neural Networks and Learning Systems, 35(2):2014–
> 2027, 2024.
>
> [4]: Xianyu Zuo, Wenbo Zhang, Xiangyu Wang, Lanxue Dang, Baojun Qiao, and Yadi Wang. Unsu-
> pervised Feature Selection via Maximum Relevance and Minimum Global Redundancy. Pattern
> Recognition, 164:111483, 2025.
>
> [5]: Mengbo You, Aihong Yuan, Dongjian He, and Xuelong Li. Unsupervised Feature Selection via
> Neural Networks and Self-Expression with Adaptive Graph Constraint. Pattern Recognition,
> 135:109173, 2023b.
>
> [6]: Wael Hassanieh and Abdallah Chehade. Selective Deep Autoencoder for Unsupervised Feature
> Selection. In Proceedings of the Thirty-Eighth AAAI Conference on Artificial Intelligence and
> Thirty-Sixth Conference on Innovative Applications of Artificial Intelligence and Fourteenth Sym-
> posium on Educational Advances in Artificial Intelligence. AAAI Press, 2024b.

---

> ### Author Response · Authors · 2025-11-17
>
> > **The continuation of the answer to the Q1**
>
> We now outline the theoretical and practical limitations of existing approaches and demonstrate these limitations empirically. The first set of limitations pertains to matrix-based methods. To illustrate them, we consider a tensor dataset of size $\mathbb{R}^{I_1 \times I_2 \times I_3}$, where $I_3$ denotes the number of samples and $I_1 \times I_2$ constitutes the multi-dimensional feature space; however, these limitations naturally generalize to tensors of arbitrary order $N$. We would also like to note that the generalization of our framework to arbitrary $N$-mode tensor data is discussed in detail in our response to the first reviewer’s second comment, and the effectiveness of this generalized framework is demonstrated experimentally in our responses to the reviewer’s third and fourth comments.
>
> Matrix-based methods begin by vectorizing each sample into an $I_1I_2$-dimensional vector and stacking these vectors as rows of a data matrix, effectively operating on the mode-3 unfolding of size $\mathbb{R}^{I_1I_2 \times I_3}$. This preprocessing step constitutes a major limitation, as it destroys the intrinsic multi-way structure of tensor data, including cross-mode dependencies, thereby preventing both linear and neural network–based matrix methods from capturing the true relationships within the data. As a result, their performance on downstream tasks such as clustering is inevitably degraded. After vectorization, these methods typically learn a feature-weighting matrix of size $\mathbb{R}^{p \times I_1I_2}$ (with $p < I_1I_2$) and select the top $p$ features by computing the column-wise $\ell_2$ norms.
>
> Tensor-based approaches, in contrast, operate directly on the multi-way structure. However, existing methods such as **CPUFS** and **GRLTR**, despite effectively modeling multi-dimensional interactions, ultimately rely on the same feature-selection mechanism as matrix-based approaches: they still construct a $\mathbb{R}^{p \times I_1I_2}$ feature-weighting matrix and select features via column-wise $\ell_2$ norms. Consequently, they inherit two core limitations: (1) the computational burden of searching across the entire $I_1I_2$ feature space, and (2) the inability to exploit cross-mode dependencies during feature selection, since features are still treated in their vectorized form.
>
> Our proposed **MSLFS** method overcomes these issues through a distributed, mode-aware feature-selection strategy. Instead of selecting from all $I_1I_2$ features simultaneously, **MSLFS** identifies $m_1 \le p$ representative mode-1 slices and $m_2 \le p$ representative mode-2 slices such that $m_1 m_2 = p$. The intersections of these slices yield the mode-3 fibers that form the final set of $p$ informative features, eliminating the need to evaluate the full vectorized feature space.  The effectiveness of this selection approach is established by **Theorem 3.1**. This design not only reduces the computational complexity of **MSLFS** to scale linearly with both the number of samples and features but also preserves and leverages the inherent multi-way dependencies of tensor data during selection. These are precisely the limitations that **MSLFS** is designed to address in unsupervised feature selection.
>
> To empirically validate these limitations, we evaluate **MSLFS** against leading matrix-based, tensor-based, and deep learning methods over an extensive set of experiments. Complete experimental descriptions are included in our replies to the reviewer’s third and fourth comments.

---

> ### Author Response · Authors · 2025-11-17
>
> > **Q2. Sparse regularization and graph regularization have been widely studied and applied in a large number of works, and no novelties can be found.**
>
> We thank the reviewer for their insightful comment. In response, we agree with the reviewer that sparsity and graph regularization have been widely studied. However, the way these regularization terms are incorporated in **MSLFS** is fundamentally different from their use in existing matrix- and tensor-based methods. For clarity, we consider the common setting where the data form a 3-mode tensor $\mathcal{X} \in \mathbb{R}^{I_1 \times I_2 \times I_3}$, where $I_3$ denotes the number of samples and $I_1 \times I_2$ constitutes the multi-dimensional feature space. All arguments readily extend to tensors of arbitrary order $N$.
>
> Existing approaches first vectorize or unfold this $3$-mode tensor into a single high-dimensional feature vector of size $I_1I_2$, and only then apply sparsity or graph regularization on this flattened representation. Such a design destroys the natural multi-way structure and forces regularization to operate on a flattened space, and results in high computational cost because sparsity/graph constraints must act over all $I_1 I_2$ features jointly.
>
> In contrast, our proposed MSLFS method distributes the regularization terms across modes, acting directly on the mode-specific slice subspaces before any vectorization occurs. Concretely, for each feature mode $(n \in {1,2})$, the method selects $m_n$ representative slices, where $m_n \le I_n$, through imposing mode-wise sparsity. These slices approximate the mode-$n$ subspace $\mathcal{S}(\mathbf{X}^{(n)})$, and their intersections yield a compact set of informative mode-3 fibers (features). The $m_n$ values therefore control the number of slices retained along each feature mode and determine the granularity of feature selection in that mode.
> This distributed strategy yields several important advantages:
>
> **Preservation of multi-way structure**. By applying regularization on each mode independently, MSLFS retains the intrinsic geometry of each dimension, e.g., vertical structure (mode-1) and horizontal structure (mode-2) in images, before combining them through slice intersections. Existing vectorized methods cannot differentiate these roles; they collapse all structure into a single axis, masking cross-mode dependencies.
>
> **Computational efficiency.** Regularizing mode-wise matrices of sizes $I_n \times m_n$ greatly reduces the cost compared to regularizing a single $I_1 I_2$-dimensional matrix $(I\_1I\_2 \\times p, \\; p \\gt m_n)$ or graph $(I\_1I\_2 \\times I\_1I\_2)$. This allows MSLFS to avoid cubic or near cubic costs common in tensor and matrix methods. Furthermore, the sparsity and graph regularizers admit mode-wise factorizations, a property that is mathematically impossible when applying such regularizers to a vectorized representation. To mathematically illustrate this factorization across modes, consider the mode-wise sparsity and graph regularization terms in **MSLFS**:
> > \\begin{align}
> > \\|(\\mathbf{W}^{(2;m\_2)}\\otimes \\mathbf{W}^{(1;m\_1)})^\\top\\|\_{2,1}\& = \\|\\mathbf{W}^{{(2;m\_2)}^\\top}\\|\_{2,1}\\|\\mathbf{W}^{{(1;m\_1)}^\\top}\|\_{2,1} \\notag \\\\
> > \&= \\mathrm{Tr} (\\mathbf{W}^{(2;m\_2)}\\mathbf{U}^{(2)}\\mathbf{W}^{{(2;m\_2)}^\\top}) \\;
> > \\mathrm{Tr}(\\mathbf{W}^{(1;m\_1)}\\mathbf{U}^{(1)}\\mathbf{W}^{{(1;m\_1)}^\\top}).
> > \\end{align}
>
> > \\begin{align}
> \&\\mathrm{Tr}\\big[(\\mathbf{H}^{(2;m\_2)}\\otimes \\mathbf{H}^{(1;m\_1)})^\\top(\\mathbf{L}^{(2)}\\otimes \\mathbf{L}^{(1)})(\\mathbf{H}^{(2;m\_2)}\\otimes \\mathbf{H}^{(1;m\_1)})\\big] = \\notag \\\\
> \&\\mathrm{Tr}(\\mathbf{H}^{{(2;m\_2)}^\\top}\\mathbf{L}^{(2)}\\mathbf{H}^{(2;m\_2)})\\;\\mathrm{Tr}(\\mathbf{H}^{{(1;m\_1)}^\\top}\\mathbf{L}^{(1)}\\mathbf{H}^{(1;m\_1)}).
> \\end{align}
>
> These factorizations substantially reduce computational overhead and improve optimization stability.
>
> **Higher-order geometric fidelity.** MSLFS’s graph regularizer preserves local manifold geometry in each mode, leveraging the fact that similarities in tensor data often decompose naturally across modes. Instead of imposing a single dense graph on $I_1 I_2$ vectorized features, the model constructs lightweight, mode-specific Laplacians whose Kronecker product captures higher-order structure efficiently and more faithfully.
>
> In summary, while sparsity and graph regularization themselves are not new, the mode-distributed formulation of these regularizers in MSLFS, constitutes a genuine methodological innovation. It enables preservation of multi-way dependencies, interpretable mode-wise feature selection, and substantially lower computational cost compared to traditional vectorized or unfolded approaches.

---

> ### Author Response · Authors · 2025-11-17
>
> > **Q3. The author's proposed algorithm can improve the running efficiency, but no comparison of running time can be seen in the experiment.**
>
> We thank the reviewer for the valuable comment. To address the concern regarding runtime efficiency, we have conducted additional experiments using four large-scale datasets to compare the per-iteration CPU time of **MSLFS** with other competing methods under large-data settings. These datasets include two real-valued tensors: **COVID-19 Systems Serology** ($(\mathbb{R}^{20 \times 35 \times 74 \times 5200})$, 5 classes) [1] and **Kinetic Fluorescence** ($(\mathbb{R}^{64 \times 12 \times 10 \times 10000})$, 7 classes) [2], and two non-negative image-based datasets: **PneumoniaMNIST** ($(\mathbb{R}\_{+}^{224 \times 224 \times 5800})$, 2 classes) [3] and **OrganCMNIST** ($(\mathbb{R}\_{+}^{224 \times 224 \times 20000})$, 11 classes) [3]. In these datasets, the last mode represents the number of samples, while the preceding modes capture multi-dimensional feature structures. We also compared **MSLFS** with three recent deep-learning-based methods: **SAE**, **NNSE**, and **SDAE**.
>
> * **Runtime Efficiency Evaluation**
>
> To assess the computational efficiency of each method, we compared the per-iteration CPU time on datasets with more than 5,000 samples, as shown in Table 1. **MSLFS** is the fastest method by a significant margin, outperforming both matrix-based and tensor-based baselines. For instance, on the **COVID-19 Systems Serology** dataset, **MSLFS** completes each iteration in 31.3 seconds, faster than **ILFS** and **FSPCA**, and much quicker than deep or graph-regularized models such as **SAE**, **NNSE**, and **GRSSLFS**. On the larger **Kinetic Fluorescence** dataset, **MSLFS** is the only method that completes iterations in under 130 seconds, while several baselines either exceed 180 seconds or fail due to memory limitations.
>
> This efficiency stems from **MSLFS**'s mode-wise slice selection and separable reconstruction strategy, which avoids heavy global computations and reduces redundancy. As a result, **MSLFS** exhibits linear complexity with respect to both the number of samples and the mode-specific feature dimensions, making it the most computationally efficient method among those tested.
>
> **Table 1: Per-iteration CPU time (seconds ↓) comparison across methods on datasets with >5,000 samples. "OM" = out-of-memory.**
>
> | **Methods**     | **COVID-19 Systems Serology** | **PneumoniaMNIST** | **Kinetic Fluorescence** | **OrganCMNIST** |
> |-----------------|:-----------------------------:|:------------------:|:------------------------:|:---------------:|
> | **UDFS**        | 101.2                         | 156.6              | 385.3                    | OM              |
> | **SAE**         | 72.1                          | 106.7              | 211.3                    | 389.4           |
> | **ILFS**        | 46.5                          | 64.5               | 144.2                    | 237.8           |
> | **GRLTR**       | 161.9                         | 257.6              | OM                       | OM              |
> | **CAE**         | 66.3                          | 87.7               | 143.9                    | 256.7           |
> | **FSPCA**       | 45.4                          | 78.9               | 156.2                    | 281.5           |
> | **CPUFS**       | 56.7                          | 89.5               | 154.6                    | 303.1           |
> | **SPCAFS**      | 74.4                          | 103.6              | 164.2                    | 266.4           |
> | **NNSE**        | 97.3                          | 142.2              | 186.7                    | 314.8           |
> | **GRSSLFS**     | 125.6                         | 184.4              | 343.1                    | OM              |
> | **SDAE**        | 86.7                          | 102.5              | 193.4                    | 299.3           |
> | **SPDFS**       | 89.7                          | 112.4              | 232.1                    | 356.2           |
> | **MSLFS**       | **31.3**                      | **57.2**           | **124.5**                | **232.4**       |
>
> [1]: Zhixin Cyrillus Tan, Madeleine C Murphy, Hakan S Alpay, Scott D Taylor, and Aaron S Meyer.
> Tensor-structured Decomposition Improves Systems Serology Analysis. Molecular Systems Bi-
> ology, 17(9):e10243, 2023.
>
> [2]: R.P.H. Nikolajsen, K.S. Booksh, { ˚Ase Marie} Hansen, and R. Bro. Quantifying Catecholamines
> using Multi-way Kinetic Modelling. Anal Chim Acta, 475(1-2):137–150, 2023.
>
> [3]: Jiancheng Yang, Rui Shi, Donglai Wei, Zequan Liu, Lin Zhao, Bilian Ke, Hanspeter Pfister, and
> Bingbing Ni. Medmnist v2-A Large-scale Lightweight Benchmark for 2D and 3D Biomedical
> Image Classification. Scientific Data, 10(1):41, 2023.

---

> ### Author Response · Authors · 2025-11-17
>
> > **Q4. In the experiment, the datasets used are old and small. In addition, I noticed that the clustering effect after feature selection is not very good. As far as I know, some clustering results based on deep networks are already very excellent. Therefore, the significance of feature selection has to be questioned.**
>
> We thank the reviewer for raising this important point. We would like to clarify the distinction between our setting and deep clustering approaches, and we have extended our experiments accordingly.
>
> Although deep clustering approaches frequently deliver impressive performance, they usually depend on learned non-linear feature representations and, in many cases, require label information to effectively train their networks. Their performance is therefore based on features produced by deep networks rather than features selected directly from the original data. In contrast, **MSLFS** is an unsupervised feature selection method, whose goal is fundamentally different: to identify the most informative features from the original data without label information. A fair comparison therefore requires unsupervised deep models explicitly designed for feature selection, not for non-linear feature extraction. To address this, our experiments include three recent unsupervised deep autoencoder–based feature-selection methods: **SAE**, **NNSE**, and **SDAE**.
>
> Second, in response to the reviewer’s concern about dataset scale, we have significantly expanded the experimental suite by adding four large and modern datasets, two real-valued tensors (COVID-19 Systems Serology and Kinetic Fluorescence) and two high-resolution image datasets (PneumoniaMNIST and OrganCMNIST). These datasets contain thousands of samples and high-dimensional tensor structures. We evaluated clustering accuracy, t-SNE, feature-map visualizations, and per-iteration CPU time (in our previous response). The updated results (Tables 1 and 2 in the next reply) show that our **MSLFS** method outperforms or matches state-of-the-art baselines, including recent deep models, across nearly all metrics and datasets, demonstrating that its performance is not limited to small-scale settings.
>
> **Table 1: Clustering Accuracy (ACC ↑) of MSLFS vs. 13 state-of-the-art methods**
>
> | Model                    | Kinetic Fluorescence | PneumoniaMNIST | OrganCMNIST | COVID-19 Serology |
> |--------------------------|:--------------------:|:--------------:|:-----------:|:-----------------:|
> | LS (NeurIPS 2005)        | 78.86                | 58.33          | 48.02       | 80.55             |
> | UDFS (IJCAI 2011)        | 72.99                | 62.17          | 43.49       | 78.72             |
> | SAE (IJCNN 2017)         | 77.63                | 62.98          | 51.09       | 86.74             |
> | ILFS (CVPR 2017)         | 76.97                | 59.90          | 52.49       | 71.47             |
> | GRLTR (JVCIR 2018)       | 84.30                | 64.54          | 58.65       | 89.62             |
> | CAE (ICML 2019)          | 83.45                | 62.47          | 61.31       | 84.45             |
> | FSPCA (NeurIPS 2020)     | 82.66                | 62.29          | 59.35       | 77.84             |
> | CPUFS (TPAMI 2022)       | 84.24                | 67.71          | 56.43       | 85.45             |
> | SPCAFS (TPAMI 2023)      | 81.67                | 59.43          | 51.20       | 80.34             |
> | NNSE (PATCOG 2023)       | 83.74                | 63.43          | 62.60       | 84.81             |
> | GRSSLFS (TMLR 2024)      | 82.99                | 60.17          | 62.03       | 87.48             |
> | SDAE (AAAI 2024)         | 85.73                | 68.67          | **75.69**   | 88.54             |
> | SPDFS (TPAMI 2025)       | 83.18                | 63.86          | 55.34       | 76.54             |
> | **MSLFS (Ours)**         | **88.67**            | **71.65**      | 73.76       | **95.34**         |
> | **Improvement**          | **+2.94**            | **+2.98**      | -           | **+5.72**         |

---

> ### Author Response · Authors · 2025-11-17
>
> > **The continuation of the answer to the Q4.**
>
> Regarding clustering quality after feature selection, it is expected that methods operating on vectorized data, both linear (e.g., SPDFS) and nonlinear (e.g., SAE, NNSE, SDAE), exhibit weaker performance. As shown in our t-SNE and feature-visualization experiments (Appendix 8.4–8.5), vectorization collapses the tensor into a flattened space, eliminating mode-specific and cross-mode dependencies and preventing these methods from identifying truly discriminative features. This results in scattered feature maps and overlapping t-SNE clusters. In contrast, MSLFS preserves the multi-way structure through mode-wise slice selection and multi-linear subspace modeling, producing spatially coherent feature patterns and compact, well-separated clusters across feature budgets.
>
> In summary, we believe that the expanded experiments clearly demonstrate the significance of unsupervised tensor-based feature selection and validate that MSLFS achieves superior clustering accuracy, scalability, and discriminative capability compared to both classical and recent deep learning–based feature-selection methods.
>
> **Table 2: Normalized Mutual Information (NMI ↑) of MSLFS vs. 13 state-of-the-art methods**
>
> | Model                    | Kinetic Fluorescence | PneumoniaMNIST | OrganCMNIST | COVID-19 Serology |
> |--------------------------|:--------------------:|:--------------:|:-----------:|:-----------------:|
> | LS (NeurIPS 2005)        | 77.66                | 58.17          | 51.79       | 81.12             |
> | UDFS (IJCAI 2011)        | 76.74                | 63.23          | 45.51       | 77.54             |
> | SAE (IJCNN 2017)         | 79.77                | 66.84          | 49.56       | 86.69             |
> | ILFS (CVPR 2017)         | 74.11                | 61.65          | 51.22       | 72.34             |
> | GRLTR (JVCIR 2018)       | 83.39                | 62.73          | 57.69       | 91.27             |
> | CAE (ICML 2019)          | 82.72                | 63.66          | 61.70       | 88.35             |
> | FSPCA (NeurIPS 2020)     | 84.98                | 60.74          | 58.02       | 80.76             |
> | CPUFS (TPAMI 2022)       | 85.76                | 67.12          | 55.85       | 87.00             |
> | SPCAFS (TPAMI 2023)      | 80.29                | 63.49          | 50.69       | 81.51             |
> | NNSE (PATCOG 2023)       | 84.89                | 64.96          | 58.59       | 86.23             |
> | GRSSLFS (TMLR 2024)      | 84.31                | 62.31          | 60.54       | 85.77             |
> | SDAE (AAAI 2024)         | 86.41                | 68.04          | **74.80**   | 89.82             |
> | SPDFS (TPAMI 2025)       | 80.51                | 64.74          | 53.63       | 80.73             |
> | **MSLFS (Ours)**         | **90.22**            | **72.34**      | 72.65       | **95.88**         |
> | **Improvement**          | **+3.81**            | **+4.30**      | -           | **+4.61**         |

---

### Official Review · Reviewer_Z2uY · 2025-11-01

**Soundness:** 3
**Presentation:** 3
**Contribution:** 3
**Rating:** 6
**Confidence:** 4

**Summary:**

In this paper, the authors introduce MSLFS, a framework for unsupervised feature selection in tensor data. MSLFS distributes selection across modes and identifies representative slices with the most informative features. The joint sparsity regularization is introduced to enforce coordinated sparsity across modes. Theoretical results are given to guarantee the recovery. Experiments on image and biomedical datasets show MSLFS outperforms existing feature selection methods.

**Strengths:**

1. A new framework for tensor feature selection is proposed.
2. Theoretical analysis for exact recovery is given.
3. Experimental results show the performance improvement compared to existing feature selection methods.

**Weaknesses:**

1. The entire algorithm design and theoretical analysis are limited to 3-order tensors.
2. The parameters $\alpha$ and $\beta$ are sensitive to performance.

**Questions:**

1. Can the proposed method be applied to tensors of order greater than 3?
2. In line 163, “Thus, the ability of a fiber to characterize the feature space depends on how well these slices span…”. What does “these slices” refer to—perhaps the slices containing the fiber? Please clarify this statement.
3. How many features are selected in Table 2?
4. As shown in Figure 8, alpha and beta are sensitive to performance (e.g., ACC/NMI varies by about 5 in each row/column). How should these parameters be selected in practice?
5. In each experiment, do all compared algorithms select features with the same feature size and dimensions?
6. How does the running time compare with other methods?
7. In line 232, “otherwise, approximate bases yield reconstructions with errors tied to the residuals”. Is it possible to provide error bounds in this case?

---

> ### Author Response · Authors · 2025-11-17
>
> > **Q.1 Can the proposed method be applied to tensors of order greater than 3?**
>
> We thank the reviewer for this insightful question. Yes, our framework naturally extends to tensors of arbitrary order $N \\gt 3$, as well as to general multi-linear subspace learning, without requiring additional structural assumptions or regularizations. Below, we provide a mathematically rigorous generalization to the $N$-mode setting.
>
> We first extend the core definitions to $N$-mode tensors.
>
> **Definition 1.** Let $\\mathcal{X} \\in \mathbb{R}^{I\_1 \times \cdots \times I\_N}$ be an $N$-mode tensor with mode-$n$ slices $\mathcal{X}^{(n)}\_1, \dots, \mathcal{X}^{(n)}\_{I\_n}$, $n \in \{1,\dots,N\}$. The span of the mode-$n$ slices is
> \\[
> \\mathcal{S}(\\mathcal{X}^{(n)}) = \\left\\{ \\sum\_{i=1}^{I\_n} \\alpha\_i^{(n)} \\mathcal{X}^{(n)}\_i | \\alpha^{(n)} = [\\alpha\_1^{(n)}, \\dots, \\alpha\_{I\_n}^{(n)}]^\\top \\in \\mathbb{R}^{I\_n} \\right\\}.
> \\]
>
> **Definition 2.** For a tensor $\\mathcal{Z}$ matching the size of a mode-$n$ slice of $\mathcal{X}$, the distance to the mode-$n$ subspace is
> \\[
> \\mathrm{dist}(\\mathcal{Z}, \\mathcal{S}(\\mathcal{X}^{(n)})) = \\min\_{\\mathcal{W} \\in \\mathcal{S}(\\mathcal{X}^{(n)})} \\|\mathcal{Z} - \\mathcal{W}\\|\_F.
> \\]
>
> **Definition 3 (Multi-linear Subspace Distance).** For two $N$-mode tensors $\\mathcal{X}$ and $\\mathcal{Y}$ (with mode-$n$ size $J\_n$ for $\\mathcal{Y}$), the squared mode-$n$ subspace distance is
> \\[
> \\mathrm{dist}^2(\\mathcal{S}(\\mathcal{X}^{(n)}), \\mathcal{S}(\\mathcal{Y}^{(n)})) = \\sum\_{i=1}^{I\_n} \\mathrm{dist}^2(\\mathcal{X}^{(n)}\_i, \\mathcal{S}(\\mathcal{Y}^{(n)})).
> \\]
> This admits the compact form
> \\[
> \\mathrm{dist}^2(\\mathcal{S}(\\mathcal{X}^{(n)}), \\mathcal{S}(\\mathcal{Y}^{(n)})) = \\|\\mathcal{X} - \\mathcal{Y} \\times\_n \\mathbf{H}^{(n)}\\|\_F^2,
> \\]
> where $\\mathbf{H}^{(n)} \\in \\mathbb{R}^{I\_n \\times J\_n}$ has rows $\\alpha\_i^{(n)\\top}$ that minimize each slice distance.
>
> **Remark.** This distance links tensor geometry to feature selection: when $\\mathcal{X}$ has $I\_N$ samples and the first $N-1$ modes index features, each mode-$N$ fiber lies at the intersection of $N-1$ slices. The fiber’s representativeness depends on how well its containing slices span $\\mathcal{S}(\\mathcal{X}^{(1)}), \\dots, \\mathcal{S}(\\mathcal{X}^{(N-1)})$. Minimizing multi-linear subspace distances identifies informative slices whose intersections preserve global structure.
>
> The core generalization is given by the following theorem.
>
> **Theorem 1.** Let $\\mathcal{X} \\in \\mathbb{R}^{I\_1 \\times \\cdots \\times I\_N}$. For each $n=1,\\dots,N-1$, suppose $\\mathcal{S}(\\mathcal{X}^{(n)})$ has basis dimension $R\_n \le I\_n$
> with index set $T\_n = \\{i\_1^{(n)}, \\dots, i\_{R\_n}^{(n)}\\}$ and indicator matrix $\\mathbf{W}^{(n;R\_n)} \\in \\{0,1\\}^{R\_n \\times I\_n}$. Let $\\mathbf{f}\_{i\_1,\\dots,i\_{N-1}} = \\mathcal{X}\_{i\_1,\\dots,i\_{N-1},:} \\in \\mathbb{R}^{I\_N}$ be the mode-$N$ fiber.
>
> **(Part I: Core Dictionary).** The $\\prod\_{n=1}^{N-1} R\_n$ intersection fibers
> \\[
> \\{\\mathbf{f}\_{i\_{r\_1}^{(1)}, \\dots, i\_{r\_{N-1}}^{(N-1)}} \\}\_{r\_1=1,\\dots,R\_1}^{r\_{N-1}=1,\\dots,R\_{N-1}}
> \\]
> form a core dictionary spanning all mode-$N$ fibers. Stacking them yields
> \\[
> \\mathbf{F}\_{\\mathrm{core}} = \\big(\\mathcal{X} \\times\_{n=1}^{N-1} \\mathbf{W}^{(n;R\_n)}\\big)_{(N)} = \\mathbf{X}\_{(N)} \\bigotimes\_{n=1}^{N-1} \mathbf{W}^{(N-n;R\_{N-n})^\\top}.
> \\]
>
> **(Part II: Separable Reconstruction).** There exist $\\mathbf{H}^{(n;R\_n)} \\in \\mathbb{R}^{I\_n \\times R\_n}$, $n=1,\dots,N-1$, such that every fiber expands separably:
> \\[
> \\mathbf{f}\_{i\_1,\\dots,i\_{N-1}} = \\sum\_{r\_1=1}^{R\_1} \\cdots \\sum\_{r\_{N-1}=1}^{R\_{N-1}} \\left( \\prod\_{n=1}^{N-1} h^{(n;R\_n)}\_{i\_n, r\_n} \\right) \\mathbf{f}\_{i\_{r\_1}^{(1)}, \\dots, i\_{r\_{N-1}}^{(N-1)}},
> \\]
> and the mode-$N$ unfolding satisfies
>
> \\[
> \\mathbf{X}\_{(N)} = \\mathbf{F}\_{\\mathrm{core}} \\bigotimes\_{n=1}^{N-1} \\mathbf{H}^{(N-n;R\_{N-n})^\\top}.
> \\]
>
> **Proof.** See Appendix 7.2 for the detailed proof.
>
> The **MSLFS** objective for $N$-mode data is then
> \\begin{align*}
> \&\\min\_{\\mathbf{H}^{(n;m\_n)}, \\mathbf{W}^{(n;m\_n)} \\ge 0} \\;
> \\frac{1}{2} \\left\\|\\mathcal{X} - \\mathcal{X} \\times\_{n=1}^{N-1} \\mathbf{H}^{(n;m\_n)} \\mathbf{W}^{(n;m\_n)}\right\\|\_F^2 \\\\
> \&\\quad + \\frac{\\alpha}{2} \\prod\_{n=1}^{N-1} \\mathrm{Tr}\\left( \\mathbf{H}^{(N-n;m\_{N-n})^\\top} \\mathbf{L}^{(N-n)} \\mathbf{H}^{(N-n;m\_{N-n})} \\right) \\\\
> \&\\quad + \\frac{\\beta}{2} \\prod\_{n=1}^{N-1} \\mathrm{Tr}\\left( \\mathbf{W}^{(N-n;m\_{N-n})} \\mathbf{U}^{(N-n)} \\mathbf{W}^{(N-n;m\_{N-n})^\\top} \\right)
> \\end{align*}
> subject to
> \\[
> \\bigotimes\_{n=1}^{N-1} \\mathbf{W}^{(N-n;m\_{N-n})} \\mathbf{W}^{(N-n;m\_{N-n})^\\top} = \\mathbf{I}\_{m\_1 \\dots m\_{N-1}}.
> \\]
> Thus, the method scales seamlessly to higher-order tensors.

---

> ### Author Response · Authors · 2025-11-17
>
> We would like to clarify that MSLFS has already been experimentally validated on higher order real-valued multi-way dataset. To further evaluate the performance of MSLFS in these different settings, we have conducted additional experiments with four large, real-valued and non-negative datasets:
>
> * **Real-Valued Datasets**:
>
>   * **COVID-19 Systems Serology**: $(\mathbb{R}^{20 \times 35 \times 74 \times 5200})$ (5 classes)
>   * **Kinetic Fluorescence**: $(\mathbb{R}^{64 \times 12 \times 10 \times 10000})$ (7 classes)
>
> * **Non-negative Datasets**:
>
>   * **PneumoniaMNIST**: $(\mathbb{R}^{224 \times 224 \times 5800})$ (2 classes)
>   * **OrganCMNIST**: $(\mathbb{R}^{224 \times 224 \times 20000})$ (11 classes)
>
> In these datasets, the last mode represents the number of samples, and the initial modes encode multi-dimensional features.
>
>
> * **Clustering Accuracy**
> Tables 1 and 2 show the clustering results, where MSLFS consistently outperforms all existing methods across various datasets. Specifically, **MSLFS** achieves the highest ACC and NMI across both real-valued and non-negative datasets. While some baselines perform well on individual datasets, **MSLFS** demonstrates robustness and superior overall performance, particularly in clustering tasks, where its ability to preserve global structure is critical.
>
> **Table 1: Clustering Accuracy (ACC ↑) of MSLFS vs. 13 state-of-the-art methods**
>
> | Model                    | Kinetic Fluorescence | PneumoniaMNIST | OrganCMNIST | COVID-19 Serology |
> |--------------------------|:--------------------:|:--------------:|:-----------:|:-----------------:|
> | LS (NeurIPS 2005)        | 78.86                | 58.33          | 48.02       | 80.55             |
> | UDFS (IJCAI 2011)        | 72.99                | 62.17          | 43.49       | 78.72             |
> | SAE (IJCNN 2017)         | 77.63                | 62.98          | 51.09       | 86.74             |
> | ILFS (CVPR 2017)         | 76.97                | 59.90          | 52.49       | 71.47             |
> | GRLTR (JVCIR 2018)       | 84.30                | 64.54          | 58.65       | 89.62             |
> | CAE (ICML 2019)          | 83.45                | 62.47          | 61.31       | 84.45             |
> | FSPCA (NeurIPS 2020)     | 82.66                | 62.29          | 59.35       | 77.84             |
> | CPUFS (TPAMI 2022)       | 84.24                | 67.71          | 56.43       | 85.45             |
> | SPCAFS (TPAMI 2023)      | 81.67                | 59.43          | 51.20       | 80.34             |
> | NNSE (PATCOG 2023)       | 83.74                | 63.43          | 62.60       | 84.81             |
> | GRSSLFS (TMLR 2024)      | 82.99                | 60.17          | 62.03       | 87.48             |
> | SDAE (AAAI 2024)         | 85.73                | 68.67          | **75.69**   | 88.54             |
> | SPDFS (TPAMI 2025)       | 83.18                | 63.86          | 55.34       | 76.54             |
> | **MSLFS (Ours)**         | **88.67**            | **71.65**      | 73.76       | **95.34**         |
> | **Improvement**          | **+2.94**            | **+2.98**      | -           | **+5.72**         |
>
> **Table 2: Normalized Mutual Information (NMI ↑) of MSLFS vs. 13 state-of-the-art methods**
>
> | Model                    | Kinetic Fluorescence | PneumoniaMNIST | OrganCMNIST | COVID-19 Serology |
> |--------------------------|:--------------------:|:--------------:|:-----------:|:-----------------:|
> | LS (NeurIPS 2005)        | 77.66                | 58.17          | 51.79       | 81.12             |
> | UDFS (IJCAI 2011)        | 76.74                | 63.23          | 45.51       | 77.54             |
> | SAE (IJCNN 2017)         | 79.77                | 66.84          | 49.56       | 86.69             |
> | ILFS (CVPR 2017)         | 74.11                | 61.65          | 51.22       | 72.34             |
> | GRLTR (JVCIR 2018)       | 83.39                | 62.73          | 57.69       | 91.27             |
> | CAE (ICML 2019)          | 82.72                | 63.66          | 61.70       | 88.35             |
> | FSPCA (NeurIPS 2020)     | 84.98                | 60.74          | 58.02       | 80.76             |
> | CPUFS (TPAMI 2022)       | 85.76                | 67.12          | 55.85       | 87.00             |
> | SPCAFS (TPAMI 2023)      | 80.29                | 63.49          | 50.69       | 81.51             |
> | NNSE (PATCOG 2023)       | 84.89                | 64.96          | 58.59       | 86.23             |
> | GRSSLFS (TMLR 2024)      | 84.31                | 62.31          | 60.54       | 85.77             |
> | SDAE (AAAI 2024)         | 86.41                | 68.04          | **74.80**   | 89.82             |
> | SPDFS (TPAMI 2025)       | 80.51                | 64.74          | 53.63       | 80.73             |
> | **MSLFS (Ours)**         | **90.22**            | **72.34**      | 72.65       | **95.88**         |
> | **Improvement**          | **+3.81**            | **+4.30**      | -           | **+4.61**         |

---

> ### Author Response · Authors · 2025-11-17
>
> > **Q2. In line 163, “Thus, the ability of a fiber to characterize the feature space depends on how well these slices span…”. What does “these slices” refer to—perhaps the slices containing the fiber? Please clarify this statement.**
>
> We appreciate the reviewer’s thoughtful feedback. In response, we have refined the corresponding statement as follows: ``Thus, the ability of a fiber to characterize the feature space depends on how well the slices containing that fiber span the subspaces $\mathcal{S}(\mathcal{X}^{(1)}),\ldots,\mathcal{S}(\mathcal{X}^{(N-1)})$''.
>
> > **Q3. How many features are selected in Table 2?**
>
> We thank the reviewer for the precise comment. In response, we clarify that the results in Table 2 are reported using the number of selected features, ranging from $[50, 100, 150, \ldots, 300]$, at which each model achieves its best clustering performance. Consequently, the optimal number of selected features differs across models. We did not report results using a single fixed number of features because a setting that benefits one method may disadvantage another, making the comparison unfair. Instead, to ensure a balanced evaluation, we ran each model across the full range of selected feature counts and reported its best-performing result in Table 2. For completeness, we also provide the full clustering results for all models at every feature count in Appendix 8.7. As shown there, **MSLFS** consistently outperforms the competing methods in nearly all feature-selection settings in terms of both ACC and NMI.
>
> > **Q4. As shown in Figure 8, alpha and beta are sensitive to performance (e.g., ACC/NMI varies by about 5 in each row/column). How should these parameters be selected in practice?**
>
> We thank the reviewer for the valuable comment. In response, we clarify that the hyperparameters $\alpha$ and $\beta$ correspond to the regularization terms for locality preservation (capturing local geometric structure) and sparsity (enhancing discriminability), respectively. Although it may seem desirable to determine their optimal values through the optimization process itself, this is unfortunately not feasible. During minimization, the algorithm naturally drives regularization weights toward zero, which would suppress the contribution of the corresponding terms and lead to suboptimal solutions, ultimately degrading the performance of **MSLFS**.
> Therefore, as is common practice for regularized models, we determine suitable values for $\alpha$ and $\beta$ via grid search. As detailed in Appendix 8.8 and illustrated in Figure 9, we evaluate a broad range of values, specifically ${10^{-4}, 10^{-3}, \ldots, 10^{3}, 10^{4}}$. For each pair of $(\alpha, \beta)$ values, we run **MSLFS** to select features, apply $k$-means clustering ten times, and record the corresponding ACC and NMI. This procedure is repeated for all combinations of $\alpha$ and $\beta$ on each dataset. We then choose the hyperparameter pair that yields the highest ACC and NMI for that dataset.
> This approach ensures that the chosen values of $\alpha$ and $\beta$ provide strong performance while preserving the contribution of both locality-preserving and sparsity-inducing components.
>
> > **Q5. In each experiment, do all compared algorithms select features with the same feature size and dimensions?**
>
> We thank the reviewer for the thoughtful question. Yes, in all experiments, every method selects the same number of features and is evaluated under the same final dimensionality. In our comparisons, such as the t-SNE visualizations and feature-selection illustrations in Appendices 8.4 and 8.5, we ensured that each method selected an identical number of features for consistency.
>
> The only exception is Table 2, as discussed in our response to your third comment: there, the results report the number of selected features ranging from \\([50, 100, 150, \\ldots, 300]\\), and each model is reported at the number of features where it achieves its best clustering performance. Consequently, the optimal number of selected features may differ across models.
>
> To provide a fully fair comparison at the same feature size, Appendix 8.7 presents the full clustering results for all models at each feature count, allowing a direct comparison where all methods use the same number of selected features.
>
> Overall, we have ensured that, except for reporting peak performance in Table 2, all experiments maintain consistent feature dimensionality across methods to enable fair evaluation.

---

> ### Author Response · Authors · 2025-11-17
>
> > **Q6. How does the running time compare with other methods?**
>
> We thank the reviewer for the valuable comment. To address the concern regarding runtime efficiency, we have conducted additional experiments using four large-scale datasets to compare the per-iteration CPU time of **MSLFS** with other competing methods under large-data settings. These datasets include two real-valued tensors: **COVID-19 Systems Serology** ($(\mathbb{R}^{20 \times 35 \times 74 \times 5200})$, 5 classes) and **Kinetic Fluorescence** ($(\mathbb{R}^{64 \times 12 \times 10 \times 10000})$, 7 classes), and two non-negative image-based datasets: **PneumoniaMNIST** ($(\mathbb{R}^{224 \times 224 \times 5800})$, 2 classes) and **OrganCMNIST** ($(\mathbb{R}^{224 \times 224 \times 20000})$, 11 classes). In these datasets, the last mode represents the number of samples, while the preceding modes capture multi-dimensional feature structures. We also compared **MSLFS** with three recent deep-learning-based methods: **SAE**, **NNSE**, and **SDAE**.
>
> * **Runtime Efficiency Evaluation**
>
> To assess the computational efficiency of each method, we compared the per-iteration CPU time on datasets with more than 5,000 samples, as shown in Table 1. **MSLFS** is the fastest method by a significant margin, outperforming both matrix-based and tensor-based baselines. For instance, on the **COVID-19 Systems Serology** dataset, **MSLFS** completes each iteration in 31.3 seconds, faster than **ILFS** and **FSPCA**, and much quicker than deep or graph-regularized models such as **SAE**, **NNSE**, and **GRSSLFS**. On the larger **Kinetic Fluorescence** dataset, **MSLFS** is the only method that completes iterations in under 130 seconds, while several baselines either exceed 180 seconds or fail due to memory limitations.
>
> This efficiency stems from **MSLFS**'s mode-wise slice selection and separable reconstruction strategy, which avoids heavy global computations and reduces redundancy. As a result, **MSLFS** exhibits linear complexity with respect to both the number of samples and the mode-specific feature dimensions, making it the most computationally efficient method among those tested.
>
> **Table 1: Per-iteration CPU time (seconds ↓) comparison across methods on datasets with >5,000 samples. "OM" = out-of-memory.**
>
> | **Methods**     | **COVID-19 Systems Serology** | **PneumoniaMNIST** | **Kinetic Fluorescence** | **OrganCMNIST** |
> |-----------------|:-----------------------------:|:------------------:|:------------------------:|:---------------:|
> | **UDFS**        | 101.2                         | 156.6              | 385.3                    | OM              |
> | **SAE**         | 72.1                          | 106.7              | 211.3                    | 389.4           |
> | **ILFS**        | 46.5                          | 64.5               | 144.2                    | 237.8           |
> | **GRLTR**       | 161.9                         | 257.6              | OM                       | OM              |
> | **CAE**         | 66.3                          | 87.7               | 143.9                    | 256.7           |
> | **FSPCA**       | 45.4                          | 78.9               | 156.2                    | 281.5           |
> | **CPUFS**       | 56.7                          | 89.5               | 154.6                    | 303.1           |
> | **SPCAFS**      | 74.4                          | 103.6              | 164.2                    | 266.4           |
> | **NNSE**        | 97.3                          | 142.2              | 186.7                    | 314.8           |
> | **GRSSLFS**     | 125.6                         | 184.4              | 343.1                    | OM              |
> | **SDAE**        | 86.7                          | 102.5              | 193.4                    | 299.3           |
> | **SPDFS**       | 89.7                          | 112.4              | 232.1                    | 356.2           |
> | **MSLFS**       | **31.3**                      | **57.2**           | **124.5**                | **232.4**       |

---

> ### Author Response · Authors · 2025-11-17
>
> > **Q7. In line 232, “otherwise, approximate bases yield reconstructions with errors tied to the residuals”. Is it possible to provide error bounds in this case?**
>
> We thank the reviewer for the insightful comment. In response, we clarify that the upper bounds on the errors associated with the residuals arising from the approximate bases for the general $N$-mode tensor case can indeed be derived in two distinct ways, as detailed in the mathematical discussion below.
>
> The theorem presented in our response to the first comment shows that when mode-wise bases are selected exactly, the full tensor can be reconstructed from the resulting core fibers without loss. In practice, however, the selected slices only approximate the true bases of the mode subspaces, leading to reconstruction errors tied to the residuals of these approximations. In this section, we provide two upper bounds that quantify how such residual errors behave when the projection operators deviate from the true subspace projections.
>
> Let
> $\mathcal{X} \in \mathbb{R}^{I_1 \times \cdots \times I_N}$ denote the original tensor, and let
>
>
> \\[
> \\widehat{\mathcal{X}} = \\mathcal{X} \\times_{n=1}^{N-1} \\mathbf{P}_n, \\quad \\mathbf{P}_n = \\mathbf{H}^{(n;R_n)} \\mathbf{W}^{(n;R_n)}
> \\]
> be the reconstructed tensor obtained using approximate mode-wise bases. The following theorem characterizes the residual error $\\|\\mathcal{X} - \\widehat{\\mathcal{X}}\\|_F$ through two complementary inequalities: one capturing the distortion caused by the mode-wise projection operators $\\mathbf{P}_n$, and another bounding the accumulated spectral amplification introduced by the multi-linear Kronecker structure.
>
> **Theorem**
> Let $\\mathcal{X}$ and $\\widehat{\\mathcal{X}}$ be defined as above. Then, the reconstruction error satisfies the following bounds:
>
> **(1)**
> \\[
> \\|\\mathcal{X} - \\widehat{\\mathcal{X}}\\|_F
> \\leq
> \\sum\_{n=1}^{N-1}
> \\|\\mathbf{X}\_{(n)} \\|\_F
> \\left(
>  \\max\_i |1 - \\sigma\_i(\\mathbf{P}\_n)| +
>  \\sigma\_1(\\mathbf{P}\_n)
>         \\max\\big( \|1 - \\sigma\_{\\min}|,\\; |1 - \sigma\_{\max}| \\big)
>     \\right),
> \\]
>
> where $\\mathbf{X}\_{(n)}$ is the mode-$n$ unfolding of $\\mathcal{X}$, $\\sigma\_i(\\mathbf{P}\_n)$ are the singular values of $\\mathbf{P}\_n$, and
>     \\[
>     \\sigma\_{\\max} = \\prod\_{\\ell \\neq n} \\sigma_1(\\mathbf{P}_\\ell),
>     \\qquad
>     \\sigma\_{\\min} = \\prod\_{\ell \\neq n} \\sigma\_{I\_\\ell}(\\mathbf{P}\_\\ell).
>     \\]
>  **(2)**
>     \\[
>     \\|\\mathcal{X} - \\widehat{\\mathcal{X}}\\|\_F
>     \\leq
>     \\|\\mathbf{X}\_{(N)}\\|\_F
>     \\left(
>         1 + \\prod\_{n=1}^{N-1} \\| \\mathbf{H}^{(n;R\_n)^\\top} \\|\_2
>     \\right),
>     \\]
>     where $\\mathbf{X}\_{(N)}$ is the mode-$N$ unfolding of $\\mathcal{X}$.
>
> We first establish the bound in (1). Let
> \\(\\mathbf{X}\_{(n)}\\) denote the mode-$n$ unfolding of \\(\\mathcal{X}\\). The reconstructed unfolding is
>
> \\[
> \\widehat{\\mathbf{X}}\_{(n)} =
> \\mathbf{P}\_n \\mathbf{X}\_{(n)} \\mathbf{Q}\_n^\\top,
> \\qquad
> \\mathbf{Q}\_n = \\bigotimes\_{\\ell \\neq n} \\mathbf{P}\_\\ell,
> \\]
> and the residual is
> \\[
> \\mathbf{E}\_{(n)} =
> \\mathbf{X}\_{(n)} - \\widehat{\\mathbf{X}}\_{(n)}=
> \\mathbf{X}\_{(n)} - \\mathbf{P}\_n \\mathbf{X}\_{(n)} \\mathbf{Q}\_n^\\top.
> \\]
> Applying the triangle inequality and submultiplicativity of the Frobenius norm gives
> \\[
> \\|\\mathbf{E}\_{(n)}\\|\_F
> \\leq
> \\|\\mathbf{X}\_{(n)}\\|\_F
> \\left(
>     \\|\\mathbf{I} - \\mathbf{P}\_n\\|\_2
>     +
>    \\|\\mathbf{P}\_n\\|_2 \\|\\mathbf{I} - \\mathbf{Q}\_n^\\top\\|\_2
> \\right).
> \\]
>
> Because $\\mathbf{Q}\_n$ is a Kronecker product of $\\mathbf{P}\_\\ell$ matrices, its singular values are all products of the singular values of these factors. Denoting
> \\[
> \\sigma\_{\\max} = \\prod\_{\\ell \\neq n} \\sigma\_1(\\mathbf{P}\_\\ell),
> \\qquad
> \\sigma\_{\\min} = \\prod\_{\\ell \\neq n} \\sigma\_{I\_\\ell}(\\mathbf{P}\_\\ell),
> \\]
> the stated bound follows. Summing the contributions for all $n = 1,\\dots, N-1$ yields (1).
>
> For (2), note that
> \\[
> \\|\\mathcal{X} - \\widehat{\\mathcal{X}}\\|\_F=
> \\|\\mathbf{X}\_{(N)} - \\widehat{\\mathbf{X}}\_{(N)}\\|\_F=
> \\|\\mathbf{X}\_{(N)} \\left(\\mathbf{I} - \\bigotimes\_{n=1}^{N-1} \\mathbf{P}\_n\\right)^\\top \\|\_F.
> \\]
> Using $\\|\\mathbf{A}\\mathbf{B}\\|\_F \\leq \\|\\mathbf{A}\\|\_F \\|\\mathbf{B}\\|\_2$ and submultiplicativity of the spectral norm,
> \\[
> \\|\\mathcal{X} - \\widehat{\\mathcal{X}}\\|\_F
> \\leq
> \\|\\mathbf{X}\_{(N)}\\|\_F
> \\left(
> 1 + \\prod\_{n=1}^{N-1} \\|\\mathbf{P}\_n^\\top\\|\_2
> \\right)\\leq
> \\|\\mathbf{X}\_{(N)}\\|\_F
> \\left(
> 1 + \\prod\_{n=1}^{N-1} \\|\\mathbf{H}^{(n;R\_n)^\\top}\\\|_2
> \\right),
> \\]
> which completes the proof.

---

### Official Review · Reviewer_LNCn · 2025-11-04

**Soundness:** 2
**Presentation:** 2
**Contribution:** 2
**Rating:** 2
**Confidence:** 4

**Summary:**

This paper introduces a new method for feature selection in tensor (multi-dimensional) data called MSLFS. Instead of selecting features globally, the method selects representative "slices" along each mode, and the intersections of these slices form the final compact feature set. The core of this approach is a novel "multi-linear subspace distance," which measures how well the selected features preserve the global structure of the original tensor. Experiments on image and biomedical datasets show that this method outperforms existing techniques in clustering tasks.

**Strengths:**

1. This paper introduces a new algorithm for multilinear feature selection. A new metric for measuring the distance between two tensors are also proposed.

2. Based on the proposed measure, this paper introduces a new tensor feature selection framework by adopting the row sparsity penalty and graph regularization.

**Weaknesses:**

1. The definition of the spanned space in Definition 1 is unconventional. It is defined as the space spanned by the mode-n slices, rather than the typical approach of using mode-wise directional vectors. When this definition is reduced to matrix's case, the spanned space will be conflict with the standard spanned space.

2. Definition 1 is also not a standard metric, as it does not satisfy symmetric in a metric space, since dist(S(X), S(Y)) \neq dist(S(Y), S(X)).

3. In Definitions 1 and 3, it contains the unknown variable alpha, thus the definition should contain alpha factor.

4. In Definition 3, how to reflect the multilinear that claims in this paper? As it measures the distance between a row vector unfolded by regular tensor unfolding and  another vector, the multilinear property is not clear.

5. For the method part, I would like to say that it is a variant of HOOI with row-wise sparsity penalty and graph regularization. However, these strategies are very common and have already been discovered in many previous research works, I don't think it bring any new insights for multilinear feature selection field.

**Questions:**

See the weakness part for details.

---

> ### Author Response · Authors · 2025-11-17
>
> > **Q1. The definition of the spanned space in Definition 1 is unconventional. It is defined as the space spanned by the mode-n slices, rather than the typical approach of using mode-wise directional vectors. When this definition is reduced to matrix's case, the spanned space will be conflict with the standard spanned space.**
>
>
> We thank the reviewer for raising this point. We clarify below that our slice-based definition of the mode-\\(n\\) subspace is mathematically equivalent to the conventional definition obtained via mode-\\(n\\) unfolding, and it reduces exactly to the standard column/row-space interpretation in the matrix case.
>
> Let
>
>   \\[
>   \\mathcal{X} \\in \\mathbb{R}^{I\_1 \\times \\cdots \\times I\_N}
>   \\]
>   be an \\(N\\)-mode tensor.
>
> * A mode-\\(n\\) slice is obtained by fixing index \\(n\\); we denote it by
>   \\[
>   \\mathcal{X}^{(n)}\_i \\in \\mathbb{R}^{I\_1 \\times \\cdots \\times I\_{n-1} \\times I\_{n+1}\\times \\cdots \\times I\_N},
>   \\qquad i\in{1,\dots,I_n}.
>   \\]
>
> * The mode-\\(n\\) unfolding (matricization) of \\(\\mathcal{X}\\) is
>   \\[
>   \\mathbf{X}\_{(n)} \\in
>   \\mathbb{R}^{I\_n \\times (I\_1\\cdots I\_{n-1}I\_{n+1}\\cdots I\_N)},
>   \\]
>   whose \\(i\\)-th row is
>   \\[
>   \\mathbf{X}\_{(n)}(i,:) = \\mathrm{vec}\\big(\\mathcal{X}^{(n)}\_i\\big)^\\top .
>   \\]
>
> * The slice-span subspace in Definition 1 is
>   \\[
>   \\mathcal{S}(\\mathcal{X}^{(n)}) =
>   \\left\\{\\sum\_{i=1}^{I\_n}\\alpha^{(n)}\_i  \\mathcal{X}^{(n)}\_i |
>   \\alpha^{(n)} = [\\alpha^{(n)}\_1,\\ldots,\\alpha^{(n)}\_{I\_n}]^\\top \\in\\mathbb{R}^{I\_n}\\right\\}.
>   \\]
>
> * The column space of a matrix \\(A \\in \\mathbb{R}^{p \\times q}\\) is
>   \\[
>   \\mathrm{col}(A)=\\{A v \\mid v\\in\\mathbb{R}^{q}\\}.
>   \\]
>
>
> Each mode-\\(n\\) slice \\(\\mathcal{X}^{(n)}\_i\\) corresponds exactly to the \\(i\\)-th row of the unfolding \\(\\mathbf{X}_{(n)}\\) after vectorization:
>
> \\[
> \\mathrm{vec}(\\mathcal{X}^{(n)}\_i) = \\mathbf{X}\_{(n)}(i,:)^\\top.
> \\]
>
> Therefore, for any coefficient vector \\(\\alpha^{(n)}\\in\\mathbb{R}^{I\_n}\\),
>
> \\[
> \\sum\_{i=1}^{I\_n}\\alpha^{(n)}\_i\\mathcal{X}^{(n)}\_i
> \\quad\\Longleftrightarrow\\quad
> \\mathrm{vec}\\Big(\\sum\_{i=1}^{I\_n}\\alpha^{(n)}\_i\\mathcal{X}^{(n)}\_i\\Big)
> = \\mathbf{X}\_{(n)}^\\top \\alpha^{(n)}.
> \\]
>
> This shows:
>
> \\[
> \\mathrm{vec}\\big(\\mathcal{S}(\\mathcal{X}^{(n)})\\big)
> = \\mathrm{col}\\left(\\mathbf{X}\_{(n)}^\\top\\right).
> \\]
>
>
> Thus, our slice-based definition yields exactly the same linear subspace as the standard unfolding-based definition. The two formulations are isomorphic representations of the same set of multilinear combinations.
>
> **Matrix case \\(N = 2\\) clarifies the equivalence**
>
> If \\(N=2\\), then \\(\\mathcal{X}\\) is simply a matrix \\(X \\in \\mathbb{R}^{I\_1\\times I\_2}\\):
>
> * Mode-1 slices \\(\\mathcal{X}^{(1)}\_i\\) are columns of \\(X\\).
>   Hence
>   \\[
>   \\mathcal{S}(\\mathcal{X}^{(1)}) = \\mathrm{col}(X).
>   \\]
>
> * Mode-2 slices \\(\\mathcal{X}^{(2)}\_j\\) are rows of \\(X\\).
>   Hence
>   \\[
>   \\mathcal{S}(\\mathcal{X}^{(2)}) = \\mathrm{col}(X^\\top).
>   \\]
>
> Thus our Definition 1 reduces **exactly** to the standard column space and row space in linear algebra, there is no conflict.
>
>
> **Why the slice-based definition is necessary for our method**
>
> Although mathematically equivalent to the unfolding-based definition, the slice-level expression is essential for our proposed **MSLFS** model:
>
> * the method selects a subset of slices along each mode,
> * distances between mode-wise subspaces must be computed before unfolding,
> * unfolding destroys structural correspondences between slices and fibers.
>
> The slice-based definition therefore allows us to measure multlinear subspace distances directly on the tensor, preserving its intrinsic structure and enabling interpretable slice-level feature selection.

---

> ### Author Response · Authors · 2025-11-17
>
> > **Q2. Definition 1 is also not a standard metric, as it does not satisfy symmetric in a metric space, since $dist(S(X), S(Y)) \\neq dist(S(Y), S(X))$.**
>
> We sincerely thank the reviewer for this valuable observation. We believe the comment refers to Definition 3, which introduces the proposed multi-linear subspace distance, and we appreciate the opportunity to clarify its properties.
>
> Before addressing the specific formulation, we would like to respectfully point out that, although metric axioms such as symmetry or the condition $d(x,y)=0 \\Rightarrow x=y$ are mathematically elegant, they are often too restrictive for dimensionality reduction. In many learning scenarios, two distinct entities may legitimately be mapped to the same reduced representation, which naturally leads to zero distance even when $x \\neq y$. This behavior is not a flaw; rather, it reflects the inherent goal of dimensionality reduction, to preserve the essential structure while discarding redundancy or linearly dependent components. As a consequence, many effective distances used in subspace learning do not satisfy the full axioms of a metric, precisely because enforcing such axioms would preclude these legitimate many-to-one relationships.
>
> In this regard, it is worth noting that widely used subspace distances in the matrix setting are also non-symmetric. A prominent example is the directional distance
> \\[
> \\mathrm{dist}(\\mathrm{span}(\\mathbf{A}\_1),\\, \\mathrm{span}(\\mathbf{A}\_2))
> = \\min\_{\\mathbf{H} \\in \\mathbb{R}^{p \\times n}}
> \\| \\mathbf{A}\_1 - \\mathbf{A}\_2 \\mathbf{H} \\|\_F^2,
> \\]
> where $\\mathbf{A}\_1 \\in \\mathbb{R}^{m \\times n}$ and $\\mathbf{A}\_2 \\in \\mathbb{R}^{m \\times p}$. This formulation has been adopted in several recent subspace learning works precisely because it measures how well one subspace can be reconstructed from another, and this reconstruction direction is inherently asymmetric. The effectiveness of this distance in practice illustrates that symmetry is not a necessary requirement for evaluating information preservation in reduced spaces.
>
> Moreover, the mathematical behavior of such distances aligns closely with the goals of feature selection. Consider a data matrix $\\mathbf{X} \\in \\mathbb{R}^{n \\times d}$ and a subset $\\mathbf{B}$ of its columns. If $\\mathbf{B}$ forms a basis for the column space of $\\mathbf{X}$, then $\\mathrm{span}(\\mathbf{X}) = \\mathrm{span}(\\mathbf{B})$ even though $\\mathbf{B}$ contains fewer columns. In this situation, there exists a linear map $\\mathbf{H}$ satisfying $\\mathbf{X} = \\mathbf{B}\\mathbf{H}$, implying
> \\[
> \\mathrm{dist}(\\mathrm{span}(\\mathbf{X}),\\, \\mathrm{span}(\\mathbf{B}))
> = \\min\_{\\mathbf{H}} \\| \\mathbf{X} - \\mathbf{B}\\mathbf{H} \\|\_F^2
> = 0.
> \\]
> This is mathematically inevitable: if two subspaces coincide, then the minimum reconstruction error must be zero. From the perspective of dimensionality reduction, this property is not only expected but also essential. The purpose of feature selection is precisely to identify a compact subset of features that still spans the intrinsic subspace of the data. When such a subset is found, it must incur zero distance, regardless of the fact that $\\mathbf{B}$ has strictly lower dimensionality. Subspace learning is concerned with the equality of spans, not the equality of coordinate representations; enforcing strict metric conditions such as $d(x,y)=0 \\Rightarrow x=y$ would contradict this fundamental principle.
>
> In light of these considerations, our multi-linear subspace distance in Definition 3 generalizes the same **directional principle** to tensor data. It assesses how well the multi-way structure of the original tensor can be preserved through a mode-wise reduced representation. The **distance** is intentionally defined in a **directional** manner, analogous to the matrix case, to accurately measure reconstruction ability across modes. Imposing full metric symmetry would limit the utility and expressiveness of the formulation in the context of mode-distributed feature selection.
>
> We hope this clarification resolves the concern and highlights that the proposed distance is intentionally non-symmetric, consistent with established and theoretically well-understood practices in subspace learning, while being appropriately adapted to the multi-linear tensor setting.

---

> ### Author Response · Authors · 2025-11-17
>
> > **Q3. In Definitions 1 and 3, it contains the unknown variable alpha, thus the definition should contain alpha factor.**
>
> We thank the reviewer for flagging a potential ambiguity in the use of coefficients in Definitions 1 and 3. In the revised manuscript, we have explicitly introduced the coefficient vector \\(\\alpha^{(n)} = [\\alpha^{(n)}\_1,\\alpha^{(n)}\_2,\\dots,\\alpha^{(n)}\_{I\_n}]^\\top \\in \\mathbb{R}^{I\_n}\\) already in **Definition 1**, exactly as follows:
>
> > **Definition 1.** Let \\(\\mathcal{X} \\in \\mathbb{R}^{I\_1 \\times \\cdots \\times I\_N}\\) be an \\(N\\)-mode tensor with the mode-\\(n\\) slices \\(\\mathcal{X}^{(n)}\_1,\\dots,\\mathcal{X}^{(n)}\_{I\_n}\\), where \\(n \\in \\{1,2,\\dots,N\\}\\). The space spanned by \\(\\mathcal{X}^{(n)} = \\{\\mathcal{X}^{(n)}\_i\\}\_{i=1}^{I\_n}\\) is denoted by \\(\\mathcal{S}(\\mathcal{X}^{(n)})\\) and defined as
> > \\[
> > \\mathcal{S}(\\mathcal{X}^{(n)}) = \\left\\{ \\sum\_{i=1}^{I\_n} \\alpha^{(n)}\_i \\mathcal{X}^{(n)}\_i | \\alpha^{(n)}\_i \\in \\mathbb{R} \\right\\} = \\left\\{\\mathcal{X} \\bar{\\times}\_n \\alpha^{(n)} | \\alpha^{(n)} \\in \\mathbb{R}^{I\_n} \\right\\}.
> >\\]
> > Here, \\(\\alpha^{(n)} \\in \\mathbb{R}^{I\_n}\\) is the vector of scalar coefficients defining the span of the mode-\\(n\\) slices.
>
> With this explicit definition in place, the role of \\(\\alpha^{(n)}\\) in **Definition 3** (Multi-linear Subspace Distance) is fully unambiguous:
>
> - In Definition 3, for each fixed mode-\\(n\\) slice \\(\\mathcal{X}^{(n)}\_i\\) of the source tensor \\(\\mathcal{X}\\), the projection onto \\(\\mathcal{S}(\\mathcal{Y}^{(n)})\\) (spanned by the mode-\\(n\\) slices of the target tensor \\(\\mathcal{Y} \\in \\mathbb{R}^{I_1 \\times  \\cdots J_n \\times \\cdots \\times I_N}\\)) naturally introduces internal coefficients \\(\\alpha^{(n)}\_i \\in \\mathbb{R}^{J\_n}\\) (one coefficient vector per source slice \\(i\\)). These are not input arguments of the distance function, but are implicitly determined by the orthogonal projection:
>   \\[
>   \\operatorname{Proj}\_{\\mathcal{S}(\\mathcal{Y}^{(n)})} \\mathcal{X}^{(n)}\_i = \\mathcal{Y} \\bar{\\times}\_n \\alpha^{(n)}\_i,
>   \\]
>   where \\(\\alpha^{(n)}\_i\\) solves the least-squares problem \\(\\min\_{\\alpha^{(n)}} \\|\\mathcal{X}^{(n)}\_i - \\mathcal{Y} \\bar{\\times}\_n \\alpha^{(n)}\\|\_F\\).
>
> This is precisely analogous to classical subspace distance measures: the linear combination/projection coefficients are part of the **definition of the span and the projection operator**, but they are never listed as explicit arguments of the distance function itself.
>
> We believe that the explicit introduction of \\(\\alpha^{(n)}\\) in Definition 1, combined with the standard mathematical convention that projection coefficients are implicit, fully resolves the concern while preserving clarity and conciseness.

---

> ### Author Response · Authors · 2025-11-17
>
> > **Q4. In Definition 3, how to reflect the multilinear that claims in this paper? As it measures the distance between a row vector unfolded by regular tensor unfolding and another vector, the multilinear property is not clear.**
>
> We thank the reviewer for the thoughtful comment. Below, we restate the linear and multilinear formulations and demonstrate that the proposed distance is intrinsically multilinear in the tensor sense, not a vector-level operation.
>
> **1. Linear Versus Multilinear Subspace Learning: General Formulation**
>
> **Linear Subspace Learning (LSL).**
> Given vector samples \\(\\{x\_1,\\dots,x\_M\\}\\), \\(x\_m \\in \\mathbb{R}^I\\), LSL seeks a projection
> \\[
> U \\in \\mathbb{R}^{I \\times P},  P \\lt I,
> \\]
> such that the reduced features \\(y\_m = U^\\top x\_m \\in \\mathbb{R}^P\\) satisfy an optimality criterion.
>  A well-known example is a linear subspace distance between data matrices \\(X\\) and \\(Y\\) (whose rows are \\(x\_m\\) and \\(y\_m\\), respectively)  defined as
> \\[
> \\mathrm{dist}(X,Y) = \\min\_{H \\in \\mathbb{R}^{P \\times I}} \\|X - YH\\|\_F^2 .
> \\]
>
> **Multilinear Subspace Learning (MSL).**
> In the \\(N\\)-mode tensor setting, the data consist of tensors
> \\[
> \\{\\mathcal{X}\_1,\\dots,\\mathcal{X}\_M\\},
> \\mathcal{X}\_m \\in \\mathbb{R}^{I\_1 \\times \\cdots \\times I\_{N-1}}.
> \\]
> The goal is to learn a set of mode-wise projection matrices
> \\[
> U^{(n)} \\in \\mathbb{R}^{J\_n \\times I\_n},  J\_n \\lt I\_n
> \\]
> so that the mode-$n$ reduced feature slices $\\mathcal{Y}\_m = \\mathcal{X}\_m \\times_n  U^{(n)}$ that reduce tensor dimensionality while preserving the multi-way structure of the data, satisfy an optimality criterion.
>
>
> Inspired by the linear notion of subspace distance, we have introduced the distance between the mode-\\(n\\) subspaces of two tensors \\(\\mathcal{X}\\) and \\(\\mathcal{Y}\\) (whose frontal slices are $\\mathcal{X}\_m$ and $ \\mathcal{Y}\_m$, respectively), as
> \\[
> \\mathrm{dist}\\left(\\mathcal{S}(\\mathcal{X}^{(n)}),\\mathcal{S}(\\mathcal{Y}^{(n)})\\right)
> = \\min\_{H^{(n)} \\in \\mathbb{R}^{I\_n \\times J\_n}}
> \\left\\|
> \\mathcal{X} - \\mathcal{Y} \\times\_n H^{(n)}
> \\right\\|\_F^2 .
> \\]
> This is a direct, structure-preserving analogue of the matrix case.
>
> **2. Why the Proposed Distance Is Intrinsically Multilinear**
>
> The reviewer’s concern arises from the special case where the definition is presented only in a 3-mode setting. We have clarified this in the revised version by presenting the full \\(N\\)-mode formulation. The multilinearity becomes explicit once this general setting is stated.
>
> **(a) The subspace \\(\\mathcal{S}(\\mathcal{X}^{(n)})\\) is defined in tensor space, without unfolding**
>
> For an \\(N\\)-mode tensor \\(\\mathcal{X}\\), let
> \\[
> \\mathcal{X}^{(n)}\_i \\in \\mathbb{R}^{I\_1 \\times \\cdots \\times I\_{n-1} \\times I\_{n+1} \\times \\cdots \\times I\_N}
> \\]
> denote its mode-\\(n\\) slices. Their span is
> \\[
> \\mathcal{S}(\\mathcal{X}^{(n)}) =
> \\left\\{
> \\sum\_{i=1}^{I\_n} \\alpha^{(n)}\_i \\mathcal{X}^{(n)}\_i |
> \\alpha^{(n)} \\in \\mathbb{R}^{I\_n}
> \\right\\}.
> \\]
> This is a **multilinear** object: the slice combination is linear *along mode \\(n\\)* while *all other modes retain their full tensor structure*. No vectorization or unfolding is performed; the objects remain \\((N-1)\\)-mode tensors.
>
> **(b) The reconstruction operator \\(\\mathcal{Y}\\times\_n H^{(n)}\\) is a mode-wise multilinear map**
>
> For each slice \\(\\mathcal{X}^{(n)}\_i\\), the reconstruction
> \\[
> \\min\_{\\alpha^{(n)}\_i}
> \\left\\|\\mathcal{X}^{(n)}\_i - \\mathcal{Y}\\bar{\times}\_n \\alpha^{(n)}\_i\\right\\|\_F
> \\]
> utilizes the mode-\\(n\\) tensor–vector product \\(\\bar{\\times}\_n\\), a standard multilinear contraction that:
>
> * alters only mode \\(n\\),
> * preserves all modes \\(1,\\dots,n-1,n+1,\\dots,N\\), and
> * outputs a full \\((N-1)\\)-mode tensor.
>
> Thus, the operator is not acting on unfolded vectors; it is a mode-wise multilinear projection in tensor space.
>
> **(c) The compact form makes the multilinearity explicit**
>
> Stacking all coefficients into
> \\[
> H^{(n)} \\in \\mathbb{R}^{I\_n \\times J\_n},
> \\]
> we obtain the full tensor-space distance:
> \\[
> \\mathrm{dist}\\left(\\mathcal{S}(\\mathcal{X}^{(n)}),\\mathcal{S}(\\mathcal{Y}^{(n)})\\right)^2 =
> \\left\\|
> \\mathcal{X} - \\mathcal{Y} \\times\_n H^{(n)}
> \\right\\|\_F^2.
> \\]
>
> This expression shows that:
>
> * the distance is computed on the original tensor, not an unfolding;
> * the mapping \\(\\mathcal{Y}\\mapsto \\mathcal{Y}\\times\_n H^{(n)}\\) is **mode-wise multilinear**;
> * the global multi-way structure of the tensor is preserved throughout the distance computation.
>
>
> [1]: S. Wang, W. Pedrycz, Q. Zhu, and W. Zhu, “Subspace learning for unsupervised feature selection via matrix factorization,” Pattern Recognition, vol. 48, 10–19, 2015.
>
> [2]: H. Lu, K. N. Plataniotis, and A. N. Venetsanopoulos, “A survey of multilinear subspace learning for tensor data,” Pattern Recognition, vol. 44, 1540–1551, 2011.

---

> ### Author Response · Authors · 2025-11-17
>
> > **Q5. For the method part, I would like to say that it is a variant of HOOI with row-wise sparsity penalty and graph regularization. However, these strategies are very common and have already been discovered in many previous research works, I don't think it bring any new insights for multilinear feature selection field.**
>
> We thank the reviewer for this comment. To clarify the conceptual distinction, we begin by recalling what HOOI [1] is designed to solve.
>
>
>
> **1. What HOOI Is and What It Optimizes**
>
> Higher-Order Orthogonal Iteration (HOOI) is a Tucker-decomposition algorithm whose objective is **low-rank multilinear approximation**.
>
> Given a tensor \\(\\mathcal{X} \\in \\mathbb{R}^{I\_1 \\times \\cdots \\times I\_N}\\) and Tucker ranks \\(r\_1,\\ldots,r\_N\\), HOOI solves
>
> \\[
> \\min\_{{U^{(n)}},\\mathcal{G}}
> \\left\\|\\mathcal{X} - \\mathcal{G} \\times\_1 U^{(1)} \\times\_2 U^{(2)} \\cdots \\times\_N U^{(N)}\\right\\|\_F^2,
> \\]
>
> where:
>
> * each \\(U^{(n)} \\in \\mathbb{R}^{I\_n \\times r\_n}\\) has **orthonormal real-valued columns**
>   \\((U^{{(n)}^\\top} U^{(n)} = I\_{r\_n})\\),
>
> * \\(\\mathcal{G}\\in\\mathbb{R}^{r\_1 \\times \\cdots \\times r\_N}\\) is the core tensor,
>
> * and HOOI alternates over mode-wise least-squares subproblems via truncated SVD.
>
>
> **2. The MSLFS Objective Is Mathematically Different**
>
> In contrast, **MSLFS is not a Tucker model and is not a low-rank approximation of all modes**.
> The generalized MSLFS objective for an \\(N\\)-mode tensor is:
>
>
> \\begin{align*}
> \&\\min\_{\\substack{
> H^{(n;m\_n)} \\in \\mathbb{R}\_{+}^{I\_n \\times m\_n},\
> W^{(n;m\_n)} \\in \\mathbb{R}^{m\_n \\times I\_n}\_{+}
> }}
> \\frac{1}{2}
> \\left\\|\\mathcal{X} - \\mathcal{X}
> \\times\_{n=1}^{N-1}
> H^{(n;m\_n)} W^{(n;m\_n)}
> \\right\\|\_F^2 \\\\
> \&\\text{s.t. } \\quad
> W^{(n;m\_n)} {W^{(n;m\_n)}}^\\top = I\_{m\_n},
> \\quad n = 1,\\ldots,N-1.
> \\end{align*}
>
>
> **Key structural differences:**
>
> 1. **MSLFS never reduces mode \\(N\\)**, because mode \\(N\\) corresponds to *samples*.
>    HOOI reduces *all* modes, including the sample mode, which violates the goal of feature selection.
>
> 2. **MSLFS performs feature-mode reduction only on modes \\(1,\\ldots,N-1\\)**, which correspond to feature dimensions.
>    This is a different multilinear structure from Tucker decomposition.
>
> 3. **MSLFS uses indicator matrices \\(W^{(n;m_n)}\\)** that select *representative mode-\\(n\\) slices*.
>    These indicator matrices select \\(m\_n\\) slices that best span the space of mode-\\(n\\) slices.
>    The intersection of selected slices across modes yields the representative mode-\\(N\\) fibers (i.e., features).
>
> 4. **The core tensor in MSLFS is fixed as \\(\\mathcal{X}\\) (the data tensor itself)**.
>    In HOOI, the core tensor \\(\\mathcal{G}\\) is an independently learned latent variable, whereas in MSLFS the core tensor is simply \\(\\mathcal{X}\\) itself and is not learned at all. It is important to note that \\(\\mathcal{G}\\) is not obtained by selecting $r\_n$ mode-$n$ slices of \\(\\mathcal{X}\\) along its modes.
>    This difference alone makes HOOI structurally incapable of performing feature selection.
>
>
>
> **3. Why HOOI Cannot Be Used for Feature Selection**
>
> Although both methods seem to “reduce dimensionality,” the mathematical operations differ fundamentally:
>
> * **HOOI produces rotated, dense, orthonormal subspaces.**
>   These do *not* correspond to selecting actual slices or fibers; they are mixtures of all coordinates.
>
> * **MSLFS produces coordinate-aligned slice selections** through \\(W^{(n;m\_n)}\\),
>   something impossible in Tucker models, since the basis vectors in HOOI are continuous orthogonal directions.
>
> * In HOOI, the product \\(\\mathcal{X} \\times\_n U^{(n)\\top}\\) forms projections, in which each \\(U^{(n)}\\) is an *orthonormal projection matrix*, not an indicator matrix for mode-$n$ slice selection.
>
>   Therefore, the columns of \\(U^{(n)}\\) do **not correspond to explicit feature indices**.
>
> * In MSLFS, the selection operators \\(W^{(n;m\_n)}\\) are restricted to row-orthogonal indicator structures and coupled with coefficient matrices \\(H^{(n;m\_n)}\\).
>   The product \\(H^{(n;m\_n)} W^{(n;m\_n)}\\) is **not** orthogonal and is designed to reconstruct the spans of feature slices, not to form an orthonormal mode-wise basis.
>
> Because Tucker decomposition **mixes** all features, it fundamentally destroys interpretability and cannot identify representative slice subsets.
>
> Given these structural differences:
> **MSLFS cannot be expressed as HOOI with sparsity or graph regularization.**
> They solve different problems, over different feasible sets, using different tensor operations.
>
> We also note that we have clarified the mathematical novelty of MSLFS’s mode-wise sparsity and graph regularization compared to common alternatives in our response to Reviewer 4’s second comment.
>
> [1] : Bernard N. Sheehan and Yousef Saad. Higher Order Orthogonal Iteration of Tensors (HOOI) and
> Its Relation to PCA and GLRAM, pp. 355–365. 2007.

---

### Official Review · Reviewer_aeXk · 2025-11-05

**Soundness:** 3
**Presentation:** 3
**Contribution:** 3
**Rating:** 6
**Confidence:** 2

**Summary:**

This paper presents a novel tensor feature selection method, i.e., Multi-Linear Subspace Learning Feature Selection (MSLFS), which addresses key limitations of existing approaches in handling multi-modal tensor data. These limitations include the neglect of inter-modal dependencies, high computational costs, and the absence of a principled criterion for preserving global structure. The MSLFS framework employs a distributed selection strategy that constructs informative feature subsets through intersections of representative slices from each mode. A central contribution is the introduction of a multi-linear subspace distance, which provides a quantitative measure of how well the selected features preserve the original tensor's global multi-way geometry. Furthermore, MSLFS incorporates both a joint sparsity constraint and a higher-order graph constraint to maintain the integrity of the overall tensor structure while preserving local neighborhood relationships.

**Strengths:**

1.	This paper has a solid theoretical foundation and clear interpretability.
2.	The framework and methodology of this paper are complete, with a well-integrated objective function, a detailed optimization process, and a comprehensive complexity analysis.
3.	The experiments presented in this paper are solid and reliable. Extensive experiments on various benchmark datasets demonstrate that its performance consistently outperforms existing methods, and this is supported by comprehensive ablation experiments and visualization results.

**Weaknesses:**

1.	The related work section lacks adequate discussion of recent advances (2024–2025), featuring only one work from 2025.
2.	This model includes multiple hyperparameters (e.g., α, β, and the k-nearest neighbor algorithm). Although sensitivity analysis is provided, it will be subject to certain limitations in practical applications.

**Questions:**

1.	The related work section lacks adequate discussion of recent advances (2024–2025), featuring only one work from 2025. It would be beneficial to incorporate additional analyses of recent literature.
2.	When the number of tensor modes exceeds 3, does the definition of multilinear subspace distance need to be adjusted? Can MSLFS be directly extended? Are additional regularization or structural assumptions required?
3.	The tensors mentioned in Appendix 7.6 may have negative values. Therefore, the MSLFS method is extended. Has this new update rule been experimentally verified? Will it affect model performance?

---

> ### Author Response · Authors · 2025-11-17
>
> > **Q1. The related work section lacks adequate discussion of recent advances (2024–2025), featuring only one work from 2025. It would be beneficial to incorporate additional analyses of recent literature.**
>
> We thank the reviewer for the insightful comment. In response, we have significantly expanded the related work section to include a more comprehensive discussion of recent advances in unsupervised feature selection (2024–2025). Specifically, we now include six notable linear and nonlinear models from the matrix-based unsupervised feature selection domain. In contrast, only two tensor-based frameworks have been proposed recently, despite their substantial potential for handling tensor data. This gap motivated the development of our method, **MSLFS**, which leverages the full multi-way structure of tensor data without resorting to vectorization, thereby preserving critical cross-mode dependencies. Below, we summarize the most recent advancements:
>
> **Matrix-Based Unsupervised Feature Selection**.
> Unsupervised feature selection in vectorized data has been actively researched. Key models include:
>
> * **ESUFS** [1]: A method that addresses key challenges in structured-graph approaches by introducing a discrete structured-graph model and a data discrepancy learning module to improve graph structure and emphasize naturally discriminative features without labels.
>
> * **UFS-CGL** [2]: A unified framework that overcomes earlier graph-guided methods' limitations by retaining class-specific information and using contrastive learning to capture richer discriminative relationships, alongside an $\ell_{1,2}$-regularized projection matrix for feature selection.
>
> * **FOG-R** [3]: A flexible framework that uses graph learning to replace rigid linear projections in spectral-based feature selection, addressing the limitations of traditional low-dimensional projections in capturing the true data manifold.
>
> * **MRMGRFS** [4]: A model that balances relevance and redundancy in feature selection. It combines spectral clustering and Jensen–Shannon divergence to refine relevance scores and minimize global redundancy.
>
> * **NNSE** [5]: This model replaces traditional linear projection with neural networks, capturing richer feature–label relationships. It integrates adaptive graph regularization to preserve local structure and enhance representational power.
>
> * **SDAE** [6]: A deep autoencoder-based framework that jointly performs feature selection and reconstruction. It identifies the most informative features to reduce dimensionality while preserving the global structure.
>
> **Tensor-Based Unsupervised Feature Selection**.
> Tensor-based methods have emerged to address the limitations of vectorized feature selection for multi-dimensional data. Notable recent models include:
>
> * **CPUFS**: This method preserves the multi-dimensional structure of tensor data through graph-regularized nonnegative CP decomposition, combined with tensor-oriented pseudo-label regression and feature selection.
>
> * **GRLTR**: A low-rank tensor representation framework that integrates graph regularization to preserve both global structure and local geometry, operating directly on tensor data to maintain its inherent multi-dimensional properties.
>
> We believe this expanded discussion now provides a more comprehensive overview of both recent matrix- and tensor-based methods in unsupervised feature selection, highlighting the unique contributions of **MSLFS** in effectively addressing the challenges of tensor data analysis.
>
> [1]: Pei Huang, Zhaoming Kong, Limin Wang, Xuming Han, and Xiaowei Yang. Efficient and Stable
> Unsupervised Feature Selection Based on Novel Structured Graph and Data Discrepancy Learn-
> ing. IEEE Transactions on Neural Networks and Learning Systems, 36(4):6229–6243, 2025.
>
> [2]: Qian Zhou, Qianqian Wang, Quanxue Gao, Ming Yang, and Xinbo Gao. Unsupervised Discrimina-
> tive Feature Selection via Contrastive Graph Learning. IEEE Transactions on Image Processing,
> 33:972–986, 2024.
>
> [3]: Hong Chen, Feiping Nie, Rong Wang, and Xuelong Li. Unsupervised Feature Selection with Flexi-
> ble Optimal Graph. IEEE Transactions on Neural Networks and Learning Systems, 35(2):2014–
> 2027, 2024.
>
> [4]: Xianyu Zuo, Wenbo Zhang, Xiangyu Wang, Lanxue Dang, Baojun Qiao, and Yadi Wang. Unsu-
> pervised Feature Selection via Maximum Relevance and Minimum Global Redundancy. Pattern
> Recognition, 164:111483, 2025.
>
> [5]: Mengbo You, Aihong Yuan, Dongjian He, and Xuelong Li. Unsupervised Feature Selection via
> Neural Networks and Self-Expression with Adaptive Graph Constraint. Pattern Recognition,
> 135:109173, 2023b.
>
> [6]: Wael Hassanieh and Abdallah Chehade. Selective Deep Autoencoder for Unsupervised Feature
> Selection. In Proceedings of the Thirty-Eighth AAAI Conference on Artificial Intelligence and
> Thirty-Sixth Conference on Innovative Applications of Artificial Intelligence and Fourteenth Sym-
> posium on Educational Advances in Artificial Intelligence. AAAI Press, 2024b.

---

> > ### Author Response · Authors · 2025-11-17
> >
> > > **The continuation of the answer to the Q1**
> >
> > We now outline the theoretical and practical limitations of existing approaches and demonstrate these limitations empirically. The first set of limitations pertains to matrix-based methods. To illustrate them, we consider a tensor dataset of size $\mathbb{R}^{I_1 \times I_2 \times I_3}$, where $I_3$ denotes the number of samples and $I_1 \times I_2$ constitutes the multi-dimensional feature space; however, these limitations naturally generalize to tensors of arbitrary order $N$.
> >
> > Matrix-based methods begin by vectorizing each sample into an $I_1I_2$-dimensional vector and stacking these vectors as rows of a data matrix, effectively operating on the mode-3 unfolding of size $\mathbb{R}^{I_1I_2 \times I_3}$. This preprocessing step constitutes a major limitation, as it destroys the intrinsic multi-way structure of tensor data, including cross-mode dependencies, thereby preventing both linear and neural network–based matrix methods from capturing the true relationships within the data. As a result, their performance on downstream tasks such as clustering is inevitably degraded. After vectorization, these methods typically learn a feature-weighting matrix of size $\mathbb{R}^{p \times I_1I_2}$ (with $p < I_1I_2$) and select the top $p$ features by computing the column-wise $\ell_2$ norms.
> >
> > Tensor-based approaches, in contrast, operate directly on the multi-way structure. However, existing methods such as **CPUFS** and **GRLTR**, despite effectively modeling multi-dimensional interactions, ultimately rely on the same feature-selection mechanism as matrix-based approaches: they still construct a $\mathbb{R}^{p \times I_1I_2}$ feature-weighting matrix and select features via column-wise $\ell_2$ norms. Consequently, they inherit two core limitations: (1) the computational burden of searching across the entire $I_1I_2$ feature space, and (2) the inability to exploit cross-mode dependencies during feature selection, since features are still treated in their vectorized form.
> >
> > Our proposed **MSLFS** method overcomes these issues through a distributed, mode-aware feature-selection strategy. Instead of selecting from all $I_1I_2$ features simultaneously, **MSLFS** identifies $m_1 \le p$ representative mode-1 slices and $m_2 \le p$ representative mode-2 slices such that $m_1 m_2 = p$. The intersections of these slices yield the mode-3 fibers that form the final set of $p$ informative features, eliminating the need to evaluate the full vectorized feature space.  The effectiveness of this selection approach is established by **Theorem 3.1**. This design not only reduces the computational complexity of **MSLFS** to scale linearly with both the number of samples and features but also preserves and leverages the inherent multi-way dependencies of tensor data during selection. These are precisely the limitations that **MSLFS** is designed to address in unsupervised feature selection.

---

> ### Author Response · Authors · 2025-11-17
>
> > **Q2. When the number of tensor modes exceeds 3, does the definition of multilinear subspace distance need to be adjusted? Can MSLFS be directly extended? Are additional regularization or structural assumptions required?**
>
> We thank the reviewer for this insightful question. Yes, our framework naturally extends to tensors of arbitrary order $N \\gt 3$, as well as to general multi-linear subspace learning, without requiring additional structural assumptions or regularizations. Below, we provide a mathematically rigorous generalization to the $N$-mode setting.
>
> We first extend the core definitions to $N$-mode tensors.
>
> **Definition 1.** Let $\\mathcal{X} \\in \mathbb{R}^{I\_1 \times \cdots \times I\_N}$ be an $N$-mode tensor with mode-$n$ slices $\mathcal{X}^{(n)}\_1, \dots, \mathcal{X}^{(n)}\_{I\_n}$, $n \in \{1,\dots,N\}$. The span of the mode-$n$ slices is
> \\[
> \\mathcal{S}(\\mathcal{X}^{(n)}) = \\left\\{ \\sum\_{i=1}^{I\_n} \\alpha\_i^{(n)} \\mathcal{X}^{(n)}\_i | \\alpha^{(n)} = [\\alpha\_1^{(n)}, \\dots, \\alpha\_{I\_n}^{(n)}]^\\top \\in \\mathbb{R}^{I\_n} \\right\\}.
> \\]
>
> **Definition 2.** For a tensor $\\mathcal{Z}$ matching the size of a mode-$n$ slice of $\mathcal{X}$, the distance to the mode-$n$ subspace is
> \\[
> \\mathrm{dist}(\\mathcal{Z}, \\mathcal{S}(\\mathcal{X}^{(n)})) = \\min\_{\\mathcal{W} \\in \\mathcal{S}(\\mathcal{X}^{(n)})} \\|\mathcal{Z} - \\mathcal{W}\\|\_F.
> \\]
>
> **Definition 3 (Multi-linear Subspace Distance).** For two $N$-mode tensors $\\mathcal{X}$ and $\\mathcal{Y}$ (with mode-$n$ size $J\_n$ for $\\mathcal{Y}$), the squared mode-$n$ subspace distance is
> \\[
> \\mathrm{dist}^2(\\mathcal{S}(\\mathcal{X}^{(n)}), \\mathcal{S}(\\mathcal{Y}^{(n)})) = \\sum\_{i=1}^{I\_n} \\mathrm{dist}^2(\\mathcal{X}^{(n)}\_i, \\mathcal{S}(\\mathcal{Y}^{(n)})).
> \\]
> This admits the compact form
> \\[
> \\mathrm{dist}^2(\\mathcal{S}(\\mathcal{X}^{(n)}), \\mathcal{S}(\\mathcal{Y}^{(n)})) = \\|\\mathcal{X} - \\mathcal{Y} \\times\_n \\mathbf{H}^{(n)}\\|\_F^2,
> \\]
> where $\\mathbf{H}^{(n)} \\in \\mathbb{R}^{I\_n \\times J\_n}$ has rows $\\alpha\_i^{(n)\\top}$ that minimize each slice distance.
>
>
> The core generalization is given by the following theorem.
>
> **Theorem 1.** Let $\\mathcal{X} \\in \\mathbb{R}^{I\_1 \\times \\cdots \\times I\_N}$. For each $n=1,\\dots,N-1$, suppose $\\mathcal{S}(\\mathcal{X}^{(n)})$ has basis dimension $R\_n \le I\_n$
> with index set $T\_n = \\{i\_1^{(n)}, \\dots, i\_{R\_n}^{(n)}\\}$ and indicator matrix $\\mathbf{W}^{(n;R\_n)} \\in \\{0,1\\}^{R\_n \\times I\_n}$. Let $\\mathbf{f}\_{i\_1,\\dots,i\_{N-1}} = \\mathcal{X}\_{i\_1,\\dots,i\_{N-1},:} \\in \\mathbb{R}^{I\_N}$ be the mode-$N$ fiber.
>
> **(Part I: Core Dictionary).** The $\\prod\_{n=1}^{N-1} R\_n$ intersection fibers
> \\[
> \\{\\mathbf{f}\_{i\_{r\_1}^{(1)}, \\dots, i\_{r\_{N-1}}^{(N-1)}} \\}\_{r\_1=1,\\dots,R\_1}^{r\_{N-1}=1,\\dots,R\_{N-1}}
> \\]
> form a core dictionary spanning all mode-$N$ fibers. Stacking them yields
> \\[
> \\mathbf{F}\_{\\mathrm{core}} = \\big(\\mathcal{X} \\times\_{n=1}^{N-1} \\mathbf{W}^{(n;R\_n)}\\big)_{(N)} = \\mathbf{X}\_{(N)} \\bigotimes\_{n=1}^{N-1} \mathbf{W}^{(N-n;R\_{N-n})^\\top}.
> \\]
>
> **(Part II: Separable Reconstruction).** There exist $\\mathbf{H}^{(n;R\_n)} \\in \\mathbb{R}^{I\_n \\times R\_n}$, $n=1,\dots,N-1$, such that every fiber expands separably:
> \\[
> \\mathbf{f}\_{i\_1,\\dots,i\_{N-1}} = \\sum\_{r\_1=1}^{R\_1} \\cdots \\sum\_{r\_{N-1}=1}^{R\_{N-1}} \\left( \\prod\_{n=1}^{N-1} h^{(n;R\_n)}\_{i\_n, r\_n} \\right) \\mathbf{f}\_{i\_{r\_1}^{(1)}, \\dots, i\_{r\_{N-1}}^{(N-1)}},
> \\]
> and the mode-$N$ unfolding satisfies
>
> \\[
> \\mathbf{X}\_{(N)} = \\mathbf{F}\_{\\mathrm{core}} \\bigotimes\_{n=1}^{N-1} \\mathbf{H}^{(N-n;R\_{N-n})^\\top}.
> \\]
>
> **Proof.** See Appendix 7.2 for the detailed derivation.
>
> The **MSLFS** objective for $N$-mode data is then
> \\begin{align*}
> \&\\min\_{\\mathbf{H}^{(n;m\_n)}, \\mathbf{W}^{(n;m\_n)} \\ge 0} \\;
> \\frac{1}{2} \\left\\|\\mathcal{X} - \\mathcal{X} \\times\_{n=1}^{N-1} \\mathbf{H}^{(n;m\_n)} \\mathbf{W}^{(n;m\_n)}\right\\|\_F^2 \\\\
> \&\\quad + \\frac{\\alpha}{2} \\prod\_{n=1}^{N-1} \\mathrm{Tr}\\left( \\mathbf{H}^{(N-n;m\_{N-n})^\\top} \\mathbf{L}^{(N-n)} \\mathbf{H}^{(N-n;m\_{N-n})} \\right) \\\\
> \&\\quad + \\frac{\\beta}{2} \\prod\_{n=1}^{N-1} \\mathrm{Tr}\\left( \\mathbf{W}^{(N-n;m\_{N-n})} \\mathbf{U}^{(N-n)} \\mathbf{W}^{(N-n;m\_{N-n})^\\top} \\right)
> \\end{align*}
> subject to
> \\[
> \\bigotimes\_{n=1}^{N-1} \\mathbf{W}^{(N-n;m\_{N-n})} \\mathbf{W}^{(N-n;m\_{N-n})^\\top} = \\mathbf{I}\_{m\_1 \\dots m\_{N-1}}.
> \\]
>
> Thus, the method scales seamlessly to higher-order tensors.
>
> We further note that MSLFS has already been tested on higher-order real-valued multi-way dataset, with the results provided in our response to the reviewer's third comment.

---

> ### Author Response · Authors · 2025-11-17
>
> > **Q.3 The tensors mentioned in Appendix 7.6 may have negative values. Therefore, the MSLFS method is extended. Has this new update rule been experimentally verified? Will it affect model performance?**
>
> We thank the reviewer for the detailed comment. In response, we would like to clarify that **MSLFS** has already been experimentally validated on real-valued multi-way datasets, including both non-negative and general real-valued tensors. To further evaluate the performance of **MSLFS** in these different settings, we have conducted additional experiments with four large, real-valued and non-negative datasets:
>
> * **Real-Valued Datasets**:
>
>   * **COVID-19 Systems Serology**: $(\mathbb{R}^{20 \times 35 \times 74 \times 5200})$ (5 classes)
>   * **Kinetic Fluorescence**: $(\mathbb{R}^{64 \times 12 \times 10 \times 10000})$ (7 classes)
>
> * **Non-negative Datasets**:
>
>   * **PneumoniaMNIST**: $(\mathbb{R}^{224 \times 224 \times 5800})$ (2 classes)
>   * **OrganCMNIST**: $(\mathbb{R}^{224 \times 224 \times 20000})$ (11 classes)
>
> In these datasets, the last mode represents the number of samples, and the initial modes encode multi-dimensional features.
>
> * **Experimental Validation**
>
> To evaluate the performance of **MSLFS** in both real-valued and non-negative contexts, we compared it not only with the 10 previously examined state-of-the-art methods but also with three recent deep learning–based feature selection approaches: **SAE**, **NNSE**, and **SDAE**. We evaluated the methods using clustering accuracy, and measured per-iteration CPU time on the new datasets.
>
> * **Runtime Efficiency**
>
> We evaluated the per-iteration CPU time on datasets with more than 5,000 samples, as presented in Table 1. **MSLFS** is the most computationally efficient method, outperforming both matrix-based and tensor-based baselines. For example, on the **COVID-19 Systems Serology** dataset, **MSLFS** completes each iteration in 31.3 seconds, significantly faster than **ILFS**, **FSPCA**, and deep-learning-based methods like **SAE**, **NNSE**. The gap grows even larger on the **Kinetic Fluorescence** dataset, where **MSLFS** is the only method to stay under 130 seconds per iteration, while other baselines exceed 180 seconds or fail due to memory limitations.
>
> This efficiency is attributed to **MSLFS**'s mode-wise slice selection and separable reconstruction strategy, which avoid costly global operations, reduce redundancy, and scale linearly with respect to both sample size and mode-specific feature dimensions
>
> **Table 1: Per-iteration CPU time (seconds ↓) comparison across methods on datasets with >5,000 samples. "OM" = out-of-memory.**
>
> | **Methods**     | **COVID-19 Systems Serology** | **PneumoniaMNIST** | **Kinetic Fluorescence** | **OrganCMNIST** |
> |-----------------|:-----------------------------:|:------------------:|:------------------------:|:---------------:|
> | **UDFS**        | 101.2                         | 156.6              | 385.3                    | OM              |
> | **SAE**         | 72.1                          | 106.7              | 211.3                    | 389.4           |
> | **ILFS**        | 46.5                          | 64.5               | 144.2                    | 237.8           |
> | **GRLTR**       | 161.9                         | 257.6              | OM                       | OM              |
> | **CAE**         | 66.3                          | 87.7               | 143.9                    | 256.7           |
> | **FSPCA**       | 45.4                          | 78.9               | 156.2                    | 281.5           |
> | **CPUFS**       | 56.7                          | 89.5               | 154.6                    | 303.1           |
> | **SPCAFS**      | 74.4                          | 103.6              | 164.2                    | 266.4           |
> | **NNSE**        | 97.3                          | 142.2              | 186.7                    | 314.8           |
> | **GRSSLFS**     | 125.6                         | 184.4              | 343.1                    | OM              |
> | **SDAE**        | 86.7                          | 102.5              | 193.4                    | 299.3           |
> | **SPDFS**       | 89.7                          | 112.4              | 232.1                    | 356.2           |
> | **MSLFS**       | **31.3**                      | **57.2**           | **124.5**                | **232.4**       |

---

> ### Author Response · Authors · 2025-11-17
>
> > **The continuation of the answer to the Q3.**
>
> * **Clustering Accuracy**
>
> Tables 1 and 2 show the clustering results, where **MSLFS** consistently outperforms all existing methods across various datasets. Specifically, **MSLFS** achieves the highest ACC and NMI across both real-valued and non-negative datasets. While some baselines perform well on individual datasets, **MSLFS** demonstrates robustness and superior overall performance, particularly in clustering tasks, where its ability to preserve global structure is critical.
>
> **Table 1: Clustering Accuracy (ACC ↑) of MSLFS vs. 13 state-of-the-art methods**
>
> | Model                    | Kinetic Fluorescence | PneumoniaMNIST | OrganCMNIST | COVID-19 Serology |
> |--------------------------|:--------------------:|:--------------:|:-----------:|:-----------------:|
> | LS (NeurIPS 2005)        | 78.86                | 58.33          | 48.02       | 80.55             |
> | UDFS (IJCAI 2011)        | 72.99                | 62.17          | 43.49       | 78.72             |
> | SAE (IJCNN 2017)         | 77.63                | 62.98          | 51.09       | 86.74             |
> | ILFS (CVPR 2017)         | 76.97                | 59.90          | 52.49       | 71.47             |
> | GRLTR (JVCIR 2018)       | 84.30                | 64.54          | 58.65       | 89.62             |
> | CAE (ICML 2019)          | 83.45                | 62.47          | 61.31       | 84.45             |
> | FSPCA (NeurIPS 2020)     | 82.66                | 62.29          | 59.35       | 77.84             |
> | CPUFS (TPAMI 2022)       | 84.24                | 67.71          | 56.43       | 85.45             |
> | SPCAFS (TPAMI 2023)      | 81.67                | 59.43          | 51.20       | 80.34             |
> | NNSE (PATCOG 2023)       | 83.74                | 63.43          | 62.60       | 84.81             |
> | GRSSLFS (TMLR 2024)      | 82.99                | 60.17          | 62.03       | 87.48             |
> | SDAE (AAAI 2024)         | 85.73                | 68.67          | **75.69**   | 88.54             |
> | SPDFS (TPAMI 2025)       | 83.18                | 63.86          | 55.34       | 76.54             |
> | **MSLFS (Ours)**         | **88.67**            | **71.65**      | 73.76       | **95.34**         |
> | **Improvement**          | **+2.94**            | **+2.98**      | -           | **+5.72**         |
>
> **Table 2: Normalized Mutual Information (NMI ↑) of MSLFS vs. 13 state-of-the-art methods**
>
> | Model                    | Kinetic Fluorescence | PneumoniaMNIST | OrganCMNIST | COVID-19 Serology |
> |--------------------------|:--------------------:|:--------------:|:-----------:|:-----------------:|
> | LS (NeurIPS 2005)        | 77.66                | 58.17          | 51.79       | 81.12             |
> | UDFS (IJCAI 2011)        | 76.74                | 63.23          | 45.51       | 77.54             |
> | SAE (IJCNN 2017)         | 79.77                | 66.84          | 49.56       | 86.69             |
> | ILFS (CVPR 2017)         | 74.11                | 61.65          | 51.22       | 72.34             |
> | GRLTR (JVCIR 2018)       | 83.39                | 62.73          | 57.69       | 91.27             |
> | CAE (ICML 2019)          | 82.72                | 63.66          | 61.70       | 88.35             |
> | FSPCA (NeurIPS 2020)     | 84.98                | 60.74          | 58.02       | 80.76             |
> | CPUFS (TPAMI 2022)       | 85.76                | 67.12          | 55.85       | 87.00             |
> | SPCAFS (TPAMI 2023)      | 80.29                | 63.49          | 50.69       | 81.51             |
> | NNSE (PATCOG 2023)       | 84.89                | 64.96          | 58.59       | 86.23             |
> | GRSSLFS (TMLR 2024)      | 84.31                | 62.31          | 60.54       | 85.77             |
> | SDAE (AAAI 2024)         | 86.41                | 68.04          | **74.80**   | 89.82             |
> | SPDFS (TPAMI 2025)       | 80.51                | 64.74          | 53.63       | 80.73             |
> | **MSLFS (Ours)**         | **90.22**            | **72.34**      | 72.65       | **95.88**         |
> | **Improvement**          | **+3.81**            | **+4.30**      | -           | **+4.61**         |
>
> * **Conclusion**
>
> In summary, we have experimentally verified **MSLFS** on both real-valued and non-negative multi-way datasets. The results demonstrate that **MSLFS** is not only effective in terms of clustering accuracy but also computationally efficient, outperforming several state-of-the-art methods. These experiments confirm the robustness of **MSLFS** across various types of datasets, supporting its applicability in real-valued and non-negative tensor settings.

---

### Author Response · Authors · 2025-11-28
**Request to Reviewers to Check Response Rebuttal**

Dear Reviewers,

Thank you again for the time and effort you have dedicated to reviewing our work and for the valuable insights you provided. We hope that our rebuttal has satisfactorily addressed the concerns raised in your initial review.

As the discussion period draws to a close, we kindly ask you to consider whether our responses have resolved your concerns. If so, we would greatly appreciate it if you could reconsider your initial evaluation in light of the clarifications and additional details we have provided.

Sincerely, The Authors

---

### Meta-Review · Area_Chair_6P5P · 2026-01-03

**Summary:**

Reviewer **aeXk** thinks the key contribution of this paper is the introduction of a multi-linear subspace distance, which provides a quantitative measure of how well the selected features preserve the original tensor's global multi-way geometry. Reviewer **aeXk**'s main concern is that this paper lacks adequate discussion of recent advances and the existence of multiple hyperparameters.

The authors have expanded the related work section to include a more comprehensive discussion of recent advances in unsupervised feature selection (2024–2025). But the authors did not respond to the hyperparameter setting issue.

Reviewer **LNCn** also recognizes the contribution of the proposed multi-linear subspace distance. The major concerns of Reviewer **LNCn** include the definition of the spanned space and the novelty of the proposed method (as it highly relates to HOOI).

I believe the authors have provided a good explanation of the spanned space. But the relationship between HOOI and the proposed method is not well illustrated.

Reviewer **Z2uY** recognizes the novelty of this paper. The main concerns of Reviewer **Z2uY** include the limitation of the proposed method to 3-order tensors and its sensitivity to hyperparameters. Most of the questions posed by Reviewer **Z2uY** are well responded to by the authors.

Reviewer **ADWM** thinks this paper is an incremental work and lacks the comparison of running time. The authors have provided additional experiments on runtime efficiency, and prove that the proposed model is efficient in general.

**Reviewer Concerns:**

The concerns about hyper-parameter setting remain, but the authors have well addressed the limitations of the proposed method to 3rd-order tensors and its running time efficiency.

**Reviewer Scores:**

I think both Reviewer **LNCn** and **ADWM** may raise their rating from 2 to 4 or 6.

---

### Decision · Program_Chairs · 2026-01-26

Reject